# Deciphering the critical role of interstitial volume in glassy sulfide superionic conductors

Han Su [1,2], Yu Zhong [1] ✉, Changhong Wang [2,3] ✉, Yu Liu[1,2], Yang Hu [2], Jingru Li[1], Minkang Wang[1], Longan Jiao[4], Ningning Zhou[4], Bing Xiao[4], Xiuli Wang[1], Xueliang Sun [2,3] ✉ & Jiangping Tu [1] ✉

Sulfide electrolytes represent a crucial category of superionic conductors for all-solid-state lithium metal batteries. Among sulfide electrolytes, glassy sulfide is highly promising due to its long-range disorder and grain-boundary-free nature. However, the lack of comprehension regarding glass formation chemistry has hindered their progress. Herein, we propose interstitial volume as the decisive factor influencing halogen dopant solubility within a glass matrix. We engineer a $Li_3PS_4$-$Li_4SiS_4$ complex structure within the sulfide glassy network to facilitate the release of interstitial volume. Consequently, we increase the dissolution capacity of LiI to 40 mol% in $75Li_2S$-$25P_2S_5$ glass. The synthesized glass exhibits one of the highest ionic conductivities among reported glass sulfides. Furthermore, we develop a glassy/crystalline composite electrolyte to mitigate the shortcomings of argyrodite-type sulfides by utilizing our synthesized glass as the filler. The composite electrolytes effectively mitigate Li intrusion. This work unveils a protocol for the dissolution of halogen dopants in glass electrolytes.

All-solid-state lithium metal batteries (ASSLMBs) with enhanced safety and high energy density have been proposed as highly competitive contenders to conventional lithium-ion batteries (LIBs)[1,2]. Central to ASSLMBs are solid electrolytes (SEs), which govern the transport of charge carriers within the battery[3,4]. Among the diverse families of SEs, sulfide solid electrolytes (SSEs) stand out as the most promising due to their superior ionic conductivity and suitable Young's modulus[5–8]. Glassy sulfides in SSEs offer distinct advantages over their crystalline and glass-ceramic counterparts[9]. Firstly, the disorder characteristic of glass facilitates the incorporation of various species over a wide range, allowing for the optimization of electrolyte performance in terms of ionic conductivity, chemical/interfacial stability, and mechanical properties. Secondly, the absence of grain boundaries in glass enables the easy attainment of high relative density in the cold-pressed pellet of glassy SSEs[10,11]. According to fracture mechanics, a pellet with high relative density will have a flawless surface, effectively preventing Li metal intrusion[12–15]. Moreover, glassy SSEs can eliminate electronic conductivity resulting from heterogeneous grain boundaries[16,17], thus suppressing Li dendrite growth within SEs[18].

The history of glassy sulfides dates back to the 1980s when they were found to exhibit significantly enhanced ionic conductivity in contrast to glassy oxides[19,20]. Substantial efforts have since been dedicated to investigating the structure and Li$^+$ conducting mechanisms of glassy sulfides. In sulfide glass, a typical glassy network forms through the chemical reaction between glass formers (e.g., $P_2S_5$, $SiS_2$, $B_2S_3$) and glass modifiers (e.g., $Li_2S$). Glass formers constitute the glass

[1]State Key Laboratory of Silicon and Advanced Semiconductor Materials, Key Laboratory of Advanced Materials and Applications for Batteries of Zhejiang Province, School of Materials Science and Engineering, Zhejiang University, Hangzhou 310027, China. [2]Department of Mechanical and Materials Engineering, University of Western Ontario1151 Richmond St., London, ON N6A 3K7, Canada. [3]Eastern Institute for Advanced Study, Eastern Institute of Technology, Ningbo, Zhejiang 315200, PR China. [4]Carl Zeiss (Shanghai) Co., Ltd., 60 Mei Yue Road, Pilot Free Trade Zone, Shanghai 200131, PR China. ✉e-mail: yu_zhong@zju.edu.cn; cwang@eitech.edu.cn; xsun9@uwo.ca; tujp@zju.edu.cn

network, while glass modifiers break the sulfur chain, facilitating the incorporation of charge carrier (Li⁺) into the glass matrix. Elemental chemistry-based optimization is commonly employed to improve the chemical and electrochemical stability of glassy sulfides. For instance, replacing the glass former $P_2S_5$ with $SnS_2$ or $In_3S_2$ can enhance the chemical stability of electrolytes according to the soft-hard-acid-base theory[21]. Regarding Li⁺ conducting kinetics in the glass, factors such as Li⁺ concentration, Li⁺-anion binding energy, and strain energy for structure deformation are all considered within the framework of the Anderson and Stuart energy barrier model[22]. Accordingly, there are two practical methods to improve the ionic conductivity of glassy sulfides: (1) increasing the Li⁺ concentration, and (2) tuning the chemistry and network structure of glass to weaken Li⁺-anion binding

and enhance structure-deforming ability. Over the past decades, numerous strategies have been reported to pursue high-performance glassy sulfides. For example, enhancing the relative ratio of $Li_2S$ to increase the Li⁺ concentration in the $Li_2S$-$P_2S_5$ glass has proven effective in facilitating Li⁺ transport[23,24]. Additionally, mixing glass forming/modifying anions or cations in binary glass systems to form a ternary glass system with modified Li⁺-anion binding strength and structure-deforming capability has yielded favorable results in reducing the activation energy for Li⁺ motion[25,26]. Among these strategies, the incorporation of halogen dopants LiX (X = Cl, Br, I) has been particularly desirable due to its ability to simultaneously improve ionic conductivity and Li/electrolyte interfacial stability[27–29]. The enhancing mechanisms are that the increase of Li⁺ concentration and the

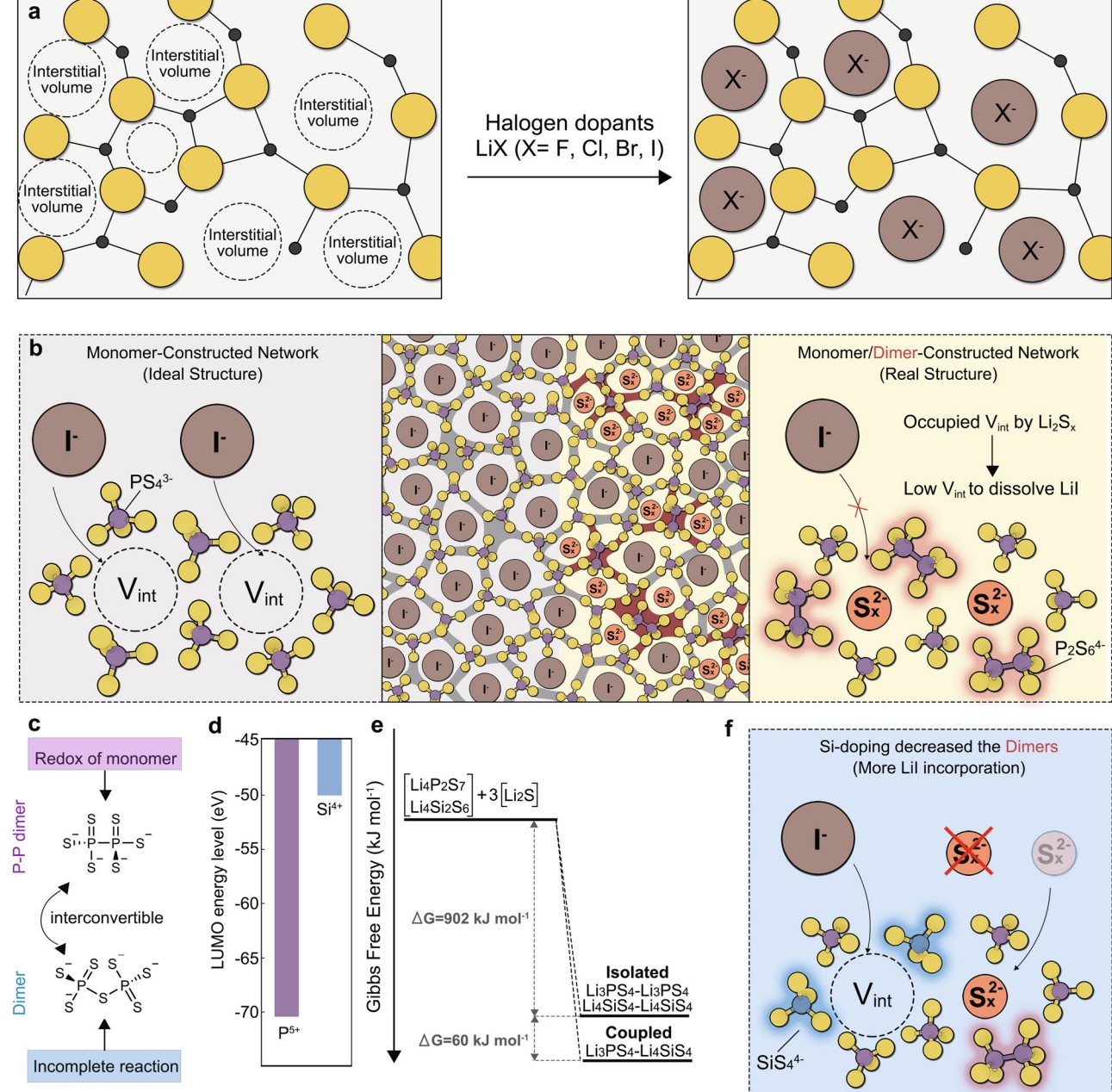

**Fig. 1 | Analysis on the halogen dopants' dissolving behavior in the glass sulfide.** **a** Incorporating process of halogen dopants in the glass network. **b** The schematic illustration of ideal and realistic LiI-doped $75Li_2S$-$25P_2S_5$ glass network structure. **c** The formation mechanism of P-P dimers and dimers. **d** The LUMO value of $P^{5+}$ and $Si^{4+}$. **e** The reaction Gibbs free energy from precursors to isolated $Li_3PS_4$/$Li_4SiS_4$ and coupled $Li_3PS_4$/$Li_4SiS_4$. **f** The schematic diagram of the Si-doping effect.

weakened Li$^+$-anion binding in glass by LiX doping can accelerate Li$^+$ transport. Additionally, the LiX-rich solid electrolyte interphase (SEI) formed through the decomposition of LiX-doped glassy SSE can passivate the Li/SSE interface[30,31].

Recent studies have shown that halogen dopants exhibit distinct behavior compared to traditional glass formers or modifiers, as they interstitially dissolve within the glass network (Fig. 1a)[32]. Furthermore, earlier studies discovered that the ionic conductivity and Li/electrolyte interfacial stability of LiX-incorporated sulfide glass are positively correlated with the concentration of halogen dopants in the glass[28,33]. This intriguing observation indicates that increasing the solubility of LiX can further optimize glassy SSEs to achieve unprecedented levels of ionic conductivity and Li/electrolyte interfacial stability. Previous studies have reported that the dissociation energy of the halogen dopants (solute) significantly influences their solubility. Consequently, LiI with the lowest dissociation energy demonstrates the highest solubility among LiX[33]. Additionally, the properties of the glass network (solvent) and their interaction with the halogen dopants also have a significant impact on the dissolution process of LiX. However, few studies have delved into this intricate interplay. Addressing this knowledge gap could provide a viable approach to improve the solubility of halogen dopants, thereby advancing the boundaries of ionic conductivity and Li/electrolyte interfacial stability in glassy SSEs. Therefore, an in-depth deciphering of the dissolution process of halogen dopants is imperative.

In this work, we propose the interstitial volume of glass (V$_{int}$) as a vital factor influencing the dissolution behavior of halogen dopants within a glassy network. We first conduct theoretical simulations to elucidate the structural chemistry of 75Li$_2$S-25P$_2$S$_5$ glassy SSEs doped with LiI. Our findings reveal that lithium sulfide or lithium polysulfide (Li$_2$S$_x$), generated by side reactions during synthesis, can occupy interstitial sites, thus impeding LiI dissolution at these sites. To address this issue, we modify the reaction pathways during glass formation by constructing a stable Li$_3$PS$_4$-Li$_4$SiS$_4$ complex structure to suppress the formation of Li$_2$S$_x$. As a result, the dissolution capacity of LiI in 75Li$_2$S-25P$_2$S$_5$ glass reaches a value of 40 mol%. The resulting glassy electrolytes exhibit one of the highest ionic conductivity values of 2.21 × 10$^{-3}$ S cm$^{-1}$ among glassy sulfides. When applied as the SE directly, this glassy SSE exhibits a decent critical current density (CCD) value (1.2 mA cm$^{-2}$ with a capacity of 1.2 mAh cm$^{-2}$). Moreover, we design a glassy/crystalline composite electrolyte to mitigate the low relative density and high electronic conductivity issues of crystalline argyrodite-type SSEs, using our synthesized glass as the filler. The composite electrolyte exhibits a notable CCD value (2.9 mA cm$^{-2}$ with a capacity of 2.9 mAh cm$^{-2}$). Additionally, the Li symmetric cell employing the composite electrolyte demonstrates remarkably stable cycling for 3000 h at a current density of 1 mA cm$^{-2}$ and an areal capacity of 1 mAh cm$^{-2}$. Even under more stringent conditions, with a current density of 1 mA cm$^{-2}$ and an areal capacity of 3 mAh cm$^{-2}$, it remains functional for 900 h. Furthermore, the Li ‖ LiNi$_{0.83}$Co$_{0.12}$Mn$_{0.05}$O$_2$ (NCM83125) full cells utilizing the composite electrolyte retain 82.4% of its initial discharging capacity after 500 cycles at a current density of 0.25 mA cm$^{-2}$. The deciphering of halogen dissolution chemistry presented in this study offers a pragmatic outlook for the development of versatile and high-performance glass sulfide materials in the future.

## Results and discussion

### The competition mechanism of halogen dopants and Li$_2$S$_x$ in interstitial volume

The structural chemistry of the state-of-the-art 75Li$_2$S-25P$_2$S$_5$ glassy SSEs, represented by the simplified formula Li$_3$PS$_4$, has been well established. As previously reported[24], its glass network mainly comprises ortho-thiophosphate (PS$_4$$^{3-}$, monomer) and minor quantities of hypo-thiodiphosphate (P$_2$S$_6$$^{4-}$, P-P dimer), along with pyro-thiophosphate (P$_2$S$_7$$^{4-}$, dimer) (Fig. S1). Then, to investigate the changes in 75Li$_2$S-25P$_2$S$_5$ glass structure after LiI incorporation, we performed ab initio molecular dynamic (AIMD) simulations of 75Li$_2$S-25P$_2$S$_5$ glass and LiI-doped 75Li$_2$S-25P$_2$S$_5$ glass by melting and rapid-quenching crystalline Li$_3$PS$_4$ and Li$_4$PS$_4$I (Fig. S2). Our simulation results suggest that the addition of LiI has a minimal effect on the chemistry of the glass network (Figs. S3 and S4). Both 75Li$_2$S-25P$_2$S$_5$ glass and LiI-doped 75Li$_2$S-25P$_2$S$_5$ glass exhibit a similar glass network primarily consisting of PS$_4$$^{3-}$, with small quantities of P$_2$S$_7$$^{4-}$ and P$_2$S$_6$$^{4-}$. Also, our simulation results confirm the insight from the previous report that the doped LiI does not replace the sulfur in the PS$_4$$^{3-}$, P$_2$S$_7$$^{4-}$ or P$_2$S$_6$$^{4-}$, but dissolves interstitially in the glass network[32]. According to the law of conservation, the formation of P$_2$S$_6$$^{4-}$ and P$_2$S$_7$$^{4-}$ leads to the generation of Li$_2$S$_x$, which occupies the interstitial positions within glass network rather than altering its structure. Since both LiI and Li$_2$S$_x$ occupy interstitial sites, the V$_{int}$ filled by Li$_2$S$_x$ hinders the additional dissolution of LiI. Ideally, in a LiI-doped 75Li$_2$S-25P$_2$S$_5$ glass with the simplified formula Li$_3$PS$_4$-xLiI, a large V$_{int}$ would be available for the dissolution of LiI if the glass network consisted solely of PS$_4$$^{3-}$. However, side reactions occurring during the actual preparation process inevitably lead to the formation of P$_2$S$_6$$^{4-}$ and P$_2$S$_7$$^{4-}$, resulting in a hybrid network comprising both monomers and dimers. This results in a relatively smaller V$_{int}$ compared to that of a PS$_4$$^{3-}$-based glass network for the incorporation of LiI, as Li$_2$S$_x$ occupies a certain volume (Fig. 1b).

### Design of high-performance glassy sulfides

Recognizing the competition of interstitial positions between LiI and Li$_2$S$_x$, it is apparent that reducing the relative amount of Li$_2$S$_x$ in the glass system will increase the available V$_{int}$, thereby facilitating further dissolution of LiI. The formation of Li$_2$S$_x$ species stems from the formation chemistry of P$_2$S$_6$$^{4-}$ and P$_2$S$_7$$^{4-}$ during the preparation process, necessitating the tracing of such reactions (Fig. 1c). The formation of P$_2$S$_6$$^{4-}$ results from a redox reaction between P$^{5+}$ and S$^{2-}$ (Equation (1)). Given the stability of P(+V)$_2$S$_5$ molecules at room temperature, it is plausible to infer that the generation of P(+IV)$_2$S$_6$$^{4-}$ is induced by localized overheating resulting from the ball-milling process employed in synthesizing glassy electrolytes. Besides, according to the theory of electron transfer, the impact of dipole moments on electron transfer reactions is significant[34,35]. Increasing dipole moments of the reactant or aligning the dipole with the direction of electron transfer can facilitate electron transfer reactions. Considering this, the significant dipole moments (5.905 Debye) induced by unsaturated P = S bonds in Li$_3$PS$_4$ molecules may act as inducers for the acceleration of redox reactions (Fig. S5). Regarding the formation of P$_2$S$_7$$^{4-}$, it can be induced by reversible reactions between P$_2$S$_6$$^{4-}$ and Li$_2$S$_x$ (Equation (2)), as well as incomplete reactions between glass formers and glass modifiers. Consequently, a strategy aimed at reducing the ratios of Li$_2$S$_x$ in 75Li$_2$S-25P$_2$S$_5$ systems should prioritize decreasing the potential for self-redox of Li$_3$PS$_4$ and promoting reactions between glass formers and glass modifiers.

$$2Li_3PS_4 \rightarrow Li_4P_2S_6 + (x-2)/(x-1)Li_2S + 1/(x-1)Li_2S_x (\text{redox reaction}) \quad (1)$$

$$Li_4P_2S_6 + Li_2S_x \leftrightarrow Li_4P_2S_7 + Li_2S_{x-1} (\text{reversible}) \quad (2)$$

Accordingly, we propose a modification of the glass network by incorporating SiS$_2$. Firstly, replacing partial P$_2$S$_5$ with SiS$_2$, whose cations possess higher lowest unoccupied molecular orbital (LUMO) values (Fig. 1d) can reduce the overall occurrence of redox reactions in the glass system. Moreover, the Li$_4$SiS$_4$ molecule without unsaturated bonds exhibits a significantly lower dipole moment of 2.973 Debye compared to that of the Li$_3$PS$_4$ molecule. After the incorporation of Li$_4$SiS$_4$, the Li$_3$PS$_4$-Li$_4$SiS$_4$ complex displays a dipole moment of 6.025 Debye, while the dipole moment of Li$_3$PS$_4$ in the complex is decreased

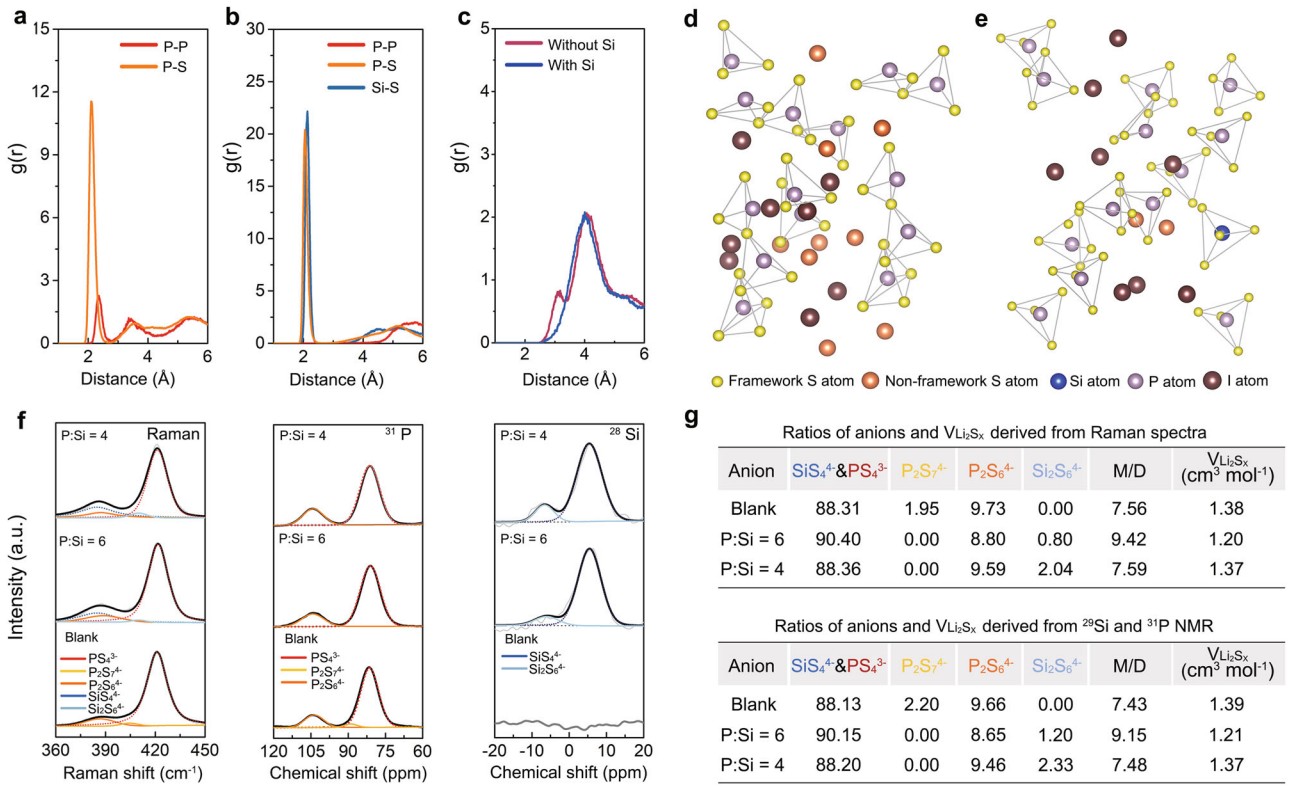

**Fig. 2 | Investigations of anion clusters in the sulfide glass after the Si incorporation. a** The g(r) profiles of P-P and P-S pairs in the equilibrium $Li_4PS_4I$ glass after AIMD simulations. **b** The g(r) profiles of P-P, P-S, and Si-S pairs in the equilibrium Si-doped $Li_{4.125}Si_{0.125}P_{0.875}S_4I$ glass after AIMD simulations. **c** The g(r) profiles of the framework S-I pair in $Li_4PS_4I$ glass (without Si) and Si-doped $Li_{4.125}Si_{0.125}P_{0.875}S_4I$ glass (with Si). **d, e** The snapshot of the equilibrium $Li_4PS_4I$ glass after AIMD simulations (**d**) and the snapshot of the equilibrium Si-doped $Li_{4.125}Si_{0.125}P_{0.875}S_4I$ glass after AIMD simulations (**e**). **f** Raman, $^{31}P$ and $^{29}Si$ ss-NMR spectra of the blank, P:Si = 6 and P:Si = 4 electrolytes. **g** Ratios of anion clusters and minimum $V_{Li2sx}$ derived from Raman, $^{31}P$, and $^{29}Si$ ss-NMR spectra. (M/D represents the ratio of monomers to dimers).

from 5.905 to 4.760 Debye (Fig. S6). This result indicates that the charge compensation resulting from the coordination of $Li^+$ in $Li_4SiS_4$ with the S atom in the P = S bond will help to decrease the dipole moment of $Li_3PS_4$, which may potentially suppress electron transfer reactions between $P^{5+}$ and $S^{2-}$. Subsequently, as illustrated in Fig. 1e, the interaction between $Li_3PS_4$ and $Li_4SiS_4$ lowers the overall Gibbs free energy, aiding in the transition from precursors to monomers. Considering the above factors, Si-doping will reduce the amounts of $P_2S_6^{4-}$ and $P_2S_7^{4-}$ in the glassy sulfide, thereby lowering the proportion of $Li_2S_x$ to free up more $V_{int}$ for the incorporation of LiI (Fig. 1f). Furthermore, appropriate doping levels of $SiS_2$ in the $Li_2S-P_2S_5$ system will not result in a significant deterioration of the Li/electrolyte interface, as previously reported[36,37]. Accordingly, a series of Si-doped $0.6((75 + 0.5x)Li_2S-(25-0.5x)P_2S_5-xSiS_2)-40LiI$ electrolytes were synthesized. The investigated systems are thereby denoted as follows: blank (without $SiS_2$ doping), P:Si = 7 (x = 6.25), P:Si = 6 (x = 7.14), P:Si = 5 (x = 8.33), P:Si = 4 (x = 10).

To validate the proposed design approach, atomistic simulations using the AIMD technique were conducted. Given that our AIMD simulation methodology relies on the melt-quenching of crystalline materials, we select $Li_4PS_4I$, which contains the highest reported LiI content among crystalline sulfides, for our simulations. Despite the LiI content (33.3 mol%) in $Li_4PS_4I$ being lower than that in our experimental samples (40 mol%), the changes in the glassy framework structure reflected during the simulation process can still provide valuable insights. As illustrated in Fig. 2a, the $PS_4^{3-}$ polyhedral structure in $Li_4PS_4I$ glass undergoes significant disruption upon equilibration at 600 K, which is evident from the sharp peaks located at ~2.30 Å and 3.60 Å of the P-P pair and the broad peak observed in the range of

3.70–4.50 Å of the P-S pair in the corresponding g(r) profiles. The presence of $P_4S_7^{4-}$ and $P_4S_6^{4-}$ can be inferred from these observations. Then, the impact of Si doping on I-rich $Li_4PS_4I$ glass was investigated using $Li_{4.125}Si_{0.125}P_{0.875}S_4I$ as a representative precursor, as shown in Fig. S7. As revealed in Fig. S8, the introduction of $SiS_4^{4-}$ successfully stabilizes the initial $PS_4^{3-}$ structure, even at a high temperature up to 2000 K. Consequently, the $Li_{4.125}Si_{0.125}P_{0.875}S_4I$ glass exhibits a predominantly monomeric structure upon equilibration at 600 K, as evidenced by the absence of observable dimers in the g(r) profiles (Fig. 2b), thereby validating that Si doping can effectively suppress dimers' formation. Moreover, to ensure that the pair distribution results are independent of the input structures used in our modeling, two additional validation structures for both $Li_4PS_4I$ and $Li_{4.125}Si_{0.125}P_{0.875}S_4I$ were constructed and applied to the consistent molecular simulations, respectively (Fig. S9). As a result, comparable outcomes are observed across varying structures of the same materials (Fig S10). Furthermore, while the g(r) profile of the Li-I pair indicates complete dissociation of LiI in both $Li_4PS_4I$ and $Li_{4.125}Si_{0.125}P_{0.875}S_4I$ glass (Fig. S11), snapshots in Fig. 2d, e after equilibrium reveal that, compared to $Li_{4.125}Si_{0.125}P_{0.875}S_4I$, $Li_4PS_4I$ exhibits significant occupancy of $S_x^{2-}$ in the interstitial sites. Considering the limited interstitial space, the presence of these non-framework S atoms will compress I atoms and cause them to move toward the framework due to the coulombic repulsion. By extracting the radial distribution between S atoms from the framework and I atoms, a trend of closer proximity between certain I atoms and the framework S is observed in $Li_4PS_4I$ glass (Fig. 2c), which may hinder further LiI incorporation.

Furthermore, experimental verification of the changes in anion clusters due to Si doping was conducted using Raman spectra

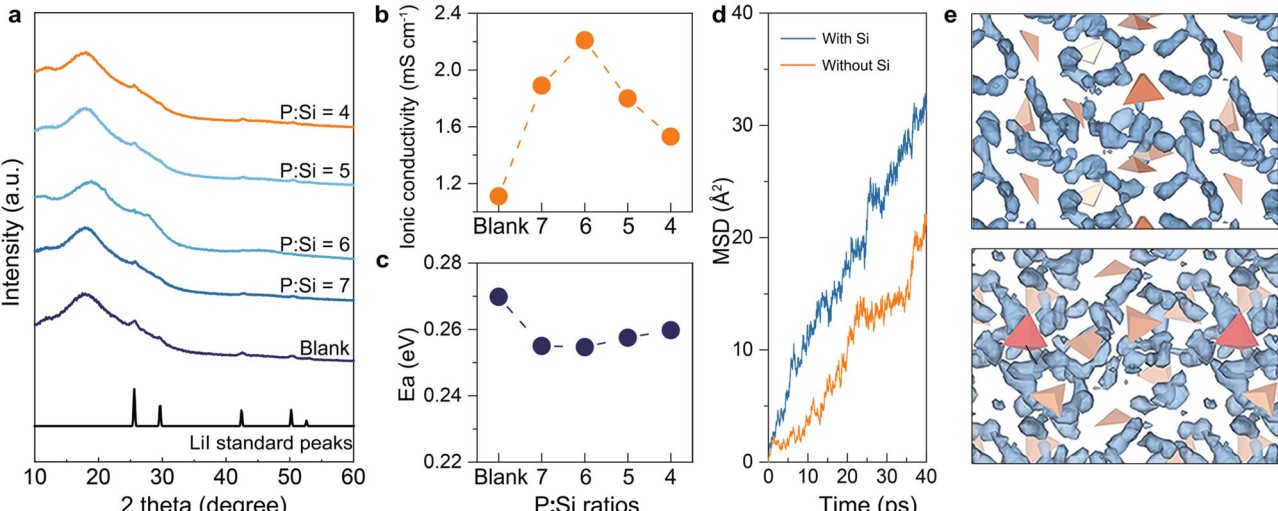

**Fig. 3 | Phase characterizations and ion transport capability of the synthesized electrolytes. a** The XRD spectra of synthesized series electrolytes. **b** The ionic conductivity of synthesized series electrolytes. **c** The Li$^+$ conduction activation energy of synthesized series electrolytes. **d** The lithium MSD plot for Li$_4$PS$_4$I (orange) and Li$_{4.125}$Si$_{0.125}$P$_{0.875}$S$_4$I (blue) glass at 600 K from AIMD simulations. **e** The Li-ion trajectory (blue) in Li$_4$PS$_4$I (top) and Li$_{4.125}$Si$_{0.125}$P$_{0.875}$S$_4$I (bottom) glass at 600 K from AIMD simulations. The orange tetrahedra represents PS$_4^{3-}$, the red tetrahedra represents SiS$_4^{4-}$.

combined with $^{31}$P and $^{29}$Si solid-state nuclear magnetic resonance (ss-NMR) spectra in blank, P:Si = 6, and P:Si = 4 electrolytes (Fig. 2f). Based on prior Raman/ss-NMR studies on Si and P-based sulfides[24,38], three peaks corresponding to PS$_4^{3-}$ (421.0 cm$^{-1}$ in Raman, 83.1 ppm in $^{31}$P NMR), P$_2$S$_6^{4-}$(386.9 cm$^{-1}$ in Raman, 105.0 ppm in $^{31}$P NMR) and P$_2$S$_7^{4-}$(404.5 cm$^{-1}$ in Raman, 88.9 ppm in $^{31}$P NMR) are observed in the blank electrolyte in both Raman and ss-NMR spectra. Upon partial substitution of P$^{5+}$ by Si$^{4+}$, the P$_2$S$_7^{4-}$ peak gradually disappears accompanied by the emergence of SiS$_4^{4-}$ (385.0 cm$^{-1}$ in Raman, 0.5 ppm in $^{29}$Si NMR) and Si$_2$S$_6^{4-}$ (410 cm$^{-1}$, −11.7 ppm in $^{29}$Si NMR)[29,38]. Then, ratios of anion clusters in different electrolytes were derived using the relative peak area in Raman spectra and the quantitative analysis of $^{31}$P and $^{29}$Si ss-NMR spectra in the part of methods. As demonstrated in Fig. 2g, Si-doping effectively lowers the ratio of P$_2$S$_7^{4-}$ and P$_2$S$_6^{4-}$, as anticipated by our strategy. However, excess Si incorporation (P:Si = 4) promotes considerable Si$_2$S$_6^{4-}$ formation, which in turn decreases the ratio of monomers. Therefore, the maximum ratio of PS$_4^{3-}$ and SiS$_4^{4-}$ monomers is reached when the P to Si ratio is 6. By assuming Li$_2$S$_2$ with the smallest volume among lithium polysulfides as the accompanied product of Li$_4$P$_4$S$_6$, and Li$_2$S as the accompanied product of Li$_4$P$_2$S$_7$ and Li$_4$Si$_2$S$_6$, the minimum molar volume of Li$_2$S$_x$ in each electrolyte can be derived. The minimum molar volume of Li$_2$S$_x$ in the P:Si = 6 electrolyte is~86% of that in the blank electrolyte, indicating a great amount of V$_{int}$ is released to dissolve more LiI.

## Physical and electrochemical properties of synthesized electrolytes

X-ray diffraction (XRD) analysis was conducted to elucidate the dissolving behavior of LiI in a series of synthesized electrolytes. As depicted in Fig. 3a, the XRD spectrum of the blank electrolyte showcases prominent peaks of LiI at 25.7°, 29.6°, 42.3°, and 50.2°. Upon Si doping, the intensity of these sharp LiI peaks diminishes notably in the P:Si = 7 electrolyte and disappears entirely in the P:Si = 6 electrolyte. However, excessive Si doping fails to facilitate the efficient integration of LiI into the glassy network, as evidenced by the limited release of interstitial volume (V$_{int}$) in the spectroscopic analyses described above. Consequently, faint LiI peaks are discernible in the XRD spectra of the P:Si = 5 and P:Si = 4 electrolytes. These observations suggest that a precipitate-free glassy sulfide with a high LiI concentration is only attained at the P:Si = 6 composition.

The resulting P:Si = 6 electrolyte exhibits a glass-transition temperature (T$_g$) of 132 °C and a crystallization temperature of 206 °C, as determined by differential scanning calorimetry (DSC) analysis (Fig. S12). Additionally, the pellet of the P:Si = 6 electrolyte displays a flat and compact surface following cold-pressing (Fig. S13a). By hot-pressing the pellet at T$_g$ overnight, a highly densified pellet with a relative density of ~100% was obtained (Fig. S13b). Plus, the XRD result of the hot-pressing pellet indicates that the glass nature of the sample remains unaltered after hot-pressing (Fig. S14). Then, the relative density of the cold-pressed P:Si = 6 pellet is determined to be 93.6% by comparing the density of the cold-pressed and highly densified pellet (Table S1). Furthermore, energy dispersive spectroscopy (EDS) mapping results demonstrate a homogenous distribution of elements within both the micro-particle and the pellet of the P:Si = 6 electrolyte (Figs. S15–S16).

The ionic conductivity and Li$^+$ conduction activation energy (E$_a$) of the synthesized electrolytes were evaluated using temperature-dependent electrochemical impedance spectroscopy (EIS) analysis. The relevant data, including weight, thickness, Nyquist plots, and Arrhenius plots, are provided in Table S2, Fig. S17, and Fig. S18, respectively. As depicted in Fig. 3b, the optimal ionic conductivity value of 2.21 mS cm$^{-1}$ is achieved at a P:Si ratio of 6, which represents a twofold increase compared to the blank electrolyte and ranks one of the highest among reported glassy sulfides (Table S3). Furthermore, at the P:Si ratio of 6 (Fig. 3c), the lowest E$_a$ value of 0.253 eV is also attained. On one hand, the incorporation of LiI into the glassy P:Si = 6 electrolyte, instead of existing as an impurity phase, leads to an increase in the Li$^+$ concentration, thereby enhancing Li transport kinetics. On the other hand, Si doping itself may also significantly contribute to facilitating Li$^+$ conduction. As demonstrated in Fig. 3d, after relaxation for 40 ps at 600 K, the lithium mean square displacement (MSD) value of Li$_{4.125}$Si$_{0.125}$P$_{0.875}$S$_4$I glass is higher than that of Li$_4$PS$_4$I glass, indicating a more unrestrained environment for Li$^+$ motion enabled by Si doping. Besides, compared with Li$_4$PS$_4$I glass, Li$_{4.125}$Si$_{0.125}$P$_{0.875}$S$_4$I glass not only exhibits an overall improvement in Li$^+$ transport dynamics but also presents a particularly promoted Li$^+$ transport in proximity to SiS$_4$ tetrahedra positions, which may be attributed to the increased Li$^+$ concentration near SiS$_4$ tetrahedra and entropy-stabilized effect of Si doping (Fig. 3e)[39]. Additionally, the Li$^+$ diffusion kinetics in Li$_4$PS$_4$I,

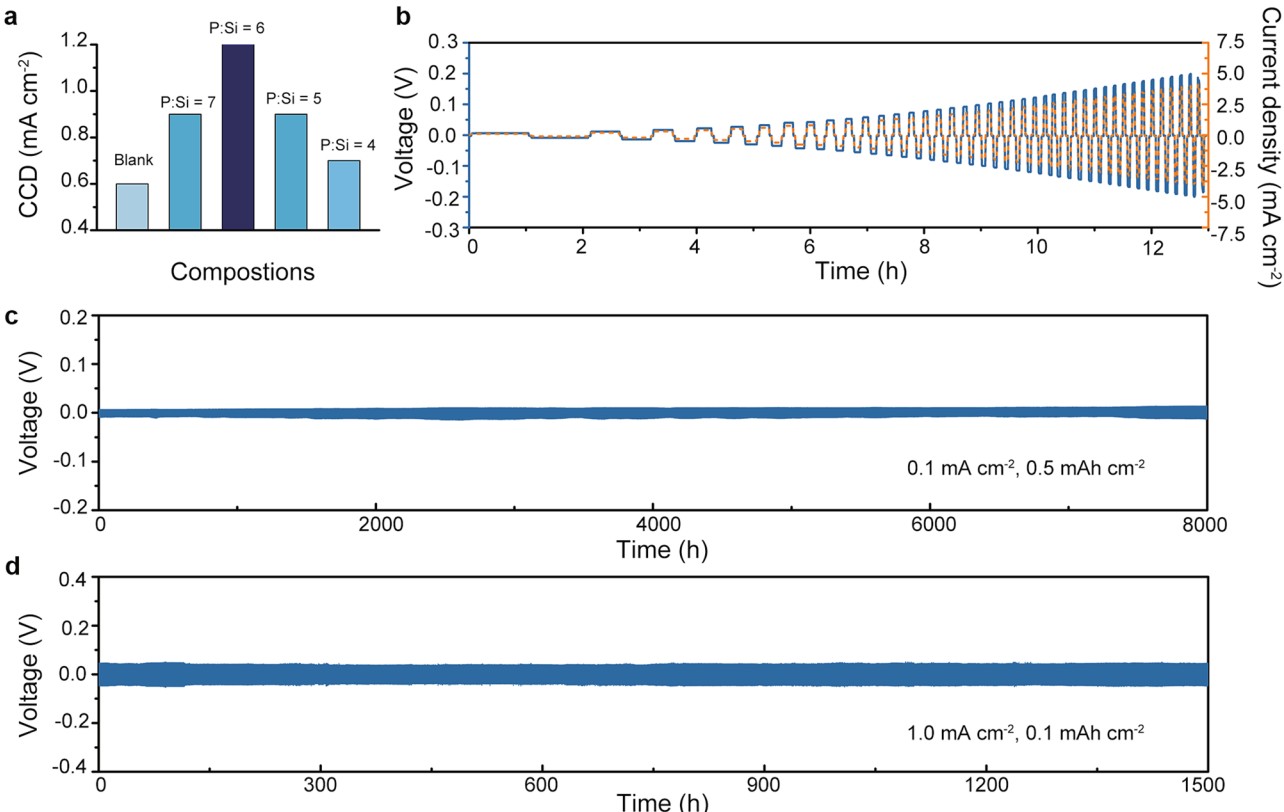

**Fig. 4 | Li compatibility evaluation for the P:Si = 6 electrolyte. a** The summary of CCD of synthesized electrolytes. **b** Galvanostatic Li plating/stripping profiles of the Li symmetric cell employing the P:Si = 6 electrolyte at step-increased current densities. The capacity is fixed to be 0.1 mAh cm⁻². **c** Galvanostatic Li plating/stripping profiles of the Li symmetric cell employing the P:Si = 6 electrolyte at a current density of 0.1 mA cm⁻² and a cut-off capacity of 0.5 mAh cm⁻². **d** Galvanostatic Li plating/stripping profiles of the Li symmetric cell employing the P:Si = 6 electrolyte at a current density of 1.0 mA cm⁻² and a cut-off capacity of 0.1 mAh cm⁻².

$Li_{4.125}Si_{0.125}P_{0.875}S_4I$, and $Li_{3.125}Si_{0.125}P_{0.875}S_4$ were compared (Fig. S19). It can be observed that solely Si or LiI doping is insufficient to elevate lithium transport dynamics to the level exhibited by $Li_{4.125}Si_{0.125}P_{0.875}S_4I$ glass with both Si and LiI doping. Therefore, it can be concluded that both the effects of LiI dissolution and Si incorporation on the Li⁺ transport kinetics should be taken accounts to explain the significant enhancement of ionic conductivity in our optimal glass. Plus, to avoid the influence of the specific input configuration on the results of diffusion rates, the Li⁺ diffusion kinetics in two additional structures of $Li_4PS_4I$ and $Li_{4.125}Si_{0.125}P_{0.875}S_4I$ glass were also calculated. As Fig S20 illustrates, distinct structures of the identical material exhibit analogous Li⁺ diffusion kinetics, which confirms the reliability of this theoretical finding.

The Li intrusion suppression capability of the synthesized electrolytes was assessed via CCD tests with a current step size of 0.1 mA cm⁻² and a consistent charging/discharging duration of 1 h for each step (Fig. S21). As demonstrated in Fig. 4a, the blank electrolyte with significant LiI impurity demonstrates a CCD value of 0.6 mA cm⁻², which is comparable to previously reported values[33]. With Si doping, the CCD value exhibits a trend of increase followed by a decrease, peaking at the P:Si ratio of 6, where it reaches the highest value of 1.2 mA cm⁻². Remarkably, the CCD value of the P:Si = 6 electrolyte stands out as highly competitive among glassy sulfides (Table S4). Furthermore, to comprehensively assess the dendrite suppression ability of the P:Si = 6 electrolyte, galvanostatic charging/discharging test with fixed capacity was conducted to eliminate the effects of void-formation[40,41]. The P:Si = 6 electrolyte exhibits stable functionality even under current density escalation up to 4.0 mA cm⁻² when the discharging/charging capacity is maintained at 0.1 mAh cm⁻², indicating its ability to withstand demanding operational conditions, provided severe contact loss is avoided (Fig. 4b). However, it is important to note that the value derived from this fixed-capacity test may not fully represent the practical CCD of the SE, where a larger capacity (>1.0 mAh cm⁻²) is required. Long-term galvanostatic discharging/charging tests were conducted on Li symmetric cells using the P:Si = 6 electrolyte to demonstrate the cycling stability. At a low current density of 0.1 mA cm⁻², the slow Li plating/stripping process ensures prompt creeping of Li and sustains the contact between Li and the P:Si = 6 electrolyte. Consequently, the Li symmetric cell using the P:Si = 6 electrolyte displayed stable cycling behavior for 8000 h even at a high cut-off capacity of 0.5 mAh cm⁻² (Fig. 4c). Additionally, at a low cut-off capacity of 0.1 mAh cm⁻², when the volume change of Li metal is insignificant, the symmetric cell with the P:Si = 6 electrolyte demonstrates stable operation for over 1500 h at a current density of 1.0 mA cm⁻² (Fig. 4d). The cycling stability exhibited by Li symmetric cells employing the P:Si = 6 electrolyte is also highly competitive when compared to previously reported instances utilizing glassy electrolytes as the interlayer (Table S5).

### The comparison between glassy P:Si = 6 and state-of-the-art sulfides

The state-of-the-art argyrodite sulfide $Li_6PS_5Cl$ and the $75Li_2S$-$25P_2S_5$ glassy electrolyte, commonly used as interlayers for Li metal, were selected as comparison groups to further evaluate the performance of the P:Si = 6 electrolyte. According to the EIS spectra at 25 °C (Fig. S22), the ionic conductivity of $75Li_2S$-$25P_2S_5$ and $Li_6PS_5Cl$ is 0.46 mS cm⁻¹ and 2.04 mS cm⁻¹, respectively, which are consistent with the previous research[42,43] and lower than that of the P:Si = 6 electrolyte. Additionally,

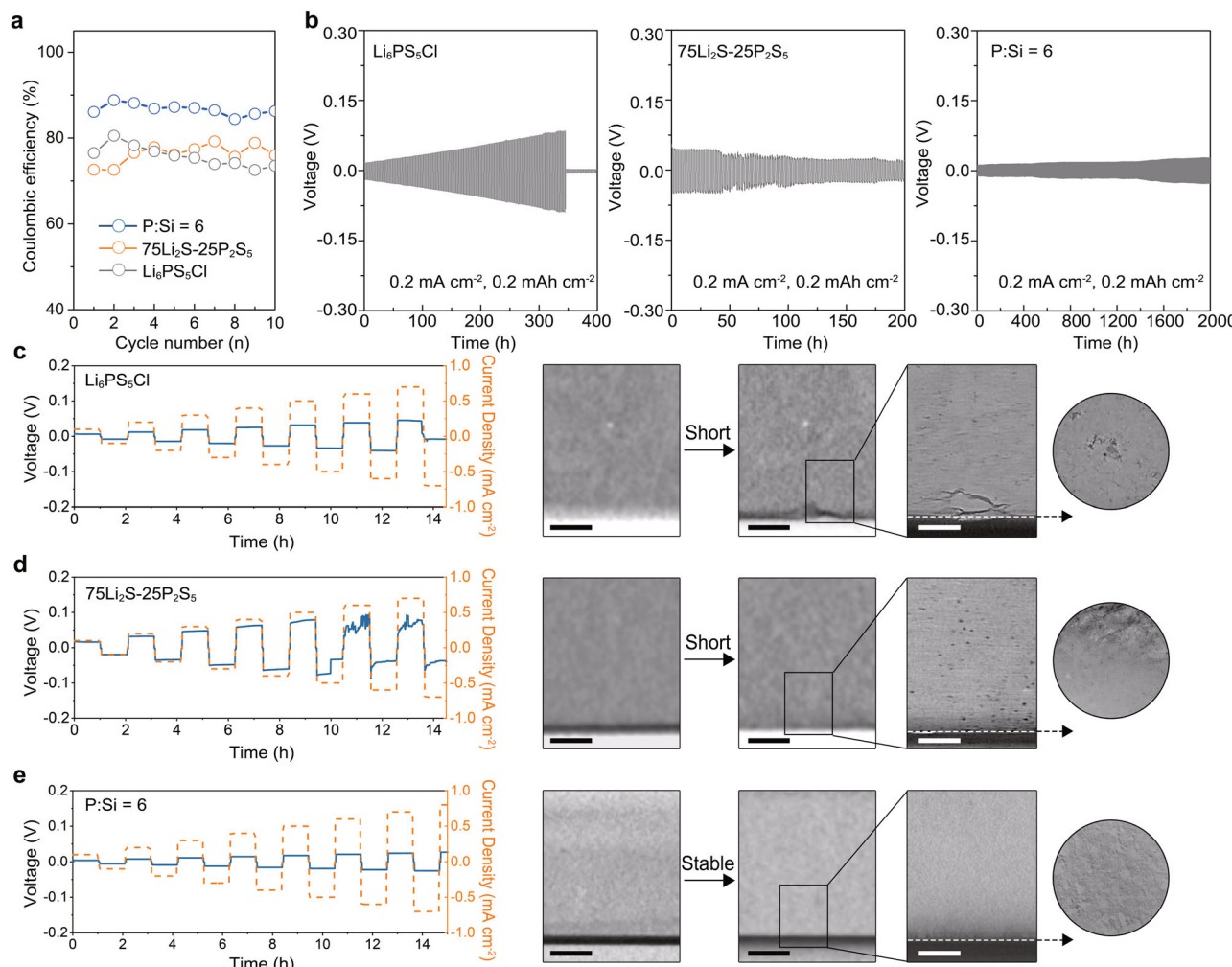

**Fig. 5 | The comparison of Li compatibility between the P:Si = 6 electrolyte and state-of-art sulfides. a** The CE profiles of three electrolytes. **b** Long-time galvanostatic cycling profiles of three electrolytes. **c–e** The profiles of the CCD test on $Li_6PS_5Cl$-based Li symmetric cell and the in-situ X-ray CT morphology of the $Li_6PS_5Cl$/Li interface before and after the CCD test (**c**). The profiles of the CCD test on $75Li_2S$-$25P_2S_5$-based Li symmetric cell and the in-situ X-ray CT morphology of the $75Li_2S$-$25P_2S_5$/Li interface before and after the CCD test (**d**). The profiles of the CCD test on P:Si = 6-based Li symmetric cell and the in-situ X-ray CT morphology of the P:Si = 6 /Li interface before and after the CCD test (**e**). The black scale bar is 250 µm and the white scale bar is 100 µm.

electronic conductivity values of $75Li_2S$-$25P_2S_5$, $Li_6PS_5Cl$ and the P:Si = 6 electrolyte were measured. Among the three electrolytes, the P:Si = 6 electrolyte displays the lowest electronic conductivity of $1.50 \times 10^{-9}$ S cm$^{-1}$, while $75Li_2S$-$25P_2S_5$ and $Li_6PS_5Cl$ exhibit electronic conductivity values of $2.60 \times 10^{-9}$ S cm$^{-1}$ and $4.12 \times 10^{-9}$ S cm$^{-1}$, respectively (Fig. S23). Plus, the micro-morphology of $Li_6PS_5Cl$, $75Li_2S$-$25P_2S_5$ and the P:Si = 6 electrolyte is represented in Fig. S24. In detail, the polycrystalline $Li_6PS_5Cl$ presents obvious edges and voids created by non-dense crystal-crystal stacking while $75Li_2S$-$25P_2S_5$ glass with long-range disorder structure displays a denser stacking without noticeable edges and voids. Specifically, a cobblestone structure with a flat and smooth grain surface is observed in the I-rich P:Si = 6 glass with a soft nature, indicating ignorable interfacial energy between grains. Furtherly, the cold-pressed $Li_6PS_5Cl$ pellet exhibits a relatively low relative density of 84.8% attributed to extensive grain boundaries. In comparison, the relative densities of the glass electrolyte $75Li_2S$-$25P_2S_5$ and P:Si = 6 are 91.0% and 93.6%, respectively, both surpassing that of $Li_6PS_5Cl$ (Table S6). Notably, the P:Si = 6 electrolyte demonstrates a higher relative density than $75Li_2S$-$25P_2S_5$. This is primarily attributed to the halogen dopant LiI, which effectively reduces the elastic modulus of sulfide electrolytes, thereby enhancing the relative density of cold-pressed electrolyte pellets[10].

Then, Li stripping/plating Coulombic efficiency (CE) values in three SSEs were evaluated by using the Li ‖ current collector half cell. Due to the potential reactivity of copper with sulfide electrolytes, stainless steel (SS) is utilized as the current collector material instead of copper. To ensure sufficient contact between the SS and electrolytes, a modest current density of 0.1 mA cm$^{-2}$ and a cut-off capacity of 0.1 mAh cm$^{-2}$ were applied for Li stripping/plating on the SS. In the first cycle, the highest initial CE value of 86.08% is achieved in the P:Si = 6 electrolyte while $Li_6PS_5Cl$ and $75Li_2S$-$25P_2S_5$ demonstrate lower initial CE values of 76.50% and 72.62%, respectively (Fig. 5a). Furthermore, as the cycle progresses, the P:Si = 6 electrolyte demonstrates the highest average CE value of 86.68%, surpassing that of $Li_6PS_5Cl$ (75.75%) and $75Li_2S$-$25P_2S_5$ (76.28%). The corresponding voltage-capacity profiles of Li ‖ SS cells are provided in Fig.S25.

From the above results, it is evident that $75Li_2S$-$25P_2S_5$ displays a lower initial CE when compared to $Li_6PS_5Cl$. This discrepancy may be attributed to several potential factors. Firstly, the $P_2S_6^{4-}$ cluster in $75Li_2S$-$25P_2S_5$ possessing a lower LUMO value than the $PS_4^{3-}$ clusters is more susceptible to reduction by lithium metal (Fig. S26). Besides, the lower P content in $Li_6PS_5Cl$ leads to a reduced presence of Li-P species with certain electronic conductivity in the SEI. However, the average CE value of $Li_6PS_5Cl$ was discovered to be lower than that of

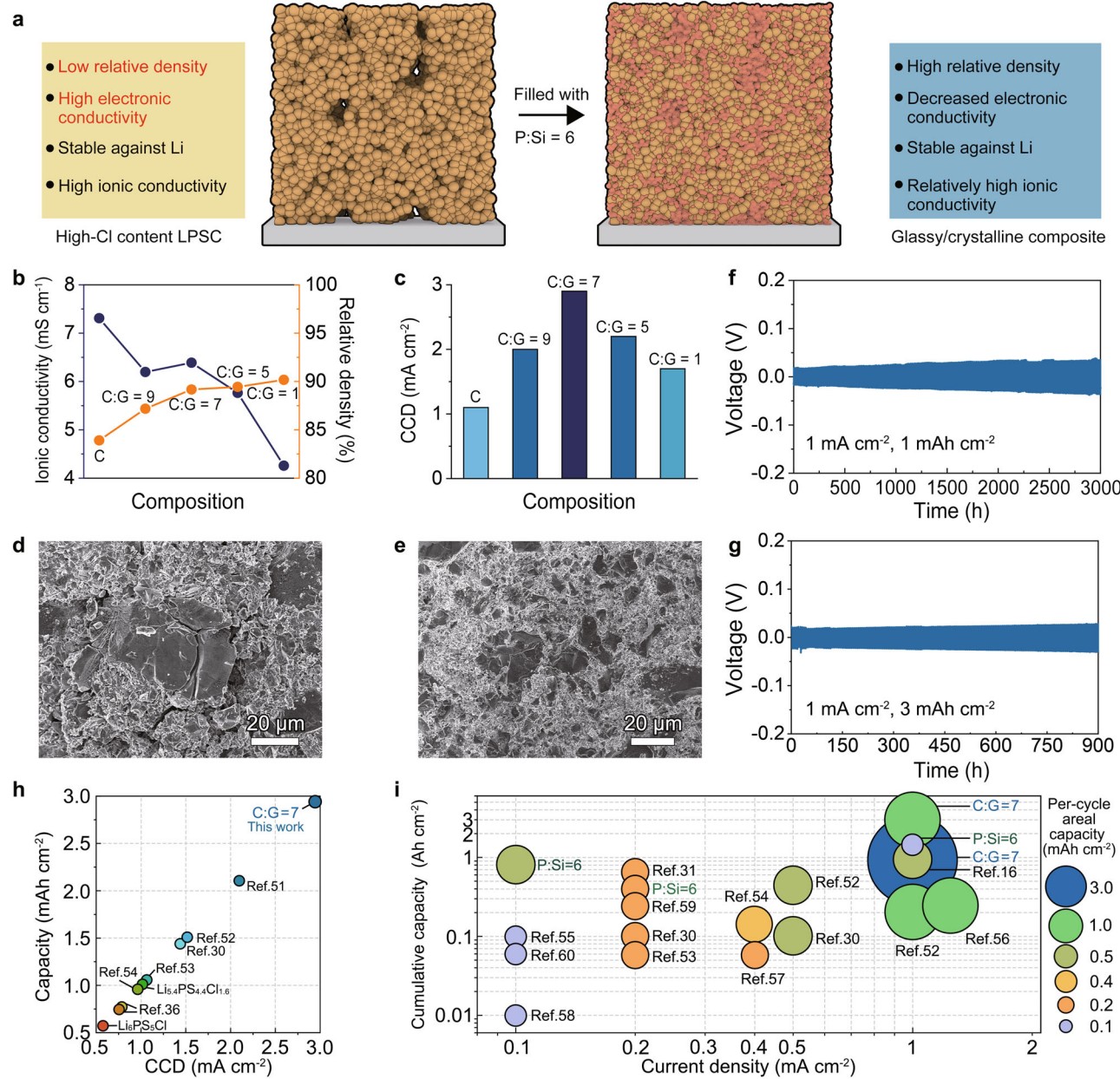

**Fig. 6 | The design of glassy/crystalline composite electrolytes. a** The schematic diagram of the composite electrolyte design. **b** The summary of the relative density and ionic conductivity of the composite electrolyte. **c** The summary of CCD of the composite electrolyte. **d** The surface morphology of the cold-pressed $Li_{5.4}PS_{4.4}Cl_{1.6}$ electrolyte. **e** The surface morphology of the cold-pressed C:G = 7 electrolyte. **f** Galvanostatic Li plating/stripping profiles of the Li symmetric cell employing the C:G = 7 electrolyte at a current density of 1 mA cm$^{-2}$ and a cut-off capacity of 1 mAh cm$^{-2}$. **g** Galvanostatic Li plating/stripping profiles of the Li symmetric cell employing the C:G = 7 electrolyte at a current density of 1 mA cm$^{-2}$ and a cut-off capacity of 3 mAh cm$^{-2}$. **h** The summary of CCD values for reported sulfide electrolytes coupled with bare-Li metal at 25 °C. See Table S8 for detailed information of each reference in the figure. **i** The summary of cycling performance of reported bare-Li symmetric cells using sulfide-based electrolytes as the interlayer at 25 °C. See Table S9 for detailed information of each reference in the figure.

$75Li_2S$-$25P_2S_5$. This may stem from the continuous infiltration of lithium into the non-dense $Li_6PS_5Cl$ pellet with cracks during electrochemical cycling, leading to increased electrolyte reduction. Regarding the P:Si = 6 electrolyte, its possession of the highest initial CE and average CE values among the three electrolytes may be attributed to two factors. Firstly, the P:Si = 6 electrolyte exhibits the highest relative density among the three electrolytes, which suppresses the Li intrusion into the electrolyte, thereby reducing the occurrence of new side reactions during cycling. Secondly, the X-ray photoelectron spectroscopy (XPS) analysis on Li/P:Si = 6 electrolyte interface revealed that, despite the presence of Si$^{4+}$, only P-cations in the P:Si = 6 electrolyte are reduced to form Li-P species, leading to the generation of $Li_2S$ and LiI

(Fig. S27). This aligns well with the previous AIMD simulations and spectroscopic characterizations of the $Li_7SiPS_8$/Li interface[44]. Against this backdrop, one key distinction between the P:Si = 6 glass and other two electrolytes lies primarily in its highest halide (LiI) content and lowest P content. Upon reaction with metallic lithium, the P:Si = 6 electrolyte forms the SEI characterized by a significant amount of LiI and minimal Li-P species. Compared to Li-P species with a bandgap of 0.70–1.76 eV, LiI possesses a broader bandgap of 4.37 eV (Fig. S28), which may effectively reduce the overall electronic conductivity of the interphase, thereby suppressing further side reactions.

Subsequently, long-time galvanostatic discharging/charging tests were performed on Li symmetric cells using different electrolytes at a

current density of 0.2 mA cm$^{-2}$ and a cut-off capacity of 0.2 mAh cm$^{-2}$. The Li symmetric cells using the P:Si = 6 electrolyte demonstrates stable cycling for 2000 h, indicating that the high relative density, adequate ionic conductivity, and high interfacial stability of the P:Si = 6 electrolyte can effectively regulate Li metal intrusion during cycling. In sharp contrast, short or micro-short circuits occur within 400 h for Li symmetric cells using Li$_6$PS$_5$Cl and 75Li$_2$S-25P$_2$S$_5$ electrolytes (Fig. 5b).

An in-situ X-ray computerized tomography (X-ray CT) combined with ex-situ scanning electron microscopy (SEM) was then performed to couple the electrochemical behaviors with the morphological evolution. Initially, mold cells were assembled and clamped using a standard holder (Fig. S29) before a preliminary X-ray scan was performed on the pristine Li/electrolytes interface with an 11 μm voxel resolution. Galvanostatic discharging/charging tests with increased current density were subsequently performed on Li symmetric cells using different electrolytes. After a total cycling time of 15 h, both soft and hard breakdowns are observed in the Li symmetric cells using 75Li$_2$S-25P$_2$S$_5$ and Li$_6$PS$_5$Cl electrolytes[45], respectively, with corresponding CCD values of 0.6 mA cm$^{-2}$ and 0.7 mA cm$^{-2}$. In contrast, the Li symmetric cell using the P:Si = 6 electrolyte exhibits stable functionality throughout the test without sudden voltage drops. A second X-ray scan was then performed on the cycled Li/electrolyte interface, with a voxel resolution similar to the first scan. Comparison of the first and second scans reveals that the crystalline Li$_6$PS$_5$Cl interlayer exhibits cracks with a diameter of 5–10 μm after the cell was short, while the structures of 75Li$_2$S-25P$_2$S$_5$ and the P:Si = 6 electrolyte remain primarily intact under this low resolution. A third scan, using a voxel resolution of 1.2 μm, was then conducted on the circled area to provide more detailed information. The high-resolution X-ray CT results of the cycled interface indicate that the Li$_6$PS$_5$Cl interlayer is significantly deteriorated and the surface of 75Li$_2$S-25P$_2$S$_5$ is corroded by Li metal, accompanied by the generation of voids and pits, while an almost intact Li/electrolyte interface is observed in the symmetric cell employing the P:Si = 6 electrolyte (Fig. 5c–e). Given the invisibility of Li intrusions in X-ray CT tests, the mold cells were disassembled, and SEM was employed to investigate the occurrence of Li intrusions across various electrolytes. As shown in Fig. S30, violent Li intrusions leading to the destruction of the pellet structure are observed at the Li/Li$_6$PS$_5$Cl interface, which is consistent with prior research[15]. Conversely, for 75Li$_2$S-25P$_2$S$_5$ with high relative density but low ionic conductivity, while no notable deformation of the pellet structure is evident, small Li intrusions are observed at both the interface and within the pellet interior, potentially contributing to the phenomenon of soft breakdown. In contrast, the glassy P:Si = 6 layer with high relative density and adequate ionic conductivity remains free of Li intrusions after cycling.

## Glassy/crystalline composite design

The suppression of Li metal intrusions realized by the P:Si = 6 electrolyte with adequate ionic conductivity, low electronic conductivity, high Li/electrolyte interfacial stability and high relative density inspires us to combine the pros of the P:Si = 6 electrolyte and high-Cl content argyrodite Li$_{5.4}$PS$_{4.4}$Cl$_{1.6}$ with high ionic conductivity[46,47], thereby producing a composite electrolyte reconciling the high ionic conductivity, low electronic conductivity, high Li/electrolyte interfacial stability and high relative density to furtherly suppress the Li intrusions. Our design, which involves the incorporation of the P:Si = 6 electrolyte into the high-Cl content argyrodite Li$_{5.4}$PS$_{4.4}$Cl$_{1.6}$, offers several advantages. Firstly, this combination benefits from superior chemical compatibility between sulfide electrolytes. Prior studies have shown that sulfides may react with polymers and halides[48,49], but high-Cl content argyrodite Li$_{5.4}$PS$_{4.4}$Cl$_{1.6}$, being a sulfide, is intrinsically stable against the P:Si = 6 electrolyte. Secondly, this design enables the maintenance of high ionic conductivity. The controlled addition of the P:Si = 6 electrolyte with adequate ionic conductivity will not significantly impede

Li$^+$ transport, thereby allowing the high Li$^+$ diffusion characteristic of high-Cl content argyrodite Li$_{5.4}$PS$_{4.4}$Cl$_{1.6}$ to be preserved. Besides, the incorporation of the P:Si = 6 electrolyte provides enhanced capability for suppressing Li metal intrusions. The P:Si = 6 electrolyte effectively fills cracks, flaws, and gaps between grain boundaries of the argyrodite, leading to increased relative density[50]. Moreover, previous research shows that high-Cl content argyrodite exhibits good stability against Li metal[30]. This study also demonstrates the superior stability of the P:Si = 6 electrolyte against Li metal. Consequently, the composite electrolyte maintains the stability against Li metal to suppress side reactions. Finally, the incorporation of P:Si = 6 glass with low electronic conductivity may reduce the overall electronic conductivity (Fig. 6a).

Then, a series of composite electrolytes were prepared by ball-milling Li$_{5.4}$PS$_{4.4}$Cl$_{1.6}$ and P:Si = 6 electrolytes at a low speed of 100 rpm. The investigated systems are thereby denoted as C (Li$_{5.4}$PS$_{4.4}$Cl$_{1.6}$), C:G = 9 (10 wt% of P:Si = 6 electrolytes), C:G = 7 (12.5 wt% of P:Si = 6 electrolytes), C:G = 5 (16.7 wt% of P:Si = 6 electrolytes), C:G = 1 (50 wt% of P:Si = 6 electrolytes). XRD phase analysis of electrolytes before and after mixing reveals that the structure of high-Cl content Li$_{5.4}$PS$_{4.4}$Cl$_{1.6}$ remains unaltered (Fig. S31), and no new species are generated, indicating no reaction between the P:Si = 6 electrolyte and high-Cl content Li$_{5.4}$PS$_{4.4}$Cl$_{1.6}$.

The EIS results demonstrate (Fig. S32) that the initial high ionic conductivity of the Li$_{5.4}$PS$_{4.4}$Cl$_{1.6}$ electrolyte (7.31 mS cm$^{-1}$) is maintained at a level of ~6 mS cm$^{-1}$ when the weight ratio of the P:Si = 6 electrolyte is below 16.7%, as depicted in Fig. 6b. However, excessive incorporation of the P:Si = 6 electrolyte leads to a notable decrease in ionic conductivity to 4.26 mS cm$^{-1}$ in the C:G = 1 electrolyte. Additionally, when the weight ratio of P:Si = 6 electrolyte exceeds 12.5%, the relative density of the electrolyte pellet increases from 83.9% to over 89% (Fig. 6b). The detailed derivation of relative density of different electrolytes is presented in Table S7. This phenomenon is confirmed by the results of SEM, which shows obvious cracks and flaws in local regions of the Li$_{5.4}$PS$_{4.4}$Cl$_{1.6}$ electrolyte (Fig. 6d and Fig. S33a), while the pellet of C:G = 7 electrolyte remains crack-free and flat (Fig. 6e and Fig. S33b). Electronic conductivity measurements were then performed, and the corresponding values for C, C:G = 7, and C:G = 1 electrolytes are determined to be 4.22 × 10$^{-9}$ S cm$^{-1}$, 2.54 × 10$^{-9}$ S cm$^{-1}$, and 2.10 × 10$^{-9}$ S cm$^{-1}$, respectively (Fig. S34). This result indicates that the incorporation of the P:Si = 6 electrolyte with low electronic conductivity reduces the overall electron transport in the composite electrolyte.

Subsequently, Li symmetric cells were assembled using prepared composite electrolytes as the interlayer to evaluate their capability of suppressing Li intrusions. The CCD test with a current step size of 0.1 mA cm$^{-2}$ was firstly performed (Fig. S35). Figure 6c illustrates that the Li$_{5.4}$PS$_{4.4}$Cl$_{1.6}$ electrolyte exhibits a CCD value of 1.1 mA cm$^{-2}$. Incorporating the P:Si = 6 glassy electrolyte into the Li$_{5.4}$PS$_{4.4}$Cl$_{1.6}$ electrolyte significantly enhances the CCD values of the composite, achieving a peak of 2.9 mA cm$^{-2}$ at the composition of C:G = 7 (Fig. 6c). Moreover, to demonstrate the superiority of using I-rich P:Si = 6 electrolyte as the filler, CCD test was performed on the symmetric cell using Li$_{5.4}$PS$_{4.4}$Cl$_{1.6}$/75Li$_2$S-25P$_2$S$_5$ composite electrolyte, with a weight ratio of 75Li$_2$S-25P$_2$S$_5$ = 12.5%, as the interlayer. The addition of 75Li$_2$S-25P$_2$S$_5$ is found to increase the CCD value to only 1.4 mA cm$^{-2}$ (Fig. S36). Long-time galvanostatic discharging/charging tests were carried out on the Li symmetric cells with the C:G = 7 electrolyte. The symmetric cell with the C:G = 7 electrolyte as the interlayer demonstrates stable cycling for over 3000 h at a current density of 1 mA cm$^{-2}$ and a cut-off capacity of 1 mAh cm$^{-2}$ (Fig. 6f). In sharp contrast, the symmetric cell using Li$_{5.4}$PS$_{4.4}$Cl$_{1.6}$ electrolyte as the interlayer can only function for 6 h (Fig. S37). Following the evaluation protocol proposed by Wang et al.[45], the Li symmetric cell using the C:G = 7 electrolyte was tested at an areal capacity of 3 mAh cm$^{-2}$ and a current density of 1 mA cm$^{-2}$. Notably, stable cycling for over 900 h is observed,

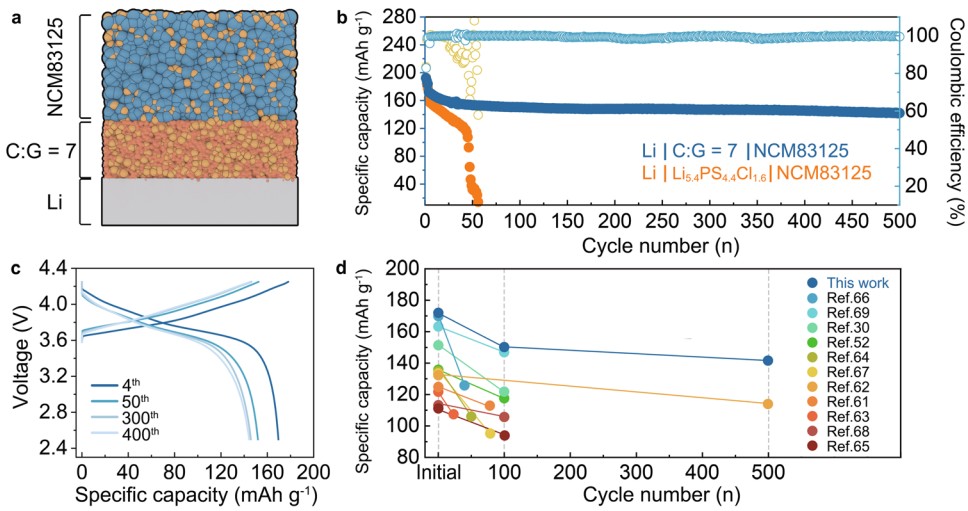

**Fig. 7 | The performance of ASSLMBs. a** The configuration of the Li | C:G = 7 | NCM83125 full cell. **b** Cycling performances of Li | Li$_{5.4}$PS$_{4.4}$Cl$_{1.6}$ | NCM83125 and Li | C:G = 7 | NCM83125 full cells at 0.25 mA cm$^{-2}$. A current density of 0.125 mA cm$^{-2}$ was applied to activate the full cell for the first 3 cycles. **c** The voltage profiles of Li | C:G = 7 | NCM83125 full cell at different cycles. **d** The summary of cycling performance of reported high-voltage ASSLMBs. See Table S10 for detailed reference of each point.

indicating the practical application potential of glassy/crystalline composite electrolytes in high energy-density ASSLMBs (Fig. 6g). It is noteworthy that the CCD value and cycling performances of the C:G = 7 electrolyte in Li symmetric cells stand out prominently among sulfide electrolytes, as Fig. 6h, i illustrates[16,30,31,36,51–60].

To further elucidate the electrochemical performance of the C:G = 7 electrolyte, ASSLMBs were constructed by pairing a high-voltage NCM83125 cathode with a 100 μm Li foil (Fig. 7a). The electrochemical performances of the ASSLMBs were evaluated by galvanostatic discharging/charging tests using Li$_{5.4}$PS$_{4.4}$Cl$_{1.6}$ or C:G = 7 electrolytes as the interlayer. The ASSLMBs using Li$_{5.4}$PS$_{4.4}$Cl$_{1.6}$ and the C:G = 7 electrolyte exhibit comparable initial discharging capacity values of 192.22 mAh g$^{-1}$ and 192.65 mAh g$^{-1}$, respectively, with corresponding CE of 82.84% and 83.53% when activated at 0.1 C (0.125 mA cm$^{-2}$). After 3 cycles of activation, the cells were then cycled at 0.2 C (0.25 mA cm$^{-2}$). The discharging capacity of the Li | Li$_{5.4}$PS$_{4.4}$Cl$_{1.6}$ | NCM83125 full cell shows a continual decrease, which is attributed to increased battery resistance caused by side reactions between intruded Li and Li$_{5.4}$PS$_{4.4}$Cl$_{1.6}$. A short circuit occurs around the 30$^{th}$ cycle, as indicated by fluctuations in the charging profiles (Fig. S38) and a significant decrease in CE. In contrast, the full cell using the C:G = 7 electrolyte demonstrates stable discharging/charging behaviors during cycling (Fig. 7b, c). Furthermore, when cycled at 0.25 mA cm$^{-2}$, the Li | C:G = 7 | NCM83125 full cell delivers an initial discharging capacity of 172.12 mAh g$^{-1}$, which is higher than that of the Li | Li$_{5.4}$PS$_{4.4}$Cl$_{1.6}$ | NCM83125 full cell (163.23 mAh g$^{-1}$). At the 500$^{th}$ cycle, the full cell using the C:G = 7 electrolyte maintains a discharging capacity of 141.98 mAh g$^{-1}$ with a capacity retention of 82.4% (Fig. 7b), which is a highly competitive result compared to reported sulfide-based ASSLMBs (Fig. 7d)[30,52,61–69].

In summary, by deciphering the formation chemistry of the glass network, we propose V$_{int}$ as a vital factor in determining the halogen dopant solubility in sulfide glass electrolytes. Based on the insights of theoretical calculations on the glass forming process, we release the V$_{int}$ in the state-of-the-art 75Li$_2$S-P$_2$S$_5$ glass by constructing a monomer-rich glass network to dissolve more LiI dopants. The resulting glass with a high ionic conductivity of $2.21 \times 10^{-3}$ S cm$^{-1}$, a high relative density of 93.6%, and a low electronic conductivity of $1.50 \times 10^{-9}$ S cm$^{-1}$ functions well in regulating the Li metal suppression. Additionally, this glass surpasses the current leading Li$_6$PS$_5$Cl and 75Li$_2$S−25P$_2$S$_5$ in terms of ionic conductivity, Li intrusions suppressing capability and

interfacial stability with Li. Moreover, a glassy/crystalline composite design is proposed to combine the pros of high-Cl content argyrodite and our optimal glass electrolytes. The composite electrolyte notably suppresses Li metal intrusions. Our composite design achieves stable cycling at practical current densities and areal capacities. Using the composite electrolytes as SSEs, all-solid-state Li || NCM83125 batteries exhibit a retention of 82.4% of initial discharging capacity after 500 cycles at a current density of 0.25 mA cm$^{-2}$. This study provides insights into the dissolution chemistry of halogen dopants in glass and a practical roadmap for developing innovative electrolyte designs. Furthermore, the insights from this study could apply beyond battery technology, offering valuable strategies for the design and optimization of functional materials with improved properties via tailored dopant incorporation.

## Methods

### Synthesis

The starting materials of 0.6((75 + 0.5x) Li$_2$S-(25-0.5x) P$_2$S$_5$−x SiS$_2$)−40 LiI series electrolytes were Li$_2$S (99.98%, Sigma−Aldrich), P$_2$S$_5$ (99%, Macklin), SiS$_2$ (99.99%, Macklin) and LiI (99.95%, anhydrous, Alfa-aesar). The reagents were weighed in the stoichiometric ratio and were mechanically milled at 550 rpm for 20 h to yield the final products. Besides, stoichiometric Li$_2$S (99.98%, Sigma−Aldrich), P$_2$S$_5$ (99%, Macklin) were mechanically milled at 500 rpm for 10 h to prepare 75Li$_2$S−25P$_2$S$_5$ glassy electrolyte. As for the synthesis of Li$_6$PS$_5$Cl electrolyte, stoichiometric Li$_2$S (99.98%, Sigma−Aldrich), P$_2$S$_5$ (99%, Macklin) and LiCl (99.9%, Aladdin) were mechanically milled at 550 rpm for 20 h. The as-milled powder obtained from the ball-milling process was further annealed at 550 °C for 5 h to yield the final crystalline product. As for the synthesis of Li$_{5.4}$PS$_{4.4}$Cl$_{1.6}$ electrolyte, stoichiometric Li$_2$S (99.98%, Sigma−Aldrich), P$_2$S$_5$ (99%, Macklin) and LiCl (99.9%, Aladdin) were mechanically milled at 550 rpm for 20 h. The as-milled precursors obtained from the ball-milling process was further annealed at 450 °C for 5 h to yield the final crystalline product. All processes were conducted in an argon-filled glove box with O$_2$ and H$_2$O < 0.1 ppm.

### Conductivity measurements

Ionic conductivity values were determined by the a.c. impedance method. Firstly, 0.14−0.16 g solid electrolyte powders were pressed into pellets in model cell with a diameter of 10 mm under 360 MPa for

5 min, with two pieces of carbon-coated Al foil (C@Al) on each side to ensure good contact. The detailed thickness of each SSE pellet is summarized in Table S2 and Table S6. The resistance of model cell was measured in the frequency range of 1 MHz to 1 Hz with an amplitude of 30 mV. The measurements were carried out at temperatures between 298.15 K to 338.15 K. Electronic conductivity values were measured by applying 0.1 V DC voltage on C@Al | electrolyte | C@Al block cells for 3600 s.

### Preparation of the glass/Li$_{5.4}$PS$_{4.4}$Cl$_{1.6}$ composite electrolyte

The starting materials of glass/Li$_{5.4}$PS$_{4.4}$Cl$_{1.6}$ composite electrolytes were P:Si = 6 or 75Li$_2$S-P$_2$S$_5$ glassy electrolytes and Li$_{5.4}$PS$_{4.4}$Cl$_{1.6}$ crystalline electrolytes. The precursors were weighed in the appropriate ratio and were mechanically milled at 110 rpm for 1 h to produce a homogenous composite electrolyte.

### Derivation of the relative density

The relative density of the electrolytes can be derived by equation (3):

$$\text{Relative Density} = \frac{\text{Real Density}}{\text{Theoretical Density}} \qquad (3)$$

The theoretical density of a crystalline electrolyte can be derived from the ratio of its atomic mass to its lattice volume. As for the glassy electrolytes, the glass powders were hot-pressed at a temperature around T$_g$ overnight, and a highly densified glass pellet was obtained. Based on the previous study[10,11], the highly densified pellet commonly exhibited a relative density of ~100%. Thereby, the density of the highly densified pellet was set to be the theoretical density of the glass electrolyte.

As for the glass/Li$_{5.4}$PS$_{4.4}$Cl$_{1.6}$ composite electrolyte, the theoretical density $\rho$ can be estimated by the Eq. (4):

$$\rho = \frac{(1+x)\rho_c\rho_g}{\rho_c + x\rho_g} \qquad (4)$$

where $\rho_c$ represents the theoretical density of Li$_{5.4}$PS$_{4.4}$Cl$_{1.6}$, $\rho_g$ represents the theoretical density of the glass, $x$ represents the weight ratio between Li$_{5.4}$PS$_{4.4}$Cl$_{1.6}$ and glass electrolytes.

### Preparation of Li symmetric cells, Li || SS half cells and Li || NCM83125 full cells

All the electrolyte samples in this study were cold-pressed for electrochemical tests. The 100 μm Li foil used in this study were purchased from China Energy Lithium Co. Ltd. The Li metal electrode applied to assemble the cell was prepared via cutting the Li foil into disks with a diameter of 10 mm. The 100 μm stainless-steel foils were cut into disks with a diameter of 10 mm to assemble the Li || SS half cells. The NCM83125 materials used in this study were purchased from Ningbo Ronbay New Energy Technology Co. Ltd. To prepare Li symmetric cells, electrolyte powders were pressed under 300 MPa for 5 min in the Swagelok mold cell with a diameter of 10 mm. Then two disks of lithium electrode were pressed onto both sides of the electrolyte layer under 30 MPa for 5 min. For the preparation of Li || SS half cells, electrolyte powders were pressed under 300 MPa for 5 min. Then, the SS disk was attached on the cathode side and further pressed under 300 MPa for 2 min. Next, the lithium disk was sequentially attached on the anode side and further pressed under 50 MPa for 1 min. In the case of Li || NCM83125 full cells, the NCM83125 composite cathode was prepared by ball-milling NCM83125 and Li$_{5.4}$PS$_{4.4}$Cl$_{1.6}$ (mass ratio = 7:3) at a speed of 110 rpm for 1 h. Full cells were then assembled in a PEEK model cell with a diameter of 10 mm. As for the Li | C:G = 7 | NCM83125 full cell, 110 mg of C:G = 7 electrolyte powders were pressed under 300 MPa for 2 min. Then 7 mg of composite cathodes (corresponding to active material loading of 6.24 mg cm$^{-2}$) were uniformly spread on one side of

the electrolyte layer and pressed under 360 MPa for 3 min. Finally, a lithium disk was sequentially attached on the anode side and further pressed under 50 MPa for 1 min. As for the Li | Li$_{5.4}$PS$_{4.4}$Cl$_{1.6}$ | NCM83125 full cell, 110 mg of Li$_{5.4}$PS$_{4.4}$Cl$_{1.6}$ electrolytes were applied as the interlayer. The calculation of the specific capacity in this study was based on the mass loading of active materials.

### Electrochemical tests

Critical current density (CCD) test and long-time Li plating/stripping measurement for Li symmetric cells were conducted on LAND-CT3001A battery test systems (Wuhan Rambo Testing Equipment Co., Ltd.). As for the CE measurement, 10 cycles of galvanostatic charging/discharging were applied on the activated Li || SS half cells at a current density of 0.1 mA cm$^{-2}$, with a discharging capacity of 0.1 mAh cm$^{-2}$ and a charging cut-off voltage of 1 V. In the case of Li || NCM83125 full cells, galvanostatic charge–discharge tests were performed on a NEWARE CT-4008 battery test system. the full cells were firstly activated at a rate of 0.1 C (0.125 mA cm$^{-2}$) for 3 cycles and then functioned at a rate of 0.2 C (0.25 mA cm$^{-2}$) in a voltage range of 2.5 V–4.25 V. All of the electrochemical tests were performed in the thermostatic chamber with a temperature of 25 °C.

### Material characterization

To determine the glass-transition temperature (T$_g$) of the P:Si = 6 sample, The differential scanning calorimetry (DSC) measurement was conducted on a TA DSC2500 instrument under N$_2$ flow. Approximately 8 mg of the sample powder was loaded in an aluminum pan and heated from 50 to 300 °C at a rate of 5 °C min$^{-1}$. Powder X-ray diffraction (XRD) was measured with Cu Kα radiation (λ = 1.54178 Å) in a 2 θ range from 10° to 80°. Powders were kept in a zero-background sample holder covered by Kapton film. Raman spectrum measurement was performed by a Renishaw in Viareflex Raman spectrophotometer with an incident laser beam at 532 nm. Solid-state $^{29}$Si and $^{31}$P MAS NMR spectra were recorded on a BRUKER 400 M spectrometer. The powder samples were placed in a pencil-type zirconia rotor of 4.0 mm o.d. The spectra were obtained at a spinning speed of 10 kHz (6.5 μs, 90° pulses) and a recycle delay of 4 s.

### In-situ X-ray CT

The inner structure evolution of the symmetric cell was analyzed with ZEISS Xradia Versa 515. Fresh cylinder battery was clamped with standard holder with ZEISS Xradia Versa instruments. Then initial scan was conducted with 11 um voxel resolution which cover the whole battery. The X-ray tube voltage used is 80 kV and the power is 7 W. A LE2 filter was used to reduce the beam hardening artifact induced by the high-density screw. The exposure time for each projection was 2 s. Image reconstruction was performed by the Xradia Reconstructor software. Then, the sample was taken out of the chamber and galvanostatic charging/discharging were performed. Later, the sample was remounted as the previous status and the intact scan with the same parameter was conducted. After that, the cycled data and initial data were compared in Dragonfly PRO software. By utilizing the scout-and-zoom function, the Li/electrolyte interface was moved to the center of the beam path and conduct 2$^{nd}$ scan with 4× objective lens to achieve 1.2 um voxel resolution. The voltage was 80 kV and exposure time was 18 s. To improve image signal to noisy ratio, ZEISS ART 3.0 Deeprecon Pro based on machine learning module was used.

### Computational methods

The geometric optimization of crystal structures, electronic density calculations, and ab initio molecular dynamics (AIMD) simulations in this work were performed using the Vienna Ab initio Simulation 5.4.4 Package (VASP 5.4.4) within the projector augmented-wave (PAW) approach, and the Perdew−Burke−Ernzerhof (PBE) generalized gradient approximation (GGA) functional was used as the exchange-

correlation functional[70,71]. The crystal structures of β-Li$_3$PS$_4$, Li$_4$PS$_4$I and were created based on the previous reports[72,73]. To mitigate the biased results introduced by a single structure input in the later molecular dynamics calculations, we created 3 structures for both Li$_4$PS$_4$I and Si-substituted Li$_4$PS$_4$I, each with distinct Li atomic occupancy conditions. For each input-structure generation, 20 potential structures were generalized and prescreened by an electrostatic energy criterion using the code implemented in the pymatgen package[74], and the doping model with the lowest electrostatic energy was chosen to be further optimized. The geometry optimizations were then performed using the conjugated gradient method. A cut-off energy of 520 eV and a k-point mesh of $3 \times 3 \times 3$ was used and the convergence threshold was set to be $10^{-5}$ eV in energy and 0.01 eV Å$^{-1}$ in force. The visualization of the electronic density was realized by VESTA.

As for the AIMD simulations, optimized models of Si-substituted Li$_4$PS$_4$I, Li$_4$PS$_4$I, and β-Li$_3$PS$_4$ samples were heated up to 2000 K by velocity scaling over 2 ps and then were rapidly cooled to 600 K to generate the glassy-like structure. The size of the simulation box for Si-substituted Li$_4$PS$_4$I, Li$_4$PS$_4$I is ~1618 Å$^3$. During the heating-cooling process, the breaking and melting of P-S bonds can be observed. After cooling to 600 K, the systems were equilibrated for 4 ps in the NVT ensemble. The molecular dynamic simulations for diffusion were then performed for 40 ps with a time step of 2 fs. A Γ-point-only grid and a lower but sufficient energy cutoff of 280 eV were applied during overall simulating process. Radial distribution functions (RDFs) of various species were calculated using the code implemented in the Vasppy package. We supply the structure data of Si-substituted Li$_4$PS$_4$I and Li$_4$PS$_4$I before and after AIMD simulations in the Supplementary Data 1.

The calculation of the reaction energy between glass formers and glass modifiers, the calculation of the dipole moment, and the calculation of the molecular orbital were realized by Gaussian 16 package. The molecular geometries for the ground states were optimized by density functional theory at the B3LYP/6-311 G+ (d) level. The energy of molecules was evaluated at the B3LYP/6-311 G+ (d) level as well. The dipole moment of molecules was evaluated at the B3LYP/def2-TZVPD level[75].

## Quantitative analysis of the $^{29}$Si and $^{31}$P NMR spectra

The $^{29}$Si and $^{31}$P NMR spectra can be coupled to derive the relative ratios of anion clusters in the glassy electrolyte since the molar ratio between P and Si is known for each electrolyte. And the correlation factor $K$ was then derived to couple the two spectra in Eq. (5) below.

$$K = \frac{(R_P(PS_4) + 2R_P(P_2S_6) + 2R_P(P_2S_7))}{N(R_{Si}(SiS_4) + 2R_{Si}(Si_2S_6))} \quad (5)$$

Where $R_P$ represents the relative ratio of anion clusters in $^{31}$P spectra, $R_{Si}$ represents the relative ratio of anion clusters in $^{29}$Si spectra, $N$ represents the molar ratio between the P and Si.

Once the $K$ was derived, we can then calculate the correlated relative ratios of Si-species ($c - R_{Si-species}$) and P-species ($c - R_{P-species}$) based on Eqs. (6) and (7) below.

$$c - R_{Si-species}$$
$$= \frac{KR_{Si-species}}{R_P(PS_4) + 2R_P(P_2S_6) + 2R_P(P_2S_7) + K(R_{Si}(SiS_4) + 2R_{Si}(Si_2S_6))} \quad (6)$$

$$c - R_{P-species}$$
$$= \frac{R_{P-species}}{R_P(PS_4) + 2R_P(P_2S_6) + 2R_P(P_2S_7) + K(R_{Si}(SiS_4) + 2R_{Si}(Si_2S_6))} \quad (7)$$

## Reporting summary

Further information on research design is available in the Nature Portfolio Reporting Summary linked to this article.

## Data availability

The datasets generated during and/or analyzed during the current study are available from the corresponding author on request.

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

## Acknowledgements

Prof. J.P.T. acknowledges the funding support from the National Natural Science Foundation of China (grant. no. U20A20126, 51971201) and the Key Research and Development Program of Zhejiang Province (2022C01071). Prof. X.L.S. acknowledges the support from the Natural Sciences and Engineering Research Council of Canada (NSERC), the Canada Research Chair Program (CRC), the Canada Foundation for Innovation (CFI), the Ontario Research Fund, and the University of Western Ontario. Prof. C.H.W. acknowledges the funding support from the Eastern Insitute of Technology (EIT), Ningbo.

## Author contributions

Y.Z., C.H.W., X.L.S., H.S., and J.P.T. conceived the project and designed the experiment. H.S. carried out the experiment and performed the theoretical study with the help of Y.L., J.R.L., Y.Z., M.K.W., Y.H., X.L.W and C.H.W., and L.A.J., B.X., N.N.Z. performed in-situ X-ray CT in this work. All authors participated in the discussion of the data. H.S., Y.Z., and C.H.W. wrote the initial manuscript with the input of all authors. X.L.S. and J.P.T. revised the manuscript and directed the work.

## Competing interests

The authors declare no competing interests.
