## [Peer Review File · Nature Communications]

Deciphering the Critical Role of Interstitial Volume in Glassy Sulfide Superionic ConductorsREVIEWER COMMENTS

Reviewer #1 (Remarks to the Author):

The manuscript is described about Si-added sulfide glass. The contents were good, but still need to improve.

Why did the author choose the composite electrolyte, not two layered configuration? Because of the direct contact of Li metal with LiPSCI, the anode/electrolyte interface would not be stable. The authors should check this point by post-mortem analysis as well as LiPSCI/glass interface.

Also, XRD and other characterization should be done in hot-pressed sample since the hot-presses sample is used for the cell.

Here are other comments.

1. At the end of introduction, "3000hours at a current density of 1 mA cm⁻²..... 900 hours at a current density of 1 mA cm⁻²". This sentence is strange. please check it again.
2. In Fig. 3d, MSD is Li or other elements. please mention it.
3. In Fig. S9, What is other peaks in DSC, for example at 200 degreeC.

Reviewer #2 (Remarks to the Author):

The authors present a study of Li₄SiS₄ incorporation into glassy superionic Li₂S-P₂S₅ to promote dissolution of LiI in the matrix. The paper begins with a fundamental simulation-based study discussing the origins of the additive-promoted dissolution and concludes with a detailed experimental study of both the electrolyte and a composite with an argyrodite in symmetric and full cell configuration. The mixture shows improvements in the critical current density without sacrificing ionic conductivity.

The paper has some interesting elements and presents a new and promising material. The attempt to devise and enact a rational design strategy is commendable, and the work appears to show some success in that regard. My background is better suited to the fundamental portion of the paper, about which I have several concerns; I will leave it to the other reviewers to comment on the applied experimental portions that represent the bulk of the remaining work.

1) Few details of the AIMD are provided. Simulating glassy materials can be challenging due to statistical sampling. How large were the simulation boxes? I could not find this information. Also, although the procedure for generating the initial configurations via melt-quench is sound, was only a single configuration used for the ultimate analysis? I am concerned that the results may be biased by the choice of configuration. The authors should show that the diffusion rates and other relevant results do not depend on the specific input configuration (i.e., by testing multiple configurations).

2) Although it seems clear that the addition of Li₄SiS₄ is improving the mobility, I do not fully understand the authors' arguments as to why. It is clear that Li₄SiS₄ is preventing

dimerization of P_xS_y units. As I understand it, the authors assert that this prevents sulfur ions from blocking the interstitial sites, promoting dissolution of LiI, which in turn aids in diffusion. I have several questions regarding this theory, which appears to be a central conclusion:

a. How does the coordination environment of iodine change following inclusion of Li_4SiS_4 in the AIMD simulations? Is it largely the same? The authors focus on the environment of S, but equivalent structural data (e.g., $g(r)$) for I is not included even though it seems central to the argument that Li_4SiS_4 is enhancing coordination of I. How accurate is the schematic picture in Fig. 1b compared to the actual simulation results?

b. Could it also be that the primary consequence of suppression of dimerization by Li_4SiS_4 is to make the PS_x units more mobile, thereby facilitating diffusion (i.e., acting as a network modifier)? In other words, how critical is the function of LiI in the modified glass matrix? If the authors want to claim that the most important ultimate function of Li_4SiS_4 doping is to aid LiI insertion, then AIMD results for Li_4SiS_4 -modified Li_2S - P_2S_5 without LiI might change the structure but should not exhibit a strong enhancement in ionic conductivity. If, on the other hand, Li_4SiS_4 in Li_2S - P_2S_5 without LiI prevents dimerization and also leads to faster ionic conduction, then it seems reasonable to conclude that enhancement of PS_x mobility is more important than the dissolution of LiI. Directly comparing simulation results for $[Li_2S-P_2S_5]+LiI$, $[Li_2S-P_2S_5]+LiI+Li_4SiS_4$, and $[Li_2S-P_2S_5]+Li_4SiS_4$ would significantly strengthen the authors' arguments for the rational design strategy and its proposed physical origin.

c. The concept of interstitial sites is a bit confusing in an amorphous material. Is the site blocking (or lack thereof) in Li_2S somehow reflected in the $g(r)$ in Fig. 2a-b?

d. If accessible interstitial volume of the glass is the critical factor, does this imply that the same benefits for LiI "interstitial" incorporation could be achieved by altering processing (e.g., faster quenching)?

e. As the authors correctly state on p. 4, there have been numerous justifications about the role of halogen dopants in mechanistic improvements of LPS-based materials. Does the current study shed any light on these?

3) Fig. 1f: I do not understand what is being shown here, and there is no scale on the free energy axis. Are these molecular calculations? And is this actually Gibbs free energy? If so, how was the entropy change computed and at what temperature? What is "coupled Li_3PS_4/Li_4SiS_4 " (is this the complex in Fig. S6)? If so, it's hard to see how these calculations of energy will translate meaningfully to the glassy solid, where the Li^+ ions are mobile, probability density of Li^+ is diffuse, and the charges are screened by the matrix. As a result, the concept of local Li-containing molecular structure is difficult to grasp. The energy reduction upon coupling is shown as 60 kJ/mol (quite large), but I'm guessing that these results are skewed by dipole interaction effects, which are highly problematic because they depend on where the Li^+ ions are placed. Indeed, the dipole moments and electron distributions in the SI (Figs. S5 and S6) seem to confirm this (e.g., the huge dipole of 11.27D for Li_3PS_4 is clearly due to the placement of Li in the zero-temperature structure).

4) Along the lines of comment #3, I do not understand the dipole discussion on p. 8 of the text. It is not immediately clear to me how lowering local dipole moments in the glass will necessarily decrease reaction rates for dimerization. Also, if I'm reading Fig. 1e correctly

(see comment #6), the polarization of PS_x units is actually increasing upon Li₄SiS₄ incorporation. =To probe the electrostatic effects, it would be better (and safer) to focus on the intrinsic polarity of the individual bonds (Si-S vs. P-S), as well as the consequence of mixing trivalent PS₄ complexes with tetravalent SiS₄ complexes (which could be crucial for disrupting the redox reaction).

5) Fig. 1c: The caption says "LUMO value of P⁵⁺ and Si⁴⁺"--in which system or complex? What is this energy with respect to? EV units should be eV.

6) Fig. 1d-e: Where are the atoms? Which plane is this along? Is the plot only showing the electron density contribution from the PS_x units? I do not understand what I am supposed to take away from this plot. It might be better to zoom in on a specific interaction and superimpose the atoms to see how the electron density is being distorted.

7) Label the atom colors in all figure captions (also in the SI).

8) Balance Equation 1.

9) Fig. 3 caption: State clearly which results are experimental and which are theoretical. How do the simulation-derived activation energies compare with the experiments?

10) Some of the language and phrasing is awkward, particularly in the early sections of the paper. These could benefit from a closer edit to make them more readable.

Reviewer #3 (Remarks to the Author):

This manuscript introduces the concept of 'interstitial volume' to propose a method for designing the composition of sulfide glassy electrolytes with LiI additive. Both experimental and computational calculations have been well-executed to support this research. The concept of interstitial volumes formed from PS₄³⁻ monomers is interesting and provides valuable insights into how to incorporate LiI into glassy electrolytes. Before the publication of this manuscript, it is necessary to consider that the "interstitial volume" concept explored in this study can be applied more broadly. I believe that this research, which involves synthesizing amorphous solid electrolytes with the addition of LiI and understanding their structure, can be highly valuable. I recommend the publication of this study if the following questions are addressed.

Sulfide solid electrolytes like Li₃PS₄-Li₄MS₄ (M = Sn, Ge, Si, etc) with reasonable Li-ion conductivities have already been reported. Why did the authors choose Si, and can this "interstitial volume" concept be applied to various M⁴⁺ (including Sn, Ge), not only Si? Additionally, the precursor SiS₂ is challenging to synthesize and expensive.

In the existing Li₃PS₄-LiI system, sufficient ion conductivity and stability were already reported. Please explain how the addition of Li₄SiS₄ can offer advantages for battery performance improvement beyond inhibiting the formation of p-p dimers.

On page 20, line 489, it was mentioned that when the C:G ratio is 7, the highest critical current density values are achieved with sufficient pellet density, high ion conductivity, and low electrical conductivity. Even if these conditions are met, a formation of continuous unstable interfaces with lithium metal could significantly degrade performance. Please describe the perspective of performance improvement related to the interface with lithium metal or interfacial products.

When using this glassy sulfide as a solid electrolyte (SE) layer, Li/Li batteries achieve the highest CCD values of 4.0 mA/cm² at a capacity of 0.1 mAh/cm². However, accurately determining the CCD value when evaluating batteries with a capacity of 0.1 mAh/cm² is challenging. (see Joule 6, 1770, 2022). This is because lithium can be formed and disappear within the porous regions of the electrolyte layer. To demonstrate a CCD of 4 mA/cm², a minimum capacity of 1 mAh/cm² should be evaluated.

The copper foil in Figure 5a is unsuitable as a current collector due to the inevitable spontaneous corrosion reaction between copper and Li₆PS₅Cl. Stainless steel (SUS) or nickel (Ni) foil is considered a more appropriate substrate.

Manuscript Title (ID: NCOMMS-23-41433): Deciphering the Critical Role of Interstitial Volume in Glassy Sulfide Superionic Conductors

Dear editor and reviewers:

Thank you for your valuable and constructive comments on our manuscript. We have carefully studied the comments and made corresponding modifications to the paper. Our point-by-point responses are provided below. All revisions in the manuscript and supporting information are highlighted in yellow. This revision process has significantly enhanced the quality of the manuscript, and we hope it now meets the standards of ***Nature Communications***. We appreciate your input.

Thank you once again.

Reviewer #1

The manuscript is described about Si-added sulfide glass. The contents were good, but still need to improve.

Response: Thanks for your valuable comments! We have addressed each point in our responses below, and we believe that your kind suggestions significantly enhance the overall quality of this work.

Comment 1. *Why did the author choose the composite electrolyte, not two layered configuration? Because of the direct contact of Li metal with LiPSCl, the anode/electrolyte interface would not be stable. The authors should check this point by post-mortem analysis as well as LiPSCl/glass interface.*

Response to comment 1:

Thanks for your valuable comments!

Firstly, we would like to address the question of why we chose a composite electrolyte over a two layered electrolyte. Considering a two-layered structure comprising a Glass/high-Cl content LPSC configuration, despite the notable benefits of our glass in relative density and electronic conductivity over high-Cl content LPSC, the CCD level (1.2 mA cm^{-2} with a capacity of 1.2 mAh cm^{-2}) of our optimal glass is only marginally higher than that of high-Cl content LPSC (1.1 mA cm^{-2} with a capacity of 1.1 mAh cm^{-2}). This is primarily due to the lower ionic conductivity of our glass compared to high-Cl content LPSC, resulting in a higher Li nucleation overpotential at the interface.¹ Therefore, it is conceivable that when the applied current density surpasses the CCD of our optimal glass, lithium metal will intrude into the glass, thereafter contacting with the second layer of high-Cl content LPSC electrolyte. In this situation, the high-Cl content LPSC with a CCD smaller than that of our optimal glass, cannot impede the ongoing intrusion of lithium dendrites. Consequently, the design of a two-layered electrolyte structure remains insufficient to address the limitations stemming from the glass electrolyte's relatively low ionic conductivity and the drawbacks of the low relative density, high electronic conductivity in the high-Cl content LPSC. However, the adoption of a composite structure, integrating the high relative density and low electronic conductivity attributes of our optimal glass with the high ionic conductivity attributes of high-Cl content LPSC electrolyte, not only mitigates mechanical failures of the electrolyte but also ensures a reduction in the overpotential

for Li nucleation at the interface. As a result, the composite electrolyte exhibits a record-high CCD value of 2.9 mAh cm^{-2} , which is far higher than that of the initial glass electrolyte and High-Cl content LPSC. For the above reasons, we chose a composite electrolyte rather than a two layered electrolyte.

Secondly, we would like to address your concerns regarding the instability of LPSC with metallic lithium. In fact, we have considered this when designing the composite structure. Based on the previous research in our group and other groups,^{2, 3} it has been observed that low-Cl content $\text{Li}_6\text{PS}_5\text{Cl}$ is truly unstable with metallic lithium, leading to the increase of impedance in Li symmetric cells using low-Cl content $\text{Li}_6\text{PS}_5\text{Cl}$ during cycling. However, as the Cl content increases, the stability of LPSC series electrolytes against Li metal significantly improves, and the impedance of Li symmetric cells using high-Cl content $\text{Li}_6\text{PS}_5\text{Cl}$ remains stable during cycling (Fig. R1a). Therefore, in designing the composite structure, we opted for a high-Cl content $\text{Li}_{5.4}\text{PS}_{4.4}\text{Cl}_{1.6}$ electrolyte. This electrolyte exhibits good stability with metallic lithium. However, its low relative density and high electronic conductivity characteristics still result in a CCD that does not meet practical application requirements. By incorporating our optimal glass as a filler, which possesses high relative density and low electronic conductivity, our composite structure not only exhibits high stability against Li metal but also effectively suppresses mechanical failure of the electrolyte, significantly elevating the CCD value of the electrolyte. Plus, it should be noted that the description in Fig. 6a may not be appropriate. When producing this diagram, our consideration of a stable/unstable interface refers to whether the formed interface can effectively control the deposition of metallic lithium to prevent lithium penetration. It does not imply the stability of the electrolyte towards metallic lithium. Given that both high-Cl content LPSC and the composite electrolyte are both stable against metallic lithium, we have revised the description in Fig. 6a (Fig. R1b).

Finally, your concern regarding the stability of the LPSC/glass interface holds considerable significance. Consequently, we performed XRD phase analysis on the electrolyte before and after mixing. (Fig. R2, denoted as Fig. S30 in the SI.) The results revealed that the structure of high-Cl content LPSC remained unaltered before and after mixing, and no new species were generated during the process. These observations indicate the absence of a reaction between our optimal glass and high-Cl content LPSC. Hence, we maintain the belief that the interface between LPSC and glass remains stable.

Fig.R1 a Voltage-time profiles of Li/Li symmetric cells using LPSCI and LPSCI1.5 electrolytes cycled at a constant current density and cut-off capacity of (left) 0.2 mA cm⁻²/0.2 mAh cm⁻² and (right) 0.5 mA cm⁻²/0.5 mAh cm⁻² at room temperature. This result is from the previous work of our group.² **b** The schematic diagram of the composite electrolyte design.

Fig.R2 XRD spectra of G:C=1:7, P:Si = 6 and High-Cl content LPSC electrolytes.

The corresponding revision in our paper is listed as follows:

-Manuscript, Results, Page 24, Fig. 6a.

Fig. 6 The design of crystalline/glass composite electrolytes. **a** The schematic diagram of the composite electrolyte design. **b** The summary of the relative density and ionic conductivity of the composite electrolyte. **c** The summary of CCD of composite electrolytes. **d** The surface morphology of the cold-pressed $\text{Li}_{5.4}\text{PS}_{4.4}\text{Cl}_{1.6}$ electrolyte. **e** The surface morphology of the C/G = 7 electrolyte. **f** Galvanostatic Li plating/stripping profiles in the Li|| C/G = 7 ||Li symmetric cell at a current density of 1.0 mA cm^{-2} and a cut-off capacity of 1 mAh cm^{-2} . **g** Galvanostatic Li plating/stripping profiles in the Li|| C/G = 7 ||Li symmetric cell at a current density of 1 mA cm^{-2} and a cut-off capacity of 3 mAh cm^{-2} . **h** The summary of CCD values for reported sulfide electrolytes coupled with bare Li metal at room temperature. See Table S8 for detailed reference for each point. **i** The summary of cycling performance of bare-Li symmetric cells using sulfide-based electrolytes as the interlayer at room temperature. See Table S9 for detailed reference for each point.

-Manuscript, Results, Page 20.

“...The investigated systems are thereby denoted as C ($\text{Li}_{5.4}\text{PS}_{4.4}\text{Cl}_{1.6}$), C:G = 9 (10 wt% of P:Si = 6 electrolyte), C:G = 7 (12.5 wt% of P:Si = 6 electrolyte), C:G = 5 (16.7 wt% of P:Si = 6 electrolyte), C:G = 1 (50 wt% of P:Si = 6 electrolyte), separately. Based on the XRD phase analysis of electrolytes before and after mixing, the structure of high-Cl content $\text{Li}_{5.4}\text{PS}_{4.4}\text{Cl}_{1.6}$ remained unaltered before and after mixing (Fig. S30), and no new species were generated during the process. These observations indicate the absence of a reaction between our optimal glass and high-Cl content $\text{Li}_{5.4}\text{PS}_{4.4}\text{Cl}_{1.6}$.”

Comment 2. Also, XRD and other characterization should be done in hot-pressed sample since the hot-presses sample is used for the cell.

Response to comment 2:

Thanks for your comments! Actually, we did not subject the samples prepared by hot pressing to electrochemical tests. All electrolyte materials used in our electrochemical tests were cold-pressed. The purpose of hot-pressing was to approximate the theoretical density of glassy-state materials with the density of hot-pressed glass materials. This methodology has been appropriately described in the **Methods**. However, it is still invaluable to provide XRD analysis for a hot-pressed sample. Therefore, we conducted XRD testing on the hot-pressed sample (Fig. R3, denoted as Fig. S14 in the SI), and the result indicated that the glass nature of the sample remained unchanged before and after hot-pressing.

Fig. R3 XRD spectra of hot-pressing P:Si = 6 electrolytes.

The corresponding revision in our paper is listed as follows:

-Manuscript, Methods

"Preparation of symmetric cells, Li||Cu half cells and Li||NMC83125 full cells

All the electrolyte samples in this study are cold-pressed for electrochemical tests...

-Manuscript, Results, Page 12.

“By hot-pressing the pellet at around the glass-transition temperature overnight, a flat and highly-densified pellet with a relative density of ~100% was obtained (Fig. S13b). Plus, the XRD result of the hot-pressing pellet indicated that the glass nature of the sample remained unchanged after hot-pressing (Fig. S14).”

Comment 3. Here are other comments.

Comment 3.1. At the end of introduction, "3000 hours at a current density of 1 mA cm⁻²..... 900 hours at a current density of 1 mA cm⁻²...". This sentence is strange. please check it again.

Response to comment 3.1:

Thanks for your valuable comments! We have revised this strange sentence.

The corresponding revision in our paper is listed as follows:

-Manuscript, Introduction, Page 5.

“...Additionally, the Li symmetric cell employing the composite electrolyte demonstrates remarkably stable cycling for 3000 hours at a current density of 1 mA cm⁻² and an areal capacity of 1 mAh cm⁻². Even under more stringent conditions, with a current density of 1 mA cm⁻² and an areal capacity of 3 mAh cm⁻², it maintains functionality for a duration of 900 hours.”

Comment 3.2. In Fig. 3d, MSD is Li or other elements. please mention it.

Response to comment 3.2:

Thanks for your valuable comments! The MSD is attributed to the Li atoms, and we are sorry for not mentioning this in the manuscript.

The corresponding revision in our paper is listed as follows:

-Manuscript, Results, Page 13, the caption of Fig. 3d.

“Fig. 3 Phase characterizations and ion transport capability of the synthesized materials. **a** The XRD spectra of synthesized series electrolytes. **b** The ionic conductivity of synthesized series electrolytes. **c** The Li⁺ conduction activation energy of synthesized series electrolytes. **d** The Lithium MSD plot for Li₄PS₄I (orange) and Li_{4.125}Si_{0.125}P_{0.785}S₄I (blue) glass at 600 K from AIMD simulations. **e** The Li-ion

probability densities of $\text{Li}_4\text{PS}_4\text{l}$ (top) and $\text{Li}_{4.125}\text{P}_{0.785}\text{Si}_{0.125}\text{S}_4\text{l}$ (bottom) glass from AIMD simulations.”

-Manuscript, Results, Page 13.

“...after relaxation for 40 ps at 600 K, the Lithium mean square displacement (MSD) value of $\text{Li}_{4.125}\text{Si}_{0.125}\text{P}_{0.785}\text{S}_4\text{l}$ glass is higher than that of $\text{Li}_4\text{PS}_4\text{l}$ glass, indicating a more unrestrained environment for Li^+ motion enabled by Si doping.”

Comment 3.3. In Fig. S9, What is other peaks in DSC, for example at 200 degreeC.

Reply to comment 3.3:

Thanks for your valuable comments! The peak at 200 °C in the DSC profile can be attributed to the crystallization peak (T_c) (Fig R4, denoted as Fig.S12 in the SI). As previously reported, the I-rich Li_3PS_4 -type glass will form $\text{Li}_4\text{PS}_4\text{l}$ -type crystal at around 180-200 °C.⁴

Fig. R4 Differential scanning calorimetry (DSC) curves of the P:Si =6 electrolyte in N_2 flow.

The corresponding revisions in our paper are listed as follows:

-Manuscript, Page 12.

“...Moreover, the obtained P:Si = 6 glass exhibits a glass-transition temperature of 132 °C and a crystallization temperature of 206 °C, as determined by differential scanning calorimetry (DSC) analysis (Fig. S12).”

Reviewer #2

The authors present a study of Li_4SiS_4 incorporation into glassy superionic $\text{Li}_2\text{S-P}_2\text{S}_5$ to promote dissolution of LiI in the matrix. The paper begins with a fundamental simulation-based study discussing the origins of the additive-promoted dissolution and concludes with a detailed experimental study of both the electrolyte and a composite with an argyrodite in symmetric and full cell configuration. The mixture shows improvements in the critical current density without sacrificing ionic conductivity.

The paper has some interesting elements and presents a new and promising material. The attempt to devise and enact a rational design strategy is commendable, and the work appears to show some success in that regard. My background is better suited to the fundamental portion of the paper, about which I have several concerns; I will leave it to the other reviewers to comment on the applied experimental portions that represent the bulk of the remaining work.

Response: Thank you for your valuable feedback, which predominantly focuses on theoretical calculations. Your insights have not only enhanced the precision and scientific rigor of our theoretical exposition in the paper but have also contributed to our own theoretical understanding.

Comment 1. *Few details of the AIMD are provided. Simulating glassy materials can be challenging due to statistical sampling. How large were the simulation boxes? I could not find this information. Also, although the procedure for generating the initial configurations via melt-quench is sound, was only a single configuration used for the ultimate analysis? I am concerned that the results may be biased by the choice of configuration. The authors should show that the diffusion rates and other relevant results do not depend on the specific input configuration (i.e., by testing multiple configurations).*

Response to Comment 1:

Thanks for your valuable comment!

The size of simulation boxes is $\sim 1618 \text{ \AA}^3$.

Following your advice, we have performed the same calculation on two additional structures for both Li_4PS_4 and $\text{Li}_{4.125}\text{Si}_{0.125}\text{P}_{0.785}\text{S}_4$, respectively (Fig. R5, represented as Fig. S9 in the SI), and comparable results across varying structures of the same material were observed. It is discovered that the $g(r)$ profiles of P-P and P-S pairs in different structures without Si doping all reveal dimer formation (Fig. R6a-c, represented as Fig. S10a-c in the SI). In contrast, the $g(r)$ profiles of P-P and P-S pairs in structures with Si doping all indicate the potential of Si doping to inhibit dimer formation (Fig. R6d-f, represented as Fig. S10d-f in the SI). Regarding the Li mean square displacement (MSD), distinct structures exhibit analogous Li^+ diffusion dynamics. Notably, structures with Si doping demonstrate enhanced Li^+ diffusion dynamics compared to those without Si doping (Fig. R7, represented as Fig. S19 in the SI). These results imply that our computational findings do not depend on the specific input structures.

Fig. R5 a The Crystal structure of $\text{Li}_4\text{PS}_4\text{I}$ applied in the manuscript and other two additional structure employed for supplementary validation. **b** The Crystal structure of $\text{Li}_{4.125}\text{Si}_{0.125}\text{P}_{0.785}\text{S}_4\text{I}$ applied in the manuscript and other two additional structure employed for supplementary validation. The light purple tetrahedra represents PS_4^{3-} , the blue tetrahedra represents SiS_4^{4-} , the green balls represent Li atoms, the yellow ball represent S atoms and the deep purple balls represent I atoms.

Fig. R6 **a** The $g(r)$ profile of P-P and P-S pair in $\text{Li}_4\text{PS}_4\text{I}$ glass with the structure applied in the manuscript. **b** The $g(r)$ profile of P-P and P-S pair in $\text{Li}_4\text{PS}_4\text{I}$ glass with the Supplementary structure 1. **c** The $g(r)$ profile of P-P and P-S pair in $\text{Li}_4\text{PS}_4\text{I}$ glass with the Supplementary structure 2. **d** The $g(r)$ profile of P-P and P-S pair in $\text{Li}_{4.125}\text{Si}_{0.125}\text{P}_{0.785}\text{S}_4\text{I}$ glass with the structure applied in the manuscript. **e** The $g(r)$ profile of P-P and P-S pair in $\text{Li}_{4.125}\text{Si}_{0.125}\text{P}_{0.785}\text{S}_4\text{I}$ glass with the Supplementary structure 1. **f** The $g(r)$ profile of P-P and P-S pair in $\text{Li}_{4.125}\text{Si}_{0.125}\text{P}_{0.785}\text{S}_4\text{I}$ glass with the Supplementary structure 2.

Fig. R7 The Lithium MSD plot for three structures of $\text{Li}_4\text{PS}_4\text{I}$ glass at 600 K (top) from AIMD simulations, and the Lithium MSD plot for three structures of $\text{Li}_{4.125}\text{Si}_{0.125}\text{P}_{0.785}\text{S}_4\text{I}$ glass at 600 K (bottom) from AIMD simulations.

The corresponding revision in our paper is listed as follows:

-Manuscript, Methods.

"...The size of the simulation box for Si-substituted $\text{Li}_4\text{PS}_4\text{I}$ and $\text{Li}_4\text{PS}_4\text{I}$ is $\sim 1618 \text{ \AA}^3$."

-Manuscript, Methods.

"...To mitigate the biased results introduced by a single structure input in the later molecular dynamics calculations, we created 3 structures for both $\text{Li}_4\text{PS}_4\text{I}$ and Si-substituted $\text{Li}_4\text{PS}_4\text{I}$, each with distinct Li atomic occupancy conditions. For each input-structure generation, 20 potential structures were generalized and prescreened by an electrostatic energy criterion using the code implemented in the pymatgen package, and the doping model with the lowest electrostatic energy was chosen to be further optimized."

-Manuscript, Results, Page 9-10.

"...Moreover, to clarify that the pair distribution results do not depend on the input structures employed during our modeling, two additional validation structures for both $\text{Li}_4\text{PS}_4\text{I}$ and $\text{Li}_{4.125}\text{Si}_{0.125}\text{P}_{0.785}\text{S}_4\text{I}$ were constructed and applied to the consistent molecular simulations, respectively (Fig. S9). As a result, comparable outcomes across varying structures of the same material were observed (Fig. S10)."

-Manuscript, Results, Page 13.

"...Plus, to avoid the influence of the specific input configuration on the results of diffusion rates, the Li^+ diffusion kinetics in two additional structures of $\text{Li}_4\text{PS}_4\text{I}$ and $\text{Li}_{4.125}\text{Si}_{0.125}\text{P}_{0.785}\text{S}_4\text{I}$ glass were also calculated. As Fig. S19 illustrated, distinct structures of the identical material exhibit analogous Li^+ diffusion kinetics, which again confirms the reliability of this theoretical finding."

Comment 2. *Although it seems clear that the addition of Li_4SiS_4 is improving the mobility, I do not fully understand the authors' arguments as to why. It is clear that Li_4SiS_4 is preventing dimerization of P_xS_y units. As I understand it, the authors assert that this prevents sulfur ions from blocking the interstitial sites, promoting dissolution of LiI , which in turn aids in diffusion. I have several questions regarding this theory, which appears to be a central conclusion:*

Comment 2.1. *How does the coordination environment of iodine change following inclusion of Li_4SiS_4 in the AIMD simulations? Is it largely the same? The authors focus on the environment of S, but equivalent structural data (e.g., $g(r)$) for I is not included even though it seems central to the argument that Li_4SiS_4 is enhancing coordination of I. How accurate is the schematic picture in Fig. 1b compared to the actual simulation results?*

Response to Comment 2.1:

Thanks for your valuable comments!

Before addressing your questions, we would like to clarify that there is a discrepancy of LiI content between the calculation models we used (33.3 mol% LiI) and the electrolyte we experimentally synthesized (40 mol% LiI). The reason is that we had to use certain input crystal structures to generate “glass” structure via melt-quenching simulations, and the structure with the highest LiI content among all the structures in ICSD is $\text{Li}_4\text{PS}_4\text{I}$ (corresponding to 33.3 mol% LiI),⁵ which was selected as the input structure. Despite this, our purpose in conducting this simulation experiment is to illustrate that dimerization can be significantly inhibited by Li_4SiS_4 doping. Therefore, the simulation can still provide valuable insights.

Regarding your question about I coordination, we have investigated the RDF differences of Li-I pairs in two systems. As depicted in Fig. R8a (represented as Fig. S11a in the SI), there is no significant difference in the RDF of Li-I pairs between $\text{Li}_4\text{PS}_4\text{I}$ and $\text{Li}_{4.125}\text{Si}_{0.125}\text{P}_{0.785}\text{S}_4\text{I}$ glass, indicating 33.3 mol% LiI can be dissolved in both systems. This result is anticipated since the LiI concentration of simulated electrolytes is lower than that of our experimental electrolytes.

After considering your valuable feedback, we delved further into this issue. Even though both $\text{Li}_4\text{PS}_4\text{I}$ and $\text{Li}_{4.125}\text{Si}_{0.125}\text{P}_{0.785}\text{S}_4\text{I}$ can fully dissolve LiI , can the promoting effect of reduced dimers on LiI solubility be observed by comparing simulation results? In cases where

the RDF of the Li-I pair cannot elucidate the competitive dissolution relationship between LiI and Li_2S_x , is there any theoretical evidence that can suggest the competition for dissolution between LiI and Li_2S_x ?

In the sulfide glass network, cations of the framework are situated at the center, with S^{2-} ions distributed around them. LiI and Li_2S_x , as substances dissolved in the interstitial sites, should be located near the framework. Therefore, we extracted S from the framework (S from the monomer and the dimer) and investigated the RDF of Framework S-I pairs in both systems, as depicted in Fig. R8b (Represented as Fig. S11b in the SI). It can be observed that a strong peak located at a position of 3 Å appeared in the system without Si doping. We attribute the appearance of this peak to “crowded” interstitial positions. In detail, I⁻ would move closer to the framework to alleviate the Coulombic repulsion in the system due to the generation of S_x^{2-} . **It is foreseeable that as the LiI content continues to increase, the “crowded” interstitial positions will eventually lead to an inability of the interstitial volume to accommodate that much LiI.** In contrast, in the Si-doped system, due to the significant reduction of Li_2S_x , the distance between framework S and I is relatively greater, endowing this system with a stronger LiI accommodating capacity.

Fig. R8 a The $g(r)$ profile of Li-I pair in $\text{Li}_4\text{PS}_4\text{I}$ glass and Si-doped $\text{Li}_{4.125}\text{Si}_{0.125}\text{P}_{0.785}\text{S}_4\text{I}$ glass. **b** The $g(r)$ profile of S-I pair in $\text{Li}_4\text{PS}_4\text{I}$ glass and Si-doped $\text{Li}_{4.125}\text{Si}_{0.125}\text{P}_{0.785}\text{S}_4\text{I}$ glass.

The corresponding revision in our paper is listed in the response to **Comment 2.3** because those two comments are highly correlated.

Comment 2.2. *Could it also be that the primary consequence of suppression of dimerization by Li_4SiS_4 is to make the PS_x units more mobile, thereby facilitating diffusion (i.e., acting as a network modifier)? In other words, how critical is the function of Lil in the modified glass matrix? If the authors want to claim that the most important ultimate function of Li_4SiS_4 doping is to aid Lil insertion, then AIMD results for Li_4SiS_4 -modified $\text{Li}_2\text{S-P}_2\text{S}_5$ without Lil might change the structure but should not exhibit a strong enhancement in ionic conductivity. If, on the other hand, Li_4SiS_4 in $\text{Li}_2\text{S-P}_2\text{S}_5$ without Lil prevents dimerization and also leads to faster ionic conduction, then it seems reasonable to conclude that enhancement of PS_x mobility is more important than the dissolution of Lil. Directly comparing simulation results for $[\text{Li}_2\text{S-P}_2\text{S}_5]+\text{Lil}$, $[\text{Li}_2\text{S-P}_2\text{S}_5]+\text{Lil}+\text{Li}_4\text{SiS}_4$, and $[\text{Li}_2\text{S-P}_2\text{S}_5]+\text{Li}_4\text{SiS}_4$ would significantly strengthen the authors' arguments for the rational design strategy and its proposed physical origin.*

Response to Comment 2.2:

Thanks for your valuable comments!

Before addressing your question, we would like to elucidate the mechanism behind the enhancement of ion conductivity in glassy sulfides through Lil doping. The dissolution of Lil in glassy sulfides leads to an increase in the Li^+ concentration. Therefore, based on previous studies,⁶ in the case where Lil is fully dissolved in the glassy framework of Li_3PS_4 glass, the ion conductivity of Li_3PS_4 -based glass sulfides with a Lil dopant will increase with the increase of Lil concentration. However, when Lil cannot continue to dissolve, a low ion conductivity phase of Lil will appear in the system, leading to a decrease in ion conductivity.

In the response to **Comment 2.1**, we mentioned that Lil can be dissolved in both Li_4PS_4 and $\text{Li}_{4.125}\text{Si}_{0.125}\text{P}_{0.785}\text{S}_4$ glass. **Therefore, the AIMD simulations of Li^+ diffusion kinetics in Fig. 3d-e were conducted to mainly illustrate that, aside from the increase of Lil concentration, Si doping itself may also enhance the ion conductivity of the Li_3PS_4 system.** The effects of Si-doping have been reported in numerous studies involving (1). The introduction of Si can increase the Li^+ concentration in the system, (2) The entropy stabilization effects of SiS_4^{4-} . Our theoretical study on Li_4PS_4 and $\text{Li}_{4.125}\text{Si}_{0.125}\text{P}_{0.785}\text{S}_4$ confirms those effects. As Fig R9 (represented as Fig. 3e in the manuscript) illustrates, the system with Si doping exhibits an overall improvement in Li^+ transport dynamics and presents a promoted Li^+ transport in proximity to SiS_4 tetrahedra positions.

Based on your comments, we learned the significance of illustrating that the sole

introduction of Li_4SiS_4 to alter the structure cannot enhance the ionic conductivity to that high level. Therefore, the Li^+ diffusion kinetics in Li_4PS_4 , $\text{Li}_{4.125}\text{Si}_{0.125}\text{P}_{0.785}\text{S}_4$, and $\text{Li}_{3.125}\text{Si}_{0.125}\text{P}_{0.785}\text{S}_4$ were compared, and the corresponding results were presented in Fig. R10 (denoted as Fig. S18 in the SI). It can be observed that solely Si doping ($\text{Li}_{3.125}\text{Si}_{0.125}\text{P}_{0.785}\text{S}_4$) or Lil doping (Li_4PS_4) is insufficient to elevate lithium transport dynamics to the level exhibited by $\text{Li}_{4.125}\text{Si}_{0.125}\text{P}_{0.785}\text{S}_4$ glass with both Si and Lil doping. Also, previous studies have reported that the maximum ionic conductivity of Li_3PS_4 - Li_4SiS_4 crystalline system with a monomer-dominant structure is $\sim 1.2 \text{ mS cm}^{-1}$, lower than that of our optimal glass electrolyte (2.21 mS cm^{-1}).⁷ **Therefore, we can conclude that the substantial enhancement of ionic conductivity in our optimal glass is not exclusively reliant on structural alterations by Si-doping. Instead, the impact of Lil dissolution should also be taken into consideration.**

The corresponding revision in our paper is listed as follows:

-Manuscript, Results, Page 12-13.

"...On the one hand, the incorporation of Lil into the glassy P:Si = 6 electrolyte, rather than serving as an impurity phase, can increase the Li^+ concentration, which promotes the Li transport kinetics. On the other hand, Si-doping itself may also play a significant role in facilitating Li^+ conduction. As demonstrated in Fig. 3d, after relaxation for 40 ps at 600 K, the Lithium mean square displacement (MSD) value of $\text{Li}_{4.125}\text{Si}_{0.125}\text{P}_{0.785}\text{S}_4$ glass is higher than that of Li_4PS_4 glass, indicating a more unrestrained environment for Li^+ motion enabled by Si doping. Besides, compared with Li_4PS_4 glass, $\text{Li}_{4.125}\text{Si}_{0.125}\text{P}_{0.785}\text{S}_4$ glass not only exhibits an overall improvement in Li^+ transport dynamics but also presents a particularly promoted Li^+ transport in proximity to SiS_4 tetrahedra positions, which may be attributed to the increased Li^+ concentration near SiS_4 tetrahedra and entropy-stabilized effect of Si doping (Fig. 3e). Additionally, the Li^+ diffusion kinetics in Li_4PS_4 , $\text{Li}_{4.125}\text{Si}_{0.125}\text{P}_{0.785}\text{S}_4$, and $\text{Li}_{3.125}\text{Si}_{0.125}\text{P}_{0.785}\text{S}_4$ were compared (Fig. S18). It can be observed that solely Si or Lil doping is insufficient to elevate lithium transport dynamics to the level exhibited by $\text{Li}_{4.125}\text{Si}_{0.125}\text{P}_{0.785}\text{S}_4$ glass with both Si and Lil doping. Therefore, it can be concluded that both the effects of Lil dissolution and SiS_4 incorporation on the Li^+ transport kinetics should be taken accounts to explain the significant enhancement of ionic conductivity in our optimal glass."

Fig. R9 The Li-ion trajectory (blue) in Li₄PS₄I (left) and Li_{4.125}Si_{0.125}P_{0.785}S₄I (right) glass at 600 K from AIMD simulations. The orange tetrahedra represents PS₄³⁻, the red tetrahedra represents SiS₄⁴⁻.

Fig. R10 The Lithium MSD plot for Li₄PS₄I, Li_{4.125}Si_{0.125}P_{0.785}S₄I and Li_{3.125}Si_{0.125}P_{0.785}S₄ at 600 K from AIMD simulations.

Comment 2.3. *The concept of interstitial sites is a bit confusing in an amorphous material. Is the site blocking (or lack thereof) in Li_2S somehow reflected in the $g(r)$ in Fig. 2a-b?*

Response to comment 2.3:

Thanks for your valuable comments!

There have been numerous reports on the interstitial sites in amorphous materials, particularly in studies related to the dissolution of gases in glass materials.⁸ In the context of glassy solid-state electrolytes, it is proposed that LiI can dissolve in the interstitial sites of the glass.⁴ In our study, it is discovered that the generation of P-P dimers and dimers will lead to the existence of residual Li_2S_x , which cannot participate in network formation or further modify the network structure. These residual Li_2S_x and LiI will compete for the interstitial spaces.

Although the RDF of the P-S pair provided in our paper may not offer a reflection on this aspect, we gained insights into how to reasonably explain the site-blocking effect while responding to your **comment 2.1**. Firstly, by comparing the structures presented in Fig. R11a-b (represented as Fig. 2a-b in the manuscript) between the “without Si doping” and “with Si doping” configurations, we observed a significant increase in non-framework S in the “without Si doping” structure. Considering the limited interstitial space, the presence of these non-framework S atoms will compress I atoms and cause them to move toward the framework due to Coulombic repulsion. By extracting the radial distribution between S atoms from the framework and I atoms, we have observed that the structure without Si doping exhibits a tendency for some I and framework S to be closer (Fig. R8b). It is foreseeable that as the LiI content continues to increase, the “crowded” interstitial positions caused by residual Li_2S_x will eventually lead to an inability of the interstitial volume to accommodate that much LiI .

Fig. R11 a The snapshot of equilibrium $\text{Li}_4\text{PS}_4\text{I}$ glass after AIMD simulations and the corresponding $g(r)$ profiles of P-P and P-S pair. **b** The snapshot of equilibrium Si-doped $\text{Li}_{4.125}\text{Si}_{0.125}\text{P}_{0.785}\text{S}_4\text{I}$ glass after AIMD simulations and the corresponding $g(r)$ profiles of P-P and P-S pair. The glass framework is represented as the solid gray lines. The light purple balls represent P atoms, the blue balls represent Si atoms, the orange balls represent S atoms dissolved in the interstitial sites, the yellow balls represent S atoms in the framework and the deep brown ball represent I atoms.

The corresponding revisions in our manuscript is listed below:

-Manuscript, Results, Page 9-10.

“In order to validate the proposed design approach, atomistic simulations using the AIMD technique were initially conducted. Given that our AIMD simulation methodology relies on the melt-quenching of crystalline materials, the above-mentioned $\text{Li}_4\text{PS}_4\text{I}$, which possesses the highest LiI content in reported crystalline sulfides was selected for our simulations. Even though the LiI content (33.3 mol %) in the $\text{Li}_4\text{PS}_4\text{I}$ is lower than our experimental samples (40 mol%), the changes in the glassy framework structure reflected during the simulation process can still provide valuable insights. As illustrated in Fig. 2a, the PS_4^{3-} polyhedral structure in $\text{Li}_4\text{PS}_4\text{I}$ glass undergoes significant disruption upon equilibration at 600 K, which is evident from the sharp peaks located at $\sim 2.30 \text{ \AA}$ and 3.60 \AA of the P-P pair and the broad peak observed in the range of $3.70\text{-}4.50 \text{ \AA}$ of the P-S pair in the corresponding $g(r)$ profile. The appearance of $\text{P}_4\text{S}_7^{4-}$ and $\text{P}_4\text{S}_6^{4-}$ can be inferred from these observations. Then, the impact of Si doping on I-rich $\text{Li}_4\text{PS}_4\text{I}$ glass was investigated by using $\text{Li}_{4.125}\text{Si}_{0.125}\text{P}_{0.785}\text{S}_4\text{I}$ as a representative precursor, as shown in Fig. S7. As revealed in Fig. S8, the introduction of SiS_4^{4-} successfully stabilizes the initial PS_4^{3-} structure even at a temperature as high as 2000 K. Consequently, $\text{Li}_{4.125}\text{Si}_{0.125}\text{P}_{0.785}\text{S}_4\text{I}$ glass exhibits a

predominantly monomeric structure upon equilibration at 600 K, as evidenced by the absence of observable dimers in the $g(r)$ profile (Fig. 2b), thereby validating that Si doping can effectively suppress dimers formation.

Furthermore, while the $g(r)$ profile of Li-I pair indicates a complete dissociation of Lil in both Li_4PS_4I and $Li_{4.125}Si_{0.125}P_{0.785}S_4I$ glass (Fig. S11a), the snapshots in Fig. 2a-b after equilibrium reveal that, compared to $Li_{4.125}Si_{0.125}P_{0.785}S_4I$, Li_4PS_4I exhibits a significant occupancy of S_x^{2-} in the interstitial positions. Considering the limited interstitial space, the presence of these non-framework S atoms will compress I atoms and cause them to move toward the framework due to the coulombic repulsion. Subsequently, by extracting the radial distribution between S atoms from the framework and I atoms, a tendency for some I and framework S to be closer is observed in Li_4PS_4I glass (Fig. S11b), which may hinder further Lil incorporation.”

Comment 2.4. *If accessible interstitial volume of the glass is the critical factor, does this imply that the same benefits for Lil “interstitial” incorporation could be achieved by altering processing (e.g., faster quenching)?*

Response to comment 2.4:

Thanks for your valuable comments!

This is a very interesting question. Although we may not be able to provide a definitive answer, we are more than willing to discuss this issue. In theory, elevated temperatures can increase the distance between atoms, potentially allowing for more Lil accommodation. From this perspective, faster quenching might be feasible. However, whether such a structure obtained from fast quenching can be maintained consistently at room temperature requires further exploration.

Comment 2.5. *As the authors correctly state on p. 4, there have been numerous justifications about the role of halogen dopants in mechanistic improvements of LPS-based materials. Does the current study shed any light on these?*

Response to comment 2.5:

Thanks for your valuable comments!

As previously reported, the halogen dopants Lil can help to decrease the elastic modulus of sulfide electrolytes, thereby enhancing the relative density of the cold-pressed electrolyte pellet.⁹ In this study, our optimized glass electrolytes with an ultra-high Lil content exhibit a remarkable relative density of 93.6%, surpassing that of Li₃PS₄ without halogen dopants (91.0%).

Based on this, we did the following revisions in our manuscript to emphasize the mechanistic improvements of our optimal glass.

-Manuscript, Results, Page 17.

“Furtherly, the cold-pressed Li₆PS₅Cl pellet exhibits a rather low relative density of 84.8 % attributed to enormous numbers of grain boundary. In comparison, the glass electrolyte 75 Li₂S- 25 P₂S₅ and P:Si = 6 exhibit relative densities of 91.0 % and 93.6 %, respectively, both surpassing the relative density of Li₆PS₅Cl. Notably, P:Si = 6 demonstrates a higher relative density than 75 Li₂S- 25 P₂S₅. This is primarily attributed to the halogen dopant Lil, which can effectively reduce the elastic modulus of sulfide electrolytes, thereby enhancing the relative density of cold-pressed electrolyte pellets.”

Comment 3. *Fig. 1f: I do not understand what is being shown here, and there is no scale on the free energy axis. Are these molecular calculations? And is this actually Gibbs free energy? If so, how was the entropy change computed and at what temperature? What is “coupled $\text{Li}_3\text{PS}_4/\text{Li}_4\text{SiS}_4$ ” (is this the complex in Fig. S6)? If so, it's hard to see how these calculations of energy will translate meaningfully to the glassy solid, where the Li^+ ions are mobile, probability density of Li^+ is diffuse, and the charges are screened by the matrix. As a result, the concept of local Li-containing molecular structure is difficult to grasp. The energy reduction upon coupling is shown as 60 kJ/mol (quite large), but I'm guessing that these results are skewed by dipole interaction effects, which are highly problematic because they depend on where the Li^+ ions are placed. Indeed, the dipole moments and electron distributions in the SI (Figs. S5 and S6) seem to confirm this (e.g., the huge dipole of 11.27D for Li_3PS_4 is clearly due to the placement of Li in the zero-temperature structure).*

Response to Comment 3:

Thanks for your valuable comments! Given that there are several questions in your third comment, we will address each one individually below.

Comment 3.1. *“I do not understand what is being shown here, and there is no scale on the free energy axis. Are these molecular calculations?”*

Response to Comment 3.1:

Firstly, we feel sorry that the way we presented Fig.1f in the manuscript is confusing. Fig. 1f primarily illustrates the Gibbs free energy changes during the ball milling process, where intermediate dimers ($\text{Li}_4\text{Si}_2\text{S}_6$ and $\text{Li}_4\text{P}_2\text{S}_7$) react with Li_2S to form monomers Li_4SiS_4 and Li_3PS_4 . The reason we did not include a scale is that the relative Gibbs free energy is more meaningful for investigating a reaction. Therefore, we have modified Fig. 1f to highlight the Gibbs reaction energy from $\text{Li}_4\text{Si}_2\text{S}_6$, $\text{Li}_4\text{P}_2\text{S}_7$ and $3\text{Li}_2\text{S}$ to isolated Li_3PS_4 - Li_3PS_4 and Li_4SiS_4 - Li_4SiS_4 in the figure. Consequently, the Gibbs reaction energies for all reactions can be obtained from the modified Fig. 1f (represented as Fig. R12 below, represented as Fig. 1d in the revised manuscript).

Fig. R12 The reaction Gibbs free energy from precursors to isolated $\text{Li}_3\text{PS}_4/\text{Li}_4\text{SiS}_4$ and coupled $\text{Li}_3\text{PS}_4/\text{Li}_4\text{SiS}_4$.

As you pointed out, these calculations are based on molecules. The choice of molecular calculations stems from the inherent uncertainty in the actual glassy state structure. While we can infer lithium diffusion behavior and radial distribution functions between atoms through AIMD approaches, the uncertainty in the glassy state structure poses challenges for static calculations. Furthermore, recognized that the earliest way to prepare the glassy solid is the melting and rapid quenching method, wherein the superheated solution is quickly cooled to form a glassy solid, we propose that the glassy state has the structural characteristics of a supercooled liquid, and its internal structural arrangement is more like a "solution". For the calculation of "solution" systems, due to its complexity, reported simulations often extract molecules from the mixed system for the calculation, which can also provide valuable insights.^{10, 11} Hence, the molecular calculations are applied to approximate the Gibbs energy changes in the reaction process in our study.

Comment 3.2. *And is this actually Gibbs free energy? If so, how was the entropy change computed and at what temperature?*

Response to Comment 3.2:

As mentioned above, molecular calculations were employed in Fig. 1f. Therefore, to answer these questions, it is noteworthy that, as outlined in the **Method**, we utilized the Gaussian software rather than VASP software to perform calculations. By default, gaussian performs the energy calculation and the corresponding thermal correction using normal conditions (T=298.15K, P = 1 atm). By calculating the frequency of each molecular vibrating mode, gaussian software can automatically perform the thermal correction to Gibbs free energy and give a result like Fig. R13.

Zero-point correction=	0.169068 (Hartree/Particle)
Thermal correction to Energy=	0.184735
Thermal correction to Enthalpy=	0.185679
Thermal correction to Gibbs Free Energy=	0.122875
Sum of electronic and zero-point Energies=	-5721.995729
Sum of electronic and thermal Energies=	-5721.980062
Sum of electronic and thermal Enthalpies=	-5721.979118
Sum of electronic and thermal Free Energies=	-5722.041922

Fig. R13 An example of Gaussian optimization + frequency calculation results.

In summary, the energy presented on Fig.1f is the Gibbs free energy and computed at room temperature. The entropy changes are automatically realized by the gaussian software by summarizing the vibrational mode of molecules. Given that our materials are solids with only vibrational degrees of freedom, the calculated Gibbs free energy results at the molecular level have a relatively high level of confidence.

Comment 3.3. *What is “coupled Li₃PS₄/Li₄SiS₄” (is this the complex in Fig. S6)? If so, it's hard to see how these calculations of energy will translate meaningfully to the glassy solid, where the Li⁺ ions are mobile, probability density of Li⁺ is diffuse, and the charges are screened by the matrix. As a result, the concept of local Li-containing molecular structure is difficult to grasp.*

Response to Comment 3.3:

Thanks for your valuable comments!

Firstly, the coupled Li₃PS₄/Li₄SiS₄ is the complex in Fig. S6.

Then, we understand that your primary concern is that the energy and dipole moment calculations for localized structures may not be representative for glassy state electrolytes, in

which Li^+ possesses a certain degree of mobility and does not remain fixed at a specific location. **This issue not only occurs in the glass but also exists in the energy calculation of crystalline electrolytes, which also exhibit a certain degree of Li^+ mobility. However, incorporating Li^+ mobility into the energy calculation is highly challenging.** On the one hand, though AIMD can simulate the Li^+ mobility, it cannot derive a precise result for energy calculation. On the other hand, though the static calculations in VASP and Gaussian can present a highly accurate result of energy and other physical quantities, they tend to optimize the structure to the “saddle point,” where energy and forces are minimized. In this case, the influence of Li^+ mobility is not presented.

At present, **the most prevailing theoretical energy deviation still relies on the static calculations.**^{12, 13} As for the well-studied crystalline solid-state electrolytes, the energy and other physical quantities calculated from the static calculation can align reasonably well with actual conditions. This provides the context for our use of molecular calculations to elucidate reaction energy and dipole moments.

Additionally, while Li diffusion occurs in solid-state electrolyte systems (whether crystalline or amorphous), it does not necessarily imply a uniform distribution of Li^+ in the system. In fact, the possible location of Li^+ is determined by the 'framework'. For example, in LGPS-type electrolytes, Li^+ diffusion primarily occurs along the c-axis, with weak diffusion along the ab-axis.¹⁴ In LPSC-type electrolytes, Li^+ diffusion involves intra-cage transport and inter-cage hopping.¹⁵ Therefore, the Li distribution results obtained from static calculations of Li-framework interactions may represent the average results of Li location in solid-state electrolytes. Considering that glass electrolytes are more like a “liquid”, the molecular calculations are thereby applied in our study to elucidate reaction energy and dipole moments. (See the response to **Comment 3.2**)

Comment 3.4. *The energy reduction upon coupling is shown as 60 kJ/mol (quite large), but I'm guessing that these results are skewed by dipole interaction effects, which are highly problematic because they depend on where the Li⁺ ions are placed. Indeed, the dipole moments and electron distributions in the SI (Figs. S5 and S6) seem to confirm this (e.g., the huge dipole of 11.27D for Li₃PS₄ is clearly due to the placement of Li in the zero-temperature structure).*

Response to Comment 3.4

Thanks for your valuable comments!

Firstly, we would like to clarify that the excessive dipole moment in Li₃PS₄ is initialed by our wrong selection of basis set. The basis set in our manuscript (B3LYP/6-311G+ (d)) is not suitable for accurate dipole moment calculations, which will always lead to unreasonably large results. To address this, we referred to relevant literature on dipole moment calculations and subsequently used the B3LYP/def2-TZVPD basis for improved accuracy in dipole moment calculations.¹⁶ The recalculated results are presented in Fig. R14 (represented as Figure S5 in the revised SI) and Fig. R15. Notably, with the appropriate basis set, the dipole moment values of Li₃PS₄ (5.905 Debye) and Li₄SiS₄ (2.973 Debye) align more closely with realistic expectations. However, through refined calculations, we have discovered that the dipole moment value of Li₃PS₄-Li₄SiS₄ (6.025 Debye) is slightly higher than that of Li₃PS₄. This is anticipated because the asymmetry of this coupled structure is enhanced. We will provide a detailed explanation of this in response to your **comments 4.2**.

Secondly, we would like to address the concern about the energy reduction after coupling. Based on our **corrected dipole calculations**, the dipole moment of the coupled structure (6.025 D) is higher than that of the Li₃PS₄ (5.905 D). Therefore, **attributing the decrease in energy to the dipole reduction is unreasonable**. Also, as Fig. R12 illustrates, the reaction energy for precursors to generate isolated Li₃PS₄-Li₃PS₄ and Li₄SiS₄-Li₄SiS₄ is 902 kJ mol⁻¹. Compared with this, the energy decrease of 60 kJ mol⁻¹ is not that large. Plus, the energy change of 60 kJ/mol is reasonable. In fact, the energy required for the formation of a hydrogen bond is approximately in the range of 15 to 30 kJ/mol, and the adsorption energy of Li⁺ ions on lithium metal surfaces is typically greater than 96 kJ/mol.¹⁷ The energy reduction after coupling can be attributed to (1) The enthalpy reduction initialed by the interactions of two molecules, and (2) The entropy increase caused by the interaction of two different anion clusters, which has been well-documented in the previous study.¹⁸

Fig. R14 The dipole moment of Li_3PS_4 molecule. The purple balls represent Li atoms, the yellow balls represent S atoms, and the orange ball represents the P atom.

Fig. R15 The dipole moment of Li_4SiS_4 and $\text{Li}_3\text{PS}_4\text{-Li}_4\text{SiS}_4$ complex. The purple balls represent Li atoms, the yellow balls represent S atoms, the grey balls represent Si atoms and the orange ball represents the P atom.

Comment 4. *Along the lines of comment #3, I do not understand the dipole discussion on p. 8 of the text. It is not immediately clear to me how lowering local dipole moments in the glass will necessarily decrease reaction rates for dimerization. Also, if I'm reading Fig. 1e correctly (see comment #6), the polarization of PS_x units is actually increasing upon Li_4SiS_4 incorporation. To probe the electrostatic effects, it would be better (and safer) to focus on the intrinsic polarity of the individual bonds (Si-S vs. P-S), as well as the consequence of mixing trivalent PS_4 complexes with tetravalent SiS_4 complexes (which could be crucial for disrupting the redox reaction).*

Response to Comment 4: Thanks for your valuable comments! Given that there are several questions in your 4th comment, we will address each one individually below.

Comment 4.1. *Along the lines of comment #3, I do not understand the dipole discussion on p. 8 of the text. It is not immediately clear to me how lowering local dipole moments in the glass will necessarily decrease reaction rates for dimerization.*

Response to Comment 4.1

Concerning the impact of dipole moments on reaction rates, we would like to firstly elucidate the mechanism of dimerization. In the Li-P-S system, two distinct dimers, $P_2S_6^{4-}$ and $P_2S_7^{4-}$, exist. The former arises from electron transfer within PS_4^{3-} (electrons transferring from S to P, resulting in P reduction), and our spectroscopic findings, including Raman and NMR data, confirm the predominance of $P_2S_6^{4-}$ within the dimeric structures. To mitigate the formation of $P_2S_6^{4-}$, the introduction of high LUMO metal cations such as Si^{4+} (less susceptible to reduction) can effectively work. On the other hand, drawing insights from studies on electron transfer reactions,¹⁹ we learned that the effect of dipole moments on electron transfer reactions is enormous. Increasing the dipole moments of the reactant or aligning the dipole with the direction of electron transfer can facilitate electron transfer reactions. In contrast, decreasing dipole moments of reactant or removing the dipole from the direction of electron transfer can suppress electron transfer reactions.

Moreover, we feel sorry that we made a wrong statement in our manuscript. The influence of dipole moments on electron transfer reactions concerning both thermodynamics and kinetics, we should not attribute the dipole effect only to the kinetics. **Consequently, we have revised the descriptions in the manuscript accordingly as below:**

-Manuscript, Results, Page 8.

“...The formation of P-P dimers results from a redox reaction between P^{5+} and S^{2-} (Equation 1). Given the stability of $P(+V)_2S_5$ molecules at room temperature, it is reasonable to conclude that the generation of $P(+IV)-P(+IV)$ dimers occurs due to localized overheating arising from the ball-milling process involved in synthesizing glassy electrolytes. Besides, based on the theory of electron transfer, the effect of dipole moments on electron transfer reactions is enormous. Increasing dipole moments of reactant or aligning the dipole with the direction of electron transfer can facilitate electron transfer reactions. Recognized this, the presence of significant dipole moments (5.905 Debye) induced by unsaturated $P=S$ bonds in Li_3PS_4 molecules may serve as an inducer for the acceleration of redox reactions (Fig. S5).”

Comment 4.2. Also, if I'm reading Fig. 1e correctly (see comment #6), the polarization of PS_x units is actually increasing upon Li_4SiS_4 incorporation. To probe the electrostatic effects, it would be better (and safer) to focus on the intrinsic polarity of the individual bonds (Si-S vs. P-S), as well as the consequence of mixing trivalent PS_4 complexes with tetravalent SiS_4 complexes (which could be crucial for disrupting the redox reaction).

Response to comment 4.2:

Your comment is highly constructive, and it has prompted us to **focus on the intrinsic polarity of Li_3PS_4 molecules** (the only species undergoing electron transfer reactions in the system), particularly considering the revised dipole calculation results that the dipole of coupled $Li_3PS_4-Li_4SiS_4$ is slightly higher than that of Li_3PS_4 (Response to **Comment 3.4**).

As illustrated in Fig. R14 and Fig. R16 (represented as Fig. S6 in the SI), before coupling, the **dipole moment of Li_3PS_4 is 5.905 Debye**. In contrast, **after coupling**, we observed a **reduction in the dipole moment of Li_3PS_4 to 4.760 Debye**. This dipole decrease is attributed to the charge compensation resulting from the coordination of Li^+ in Li_4SiS_4 with the S atom in the P=S bond, and it may potentially suppress electron transfer reactions.

Last but not least, we understand your concerns on the Fig. 1e that the polarization of Li_3PS_4 unit seems to be increased after coupling with Li. However, **this issue is caused by our wrong selection of projection directions and saturation levels**. After adjusting the 2D-imaging parameters, we can see there is no significant dipole increment of PS_4^{3-} after Li_4SiS_4 incorporation. Besides, it can be observed that the distribution of Li^+ (Positive regions) has been greatly changed due to the Si incorporation. (Fig. R17)

Fig. R16 The dipole moment of Li_4SiS_4 , $\text{Li}_3\text{PS}_4\text{-Li}_4\text{SiS}_4$ complex and Li_3PS_4 after complexing with Li_4SiS_4 . The purple balls represent Li atoms, the yellow balls represent S atoms, the grey balls represent Si atoms and the orange ball represents the P atom.

Fig. R17 The 2D electron density plot of $\text{Li}_4\text{PS}_4\text{I}$ **a** and $\text{Li}_{4.125}\text{Si}_{0.125}\text{P}_{0.785}\text{S}_4\text{I}$ **b** and the corresponding 2D atom distributions of $\text{Li}_4\text{PS}_4\text{I}$ **c** and $\text{Li}_{4.125}\text{Si}_{0.125}\text{P}_{0.785}\text{S}_4\text{I}$ **d**. The light purple tetrahedra represents PS_4^{3-} , the blue tetrahedra represents SiS_4^{4-} , the green balls represent Li atoms, the yellow ball represents S atoms and the deep purple ball represent I atoms.

The corresponding revision in our paper is listed as follows:

-Manuscript, Results, Page 8.

“...*(i)* Since the disorder nature of glassy electrolytes provides ideal environments for doping, replacing partial P_2S_5 with SiS_2 whose cations possess higher LUMO values (Fig. 1c) can decrease the total occurrence of redox reactions in the glass system. *(ii)* Besides, Li_4SiS_4 molecule without unsaturated bonds exhibits a far lower dipole moment of 2.973 Debye compared to that of Li_3PS_4 molecule. After the incorporation of Li_4SiS_4 , the Li_3PS_4 - Li_4SiS_4 complex displays a dipole moment of 6.025 Debye, while the dipole moment of Li_3PS_4 in the complex is decreased from 5.905 to 4.760 Debye (Fig. S6). This result indicates that the charge compensation resulting from the coordination of Li^+ in Li_4SiS_4 with the S atom in the P=S bond will help to decrease the dipole moment of Li_3PS_4 , which may potentially suppress electron transfer reactions between P^{5+} and S^{2-} ”

Comment 5. Fig. 1c: The caption says "LUMO value of P^{5+} and Si^{4+} "--in which system or complex? What is this energy with respect to? EV units should be eV.

Response to comment 5:

Thanks for your valuable comments!

The LUMO values represented in Fig.1c are for single cations P^{5+} and Si^{4+} in vacuum (not complex). LUMO stands for the "Lowest Unoccupied Molecular Orbital.", which represents the orbital in a molecule that is most easily accessible to incoming electrons during the redox reaction. Molecules with low-lying LUMOs are more easily to be reduced, as they can readily accept electrons during a chemical process. This energy is respect to the vacuum level.

Plus, we are sorry for our mistakes of units. The revised Fig.1c (denoted as Fig. R18 in the response letter, represented as Fig.1d in the revised manuscript) is presented below.

Fig. R18 The LUMO value of P^{5+} and Si^{4+} .

Comment 6. *Fig. 1d-e: Where are the atoms? Which plane is this along? Is the plot only showing the electron density contribution from the PSx units? I do not understand what I am supposed to take away from this plot. It might be better to zoom in on a specific interaction and superimpose the atoms to see how the electron density is being distorted.*

Response to comment 6:

Thanks for your valuable comments! Based on your concerns, we have revised Fig. 1d-e in the initial manuscript. The updated figures are both presented in both **Comment 4** (Fig. R17) and below (Fig. R19) for your convenience.

Fig. R19 The 2D electron density plot of $\text{Li}_4\text{PS}_4\text{I}$ **a** and $\text{Li}_{4.125}\text{Si}_{0.125}\text{P}_{0.785}\text{S}_4\text{I}$ **b** and the corresponding 2D atom distributions of $\text{Li}_4\text{PS}_4\text{I}$ **c** and $\text{Li}_{4.125}\text{Si}_{0.125}\text{P}_{0.785}\text{S}_4\text{I}$ **d** The light purple tetrahedra represents PS_4^{3-} , the blue tetrahedra represents SiS_4^{4-} , the green balls represent Li atoms, the yellow ball represents S atoms and the deep purple ball represent I atoms.

However, during the revision process, particularly when addressing theoretical aspects, we find it unsuitable to incorporate Fig. 1d-e into the revised manuscript. Primarily, the extraction of meaningful information from the figures might be challenging for readers. Furthermore, given the predominant focus of our research on glass electrolytes, where atoms do not occupy specific sites, the computational outcomes relying on crystal electrolytes in Fig. 1d-e may not accurately depict the true scenario.

Consequently, we have decided to substitute Fig. 1d-e with figures illustrating the models and formation mechanisms of dimers and P-P dimers (Fig. R20, represented as

Fig. 1 in the revised manuscript). In this way, we believe the readers can understand better about our logic flow.

Fig. R20 Analysis on the Halogen dopants' dissolving in the glass sulfide. **a** Representation of the incorporating process of halogen dopants in the glass network. **b** The schematic illustration of ideal and real LiI-doped 75Li₂S-25P₂S₅ glass network structure. **c** The formation mechanism of P-P dimers and dimers. **d** The LUMO value of P⁵⁺ and Si⁴⁺. **e** The reaction Gibbs free energy from precursors to isolated Li₃PS₄/Li₄SiS₄ and coupled Li₃PS₄/Li₄SiS₄. **f** The schematic diagram of the Si-doping effect. The purple balls represent P atoms, the yellow balls represent S atoms in the glass framework and the blue balls represent Si atoms.

Comment 7. Label the atom colors in all figure captions (also in the SI).

Response to comment 7:

Thanks for your valuable comments!

The colors of different atoms have been labeled in all figure captions. We are sorry for our careless dealing.

Below are the revised captions of figures with atoms.

The revised caption of Fig. 3 is listed below:

-Manuscript, Introduction, the caption of Fig. 1.

“Fig. 1 Analysis on the Halogen dopants' dissolving in the glass sulfide. **a** Representation of the incorporating process of halogen dopants in the glass network. **b** The schematic illustration of ideal and real LiI-doped $75\text{Li}_2\text{S}-25\text{P}_2\text{S}_5$ glass network structure. **c** The formation mechanism of P-P dimers and dimers. **d** The LUMO value of P^{5+} and Si^{4+} . **e** The reaction Gibbs free energy from precursors to isolated $\text{Li}_3\text{PS}_4/\text{Li}_4\text{SiS}_4$ and coupled $\text{Li}_3\text{PS}_4/\text{Li}_4\text{SiS}_4$. **f** The schematic diagram of the effect of Si-doping. The purple balls represent P atoms, the yellow balls represent S atoms in the glass framework and the blue balls represent Si atoms.”

-Manuscript, Result, the caption of Fig. 2.

“Fig. 2 Investigations of anion clusters in the sulfide glass after the Si-incorporation. **a** The snapshot of equilibrium $\text{Li}_4\text{PS}_4\text{I}$ glass after AIMD simulations and the corresponding $g(r)$ profiles of P-P and P-S pair. **b** The snapshot of equilibrium Si-doped $\text{Li}_{4.125}\text{Si}_{0.125}\text{P}_{0.785}\text{S}_4\text{I}$ glass after AIMD simulations and the corresponding $g(r)$ profiles of P-P and P-S pair. The glass framework is represented as the solid gray lines. The light purple balls represent P atoms, the blue balls represent Si atoms, the orange balls represent S atoms dissolved in the interstitial sites, the yellow balls represent S atoms in the framework and the deep brown ball represent I atoms. **c** Raman, ^{31}P and ^{29}Si ss-NMR spectra of blank, P:Si =6 and P:Si =4 electrolytes. **d** Ratios of anion clusters and minimum $V_{\text{Li}_2\text{S}_x}$ derived from Raman, ^{31}P and ^{29}Si ss-NMR spectra.”

-Manuscript, Result, the caption of Fig. 3.

“Fig. 3 Phase characterizations and ion transport capability of the synthesized materials. **a** The XRD spectra of synthesized series electrolytes. **b** The ionic conductivity of synthesized series electrolytes. **c** The Li^+ conduction activation energy of synthesized series electrolytes. **d** The Lithium MSD plot for $\text{Li}_4\text{PS}_4\text{I}$ (orange) and $\text{Li}_{4.125}\text{Si}_{0.125}\text{P}_{0.785}\text{S}_4\text{I}$ (blue) glass at 600 K from AIMD simulations. **e** The Li-ion

trajectory (blue) in $\text{Li}_4\text{PS}_4\text{I}$ (top) and $\text{Li}_{4.125}\text{Si}_{0.125}\text{P}_{0.785}\text{S}_4\text{I}$ (bottom) glass at 600 K from AIMD simulations.

The orange tetrahedra represents PS_4^{3-} , the red tetrahedra represents SiS_4^{4-} .

-Supporting Information, the caption of Fig. S1.

"Fig. S1 Structures of typical anion clusters in $75\text{Li}_2\text{S}-25\text{P}_2\text{S}_5$ glass. The light purple balls represent P atoms and the yellow balls represent S atoms."

-Supporting Information, the caption of Fig. S2.

"Fig. S3 The crystal structure of $\text{Li}_4\text{PS}_4\text{I}$. The red tetrahedra represents PS_4^{3-} , the yellow balls represent S atoms and the deep purple balls represent I atom."

-Supporting Information, the caption of Fig. S3.

"Fig. S2. The structure of anion clusters in the $\beta\text{-Li}_3\text{PS}_4$ crystal (**left**). A snapshot of $75\text{Li}_2\text{S}-25\text{P}_2\text{S}_5$ glass exhibiting the existence of $\text{P}_2\text{S}_7^{4-}$ after equilibrium (**middle**). A snapshot of $75\text{Li}_2\text{S}-25\text{P}_2\text{S}_5$ glass exhibiting the existence of $\text{P}_2\text{S}_6^{4-}$ after equilibrium (**right**). The red tetrahedra represents PS_4^{3-} , the light purple balls represent P atoms, the yellow balls represent S atoms and the green balls represent Li atom."

-Supporting Information, the caption of Fig. S4.

"Fig S4 Snapshots of $\text{Li}_4\text{PS}_4\text{I}$ at different temperatures during the annealing-cooling process. The light purple tetrahedra represents PS_4^{3-} , the blue tetrahedra represents SiS_4^{4-} , the green balls represent Li atoms, the yellow ball represent S atoms and the deep purple balls represent I atoms."

-Supporting Information, the caption of Fig. S5.

"Fig. S5. The dipole moment of Li_3PS_4 molecule. The purple balls represent Li atoms, the yellow balls represent S atoms, and the orange ball represents the P atom."

-Supporting Information, the caption of Fig. S6.

"Fig. S6 The dipole moment of Li_4SiS_4 , $\text{Li}_3\text{PS}_4\text{-Li}_4\text{SiS}_4$ complex and Li_3PS_4 after complexing with Li_4SiS_4 . The purple balls represent Li atoms, the yellow balls represent S atoms, the grey balls represent Si atoms and the orange ball represents the P atom."

-Supporting Information, the caption of Fig. S7.

"Fig. S7 The crystal structure of $\text{Li}_{4.125}\text{Si}_{0.125}\text{P}_{0.875}\text{S}_4\text{I}$. The red tetrahedra represents the PS_4^{3-} , the blue tetrahedra represents the SiS_4^{4-} , the yellow balls represent sulfur atoms and the deep purple balls represent I atoms."

-Supporting Information, the caption of Fig. S8.

"Fig. S8 Snapshots of $\text{Li}_{4.125}\text{Si}_{0.125}\text{P}_{0.875}\text{S}_4\text{I}$ at different temperatures during the annealing-cooling

process. The light purple tetrahedra represents PS_4^{3-} , the blue tetrahedra represents SiS_4^{4-} , the green balls represent Li atoms, the yellow balls represent sulfur atoms and the deep purple balls represent I atoms.

-Supporting Information, the caption of Fig. S9.

“Fig. S9 The Crystal structure of Li_4PS_4I applied in the manuscript and other two additional structure employed for supplementary validation. **b** The Crystal structure of $Li_{4.125}Si_{0.125}P_{0.785}S_4I$ applied in the manuscript and other two additional structure employed for supplementary validation. The light purple tetrahedra represents PS_4^{3-} , the blue tetrahedra represents SiS_4^{4-} , the green balls represent Li atoms, the yellow ball represent S atoms and the deep purple balls represent I atoms.”

Comment 8. Balance Equation 1.

Response to comment 8:

Thanks for your valuable comments!

We have balanced the Equation 1 and highlighted that in the revised manuscript in yellow. The revised Equation 1 is represented below:

Comment 9. Fig. 3 caption: State clearly which results are experimental and which are theoretical. How do the simulation-derived activation energies compare with the experiments?

Response to comment 9:

Thanks for your valuable comments!

In the revised caption below, the results from AIMD simulations are particularly noted. Plus, for the caption of Fig. 3b-c, it has been noted that the ionic conductivity and the activation energy are of our synthesized series electrolytes.

“Fig. 3 Phase characterizations and ion transport capability of the synthesized materials. **a** The XRD spectra of synthesized series electrolytes. **b** The ionic conductivity of synthesized series electrolytes. **c** The Li^+ conduction activation energy of synthesized series electrolytes. **d** The Lithium MSD plot for Li_4PS_4I (orange) and $Li_{4.125}Si_{0.125}P_{0.785}S_4I$ (blue) glass at 600 K from AIMD simulations. **e** The Li-ion trajectory (blue) in Li_4PS_4I (top) and $Li_{4.125}Si_{0.125}P_{0.785}S_4I$ (bottom) glass at 600 K from AIMD simulations. The orange tetrahedra represents PS_4^{3-} , the red tetrahedra represents SiS_4^{4-} .”

As for the question “How do the simulation-derived activation energies compare with the experiments?”, we did calculate the Li^+ diffusion barrier in $Li_{4.125}Si_{0.125}P_{0.785}S_4I$. Based on the Lithium MSD plot for $Li_{4.125}Si_{0.125}P_{0.785}S_4I$ at 600, 700, 800 and 900 K in Fig. R21, we derived the corresponding Arrhenius plot of $Li_{4.125}Si_{0.125}P_{0.785}S_4I$ (Fig. R22), and the activation energy

of $\text{Li}_{4.125}\text{Si}_{0.125}\text{P}_{0.785}\text{S}_4\text{I}$ was calculated to be 0.232 eV, which is close to that of our experimental electrolytes (0.253 eV) and indicates a low Li^+ diffusion barrier in $\text{Li}_{4.125}\text{Si}_{0.125}\text{P}_{0.785}\text{S}_4\text{I}$ glass with both Si and LiI doping. It should be noted that despite the simulated electrolyte having a lower LiI content compared to the experimentally synthesized electrolyte, it exhibits a lower activation energy. This is primarily attributed to the fact that AIMD simulations of Li^+ diffusions are conducted at elevated temperatures and tend to yield activation energy values lower than those observed in actual experiments, which has been widely observed in the simulation results of solid electrolytes.^{20, 21}

However, as mentioned above and in our response to **comment 2.1**, there is a **significant disparity between the composition of the simulated structure $\text{Li}_{4.125}\text{Si}_{0.125}\text{P}_{0.785}\text{S}_4\text{I}$ and the electrolyte synthesized in our experiments**. Consequently, comparing the activation energy of simulated $\text{Li}_{4.125}\text{Si}_{0.125}\text{P}_{0.785}\text{S}_4\text{I}$ and the activation energy of the experimentally synthesized electrolyte lacks substantive significance. Also, we would like to clarify that the AIMD simulations of Li^+ diffusion kinetics in Fig. 3d-e were conducted to **mainly illustrate that Si doping itself may also enhance the ion conductivity of the Li_3PS_4 system rather than completely relying on this simulation results to explain the overall enhancement of ionic conductivity in our synthesized electrolytes**, as mentioned in our response to **comment 2.2**. For those reasons, we would not like to address the activation energy obtained from simulated $\text{Li}_{4.125}\text{Si}_{0.125}\text{P}_{0.785}\text{S}_4\text{I}$ in our manuscript.

Fig. R21 The Lithium MSD plot for $\text{Li}_{4.125}\text{Si}_{0.125}\text{P}_{0.785}\text{S}_4\text{I}$ at 600, 700, 800 and 900 K.

Fig. R22 The Arrhenius plot of $\text{Li}_{4.125}\text{Si}_{0.125}\text{P}_{0.785}\text{S}_4\text{I}$ derived by theoretical calculations.

Comment 10. *Some of the language and phrasing is awkward, particularly in the early sections of the paper. These could benefit from a closer edit to make them more readable.*

Response to comment 10:

Thanks for your valuable comments!

We have revised the language and phrase in the whole manuscript.

Particularly, the corresponding revision of the early sections in our paper is presented below.

“Abstract

Glassy sulfide solid electrolytes (SSE) represent a crucial category of superionic conductors for next-generation all-solid-state lithium metal batteries (ASSLMBs) due to their long-range disorder and grain-boundary-free nature, resulting in decent ionic conductivity and high relative density. However, the lack of understanding in glass formation chemistry has considerably impeded their development. Herein, we propose interstitial volume (V_{int}) as the decisive factor for halogen dopant solubility in a glass matrix. A $\text{Li}_3\text{PS}_4\text{-Li}_4\text{SiS}_4$ complex structure inside the glassy network was constructed to release the interstitial volume. As a result, the dissolving amount of LiI in $75\text{Li}_2\text{S-25P}_2\text{S}_5$ glass can reach a reported-high value of 40 mol%. The resulting glass electrolyte exhibits a record-high ionic conductivity ($2.21 \times 10^{-3} \text{ S cm}^{-1}$)

among glassy SSEs. In addition, a novel glassy/crystalline composite electrolyte was designed to address the shortcomings of high-Cl content argyrodite-type sulfides by using our synthesized glass as the filler. The composite electrolyte showcases a record-high CCD value of 2.9 mA cm^{-2} among sulfide electrolytes, and demonstrates ultra-stable cycling at a current density of 1 mA cm^{-2} and a practical capacity of 3 mAh cm^{-2} . Furthermore, ASSLMs (Li/NCM83125) employing the composite electrolyte retain 82.4 % of the initial discharging capacity after 500 cycles at 0.2 C. This work reveals an optimized protocol for dissolution of halogen dopants in glass SSEs to achieve high-performance glassy electrolytes.

Introduction

All-solid-state lithium metal batteries (ASSLMs) with improved safety and high energy-density have been proposed as one of the most competitive contenders to conventional lithium-ion batteries (LIBs). Central to ASSLMs are solid electrolytes (SEs), which govern the transport of charge carriers within the battery. Among the diverse families of SEs, sulfide solid electrolytes (SSEs) with superior ionic conductivity and appropriate Young's modulus are recognized as the most promising among present families of SEs. In SSEs, glassy sulfides represent distinct advantages over their crystalline and glass-ceramic counterparts. Firstly, the disorder characteristic of glass enables a relatively "free" environment to incorporate various species within a broad range, which provides a facile method for optimizing electrolytes' performances in ionic conductivity, chemical/interfacial stability, and mechanical properties. Secondly, due to the absence of grain boundaries in the glass, high relative density can be easily achieved in the cold-pressed pellet of glassy SSEs. Based on the fracture mechanics, the pellet with high relative density will typically exhibit a flawless surface, thereby suppressing Li metal intrusions. Additionally, glassy SSEs can help to eliminate the electronic conductivity arising from heterogeneous grain boundaries, which can suppress Li dendrite growth inside electrolytes.

The history of glassy sulfides can be traced back to the 1980s, when glassy sulfide was discovered to exhibit significantly enhanced ionic conductivity compared to glassy oxides. Since then, substantial efforts have been devoted to exploring the structure and underlying Li^+ conducting mechanisms of glassy sulfides. A typical glassy network in sulfide glass is generated by the chemical reaction between glass formers (e.g., P_2S_5 , SiS_2 , B_2S_3) and glass modifiers (e.g., Li_2S), where glass formers serve as the glass network, and glass modifiers break the sulfur chain to reconstruct the network, thereby incorporating charge carrier (Li^+) into the glass matrix. To enhance the chemical and electrochemical stability of glassy sulfides, elemental chemistry-based optimization is commonly applied. For instance,

substituting the glass former P_2S_5 with SnS_2 or In_3S_2 can enhance the chemical stability of electrolytes according to the soft-hard-acid-base theory. As for the Li^+ conducting kinetics in the glass, Li^+ concentration, Li^+ -anion binding energy and the strain energy for structure deformation are all accounted based on the Anderson and Stuart energy barrier model. Accordingly, there are two practical pathways to improve the ionic conductivity of glassy sulfides. (1) Increasing the Li^+ concentration. (2) Tuning the chemistry and network structure of glass to weaken the Li^+ -anion binding and promote the structure-deforming ability. To pursue high performance glassy sulfides, various strategies have been reported over past decades. For instance, enhancing the relative ratio of Li_2S in order to increase the Li^+ concentration in the Li_2S - P_2S_5 glass has proved to be practical in facilitating Li^+ transport. Also, mixing glass forming/modifying anions or cations in binary glass systems to create a ternary glass system with altered Li^+ -anion binding strength and structure deforming capability has also produced positive effects in decreasing the Li^+ -motion activation energy. Among these strategies, the incorporation of halogen dopants LiX ($X = Cl, Br, I$) has been particularly desirable as it achieves simultaneous enhancements of ionic conductivity and Li/electrolyte interfacial stability in glassy SSEs. The enhancing mechanisms are that the increase of Li^+ concentration and the weakened Li^+ -anion binding in glass can accelerate Li^+ transport, and the formation of LiX -rich solid electrolyte interphase (SEI) at the interface can passivate the Li/SSE interphase.

Recent investigations have revealed halogen dopants behave differently from traditional glass formers or modifiers. Instead, they dissolve interstitially within the glass network (Fig. 1a). Moreover, previous research discovered that the ionic conductivity and Li/electrolyte interfacial stability of LiX -incorporated sulfide glass are positively correlated with the concentration of halogen dopants in the glass. This intriguing phenomenon indicates that increasing the solubility of LiX can further optimize glassy SSEs to achieve unprecedented levels of ionic conductivity and Li/electrolyte interfacial stability. Prior works have reported that the dissociation energy of the halogen dopants ("solute") plays a significant role in their solubility. Attributed to this, LiI , with the lowest dissociation energy, exhibits the highest solubility among LiX . Besides, the property of glass network ("solvent") and their interaction with the halogen dopants also significantly influence the dissolution process of LiX . However, few studies have delved into this complex interplay. Addressing this knowledge gap will offer a feasible method to enhance the solubility of halogen dopants, thereby pushing the boundaries of ionic conductivity and Li/electrolyte interfacial stability in glassy SSEs. Therefore, an in-depth deciphering of the halogen dopants' dissolution process is highly necessary.

In this work, we propose the interstitial volume of glass (V_{int}) as a vital factor for halogen dopants' dissolution behavior in a glassy network. We first perform theoretical simulations to unravel the structural chemistry of 75Li₂S-25P₂S₅ glassy SSEs with Lil dopants. It is discovered that lithium sulfide or lithium polysulfide (Li₂S_x) produced by the side reactions during synthesis will occupy interstitial sites, thus hampering Li dissolution at the interstitial sites. To resolve this issue, a novel synthetic strategy is proposed to change the glass formation reaction pathways by constructing a stable Li₃PS₄-Li₄SiS₄ complex structure, thus effectively suppressing Li₂S_x formation. As a result, the dissolving amount of Lil in 75Li₂S-25P₂S₅ glass reached a record-high value of 40 mol%. The resulting glassy electrolytes exhibit one of the highest ionic conductivity values of $2.21 \times 10^{-3} \text{ S cm}^{-1}$. When directly applied as the SE, this glassy SSE can exhibit the highest critical current density (CCD) values (1.2 mA cm⁻² with a capacity of 1.2 mAh cm⁻², 4.0 mA cm⁻² with a capacity of 0.1 mAh cm⁻²) among glassy SSEs. Moreover, using this optimal glassy SSE to design novel glassy/crystalline composite SSEs can effectively address crystalline argyrodite-type SSEs' low relative density and high electronic conductivity issues. The resulting composite electrolyte exhibits a record-high CCD value (2.9 mA cm⁻² with a capacity of 2.9 mAh cm⁻²) among SSEs. Additionally, the Li symmetric cell employing the composite electrolyte demonstrates remarkably stable cycling for 3000 hours at a current density of 1 mA cm⁻² and an areal capacity of 1 mAh cm⁻². Even under more stringent conditions, with a current density of 1 mA cm⁻² and an areal capacity of 3 mAh cm⁻², it maintains functionality for a duration of 900 hours. Furthermore, the Li/NMC83125 full cells employing the composite electrolyte exhibit a retention of 82.4 % of its initial discharging capacity after 500 cycles at a current density of 0.2 C. The deciphering of halogen dissolution chemistry presented in this study will provide a pragmatic perspective for developing versatile and high-performance glass sulfide materials in the future.

Results

The competition mechanism of halogen dopants and Li₂S_x in interstitial volume

The structural chemistry of the state-of-the-art 75Li₂S-25P₂S₅ glassy SSEs with the simplified formula of Li₃PS₄ has been well established. As previously reported, its glass network mainly comprises ortho-thiophosphate (PS₄³⁻, monomer) and small amounts of hypo-thiodiphosphate (P₂S₆⁴⁻, P-P dimer), as well as pyro-thiophosphate (P₂S₇⁴⁻, dimer) (Fig. S1). Then, in order to investigate the changes in 75Li₂S-25P₂S₅ glass structure after Lil incorporation, we conducted an ab-initio molecular dynamic (AIMD) simulations of 75Li₂S-25P₂S₅ glass and Lil-doped 75Li₂S-25P₂S₅ glass by melting and rapid-

quenching crystalline Li_3PS_4 and Li_4PS_4 (Fig. S2). As suggested by our simulation results (Fig. S3 and S4), the incorporation of Lil does not significantly impact the chemistry of the glass network. Both $75\text{Li}_2\text{S}-25\text{P}_2\text{S}_5$ glass and Lil-doped $75\text{Li}_2\text{S}-25\text{P}_2\text{S}_5$ glass exhibit a similar glass network predominantly composed of monomers, accompanied by small quantities of P-P dimers and dimers. Also, our simulation results confirm the insight from the previous report that the doped Lil does not replace the sulfur in the monomer, dimer, or P-P dimer, but instead dissolves "interstitially" in the glass network.⁴ Based on the law of conservation, the generation of P-P dimers and dimers will result in the generation of Li_2S_x , which remains in the interstitial position of the glass network rather than modifying the glassy network. Considering that both Lil and Li_2S_x reside in the interstitial sites, the V_{int} occupied by Li_2S_x will impede the further dissolution of Lil. Ideally, in a Lil-doped $75\text{Li}_2\text{S}-25\text{P}_2\text{S}_5$ glass with the minimal formula of $\text{Li}_3\text{PS}_4 \cdot x\text{Lil}$, a large V_{int} would be created for Lil's dissolution if the glass network solely comprised monomers. However, side reactions during the actual preparation process will inevitably result in the generation of dimers and P-P dimers, which produces a monomer/dimer hybrid network. This leads to a relatively low V_{int} compared to the V_{int} of monomer-based glass network for Lil's incorporation since Li_2S_x occupies a certain space (Fig. 1b).

Design of high-performance glassy sulfide

Realizing the competition on the interstitial position of the Lil and Li_2S_x , it is not arduous to derive that decreasing the relative ratio of Li_2S_x in the glass system will create more V_{int} to facilitate the further dissolving of Lil. The origin of Li_2S_x species can be attributed to the formation chemistry of dimers/P-P dimers during the preparation process, necessitating the tracing of such reactions (Fig. 1c). The formation of P-P dimers results from a redox reaction between P^{5+} and S^{2-} (Equation 1). Given the stability of $\text{P}(+\text{V})_2\text{S}_5$ molecules at room temperature, it is reasonable to conclude that the generation of $\text{P}(+\text{IV})-\text{P}(+\text{IV})$ dimers occurs due to the localized overheating arising from the ball-milling process involved in synthesizing glassy electrolytes. Besides, based on the theory of electron transfer, the effect of dipole moments on electron transfer reactions is enormous. Increasing dipole moments of the reactant or aligning the dipole with the direction of electron transfer can facilitate electron transfer reactions. Recognized this, the presence of significant dipole moments (5.905 Debye) induced by unsaturated $\text{P}=\text{S}$ bonds in Li_3PS_4 molecules may serve as an inducer for the acceleration of redox reactions (Fig. S5). As for the formation of dimers ($\text{P}_2\text{S}_7^{4-}$), it can be induced by both reversible reactions between P-P dimers and Li_2S_x (Equation 2), as well as incomplete reactions between glass formers and glass modifiers.

Consequently, a strategy aimed at reducing the ratios of Li_2S_x in $75\text{Li}_2\text{S}-25\text{P}_2\text{S}_5$ systems should prioritize decreasing the potential for self-redox of Li_3PS_4 and promoting reactions between glass formers and glass modifiers.

Accordingly, we propose a modification of glass network by incorporating SiS_2 . Firstly, replacing partial P_2S_5 with SiS_2 , whose cations possess higher LUMO values (Fig. 1d) can decrease the total occurrence of redox reactions in the glass system. Besides, the Li_4SiS_4 molecule without unsaturated bonds exhibits a far lower dipole moment of 2.973 Debye compared to that of the Li_3PS_4 molecule. After the incorporation of Li_4SiS_4 , the Li_3PS_4 - Li_4SiS_4 complex displays a dipole moment of 6.025 Debye, while the dipole moment of Li_3PS_4 in the complex is decreased from 5.905 to 4.760 Debye (Fig. S6). This result indicates that the charge compensation resulting from the coordination of Li^+ in Li_4SiS_4 with the S atom in the $\text{P}=\text{S}$ bond will help to decrease the dipole moment of Li_3PS_4 , which may potentially suppress electron transfer reactions between P^{5+} and S^{2-} . Then, as illustrated in Fig. 1e, the interaction between Li_3PS_4 and Li_4SiS_4 lowers the overall Gibbs free energy and helps to facilitate the transformation from precursors to monomers. Considering the above factors, Si-doping will decrease the amounts of P-P dimers and dimers in the glassy sulfide, thereby lowering the ratio of Li_2S_x to release more V_{int} for the incorporation of LiI (Fig. 1f). Besides, proper doping amounts of SiS_2 will not produce mixed conducting interphase (MCI) with certain electronic conductivity to result in a deterioration of the Li/electrolyte interface, as previously reported. Accordingly, a series of Si-doped $0.6((75+0.5x)\text{Li}_2\text{S}-(25-0.5x)\text{P}_2\text{S}_5-x\text{SiS}_2)$ -40LiI electrolytes were synthesized. The investigated systems are thereby denoted as blank (without SiS_2 doping), P:Si = 7 ($x = 6.25$), P:Si = 6 ($x = 7.14$), P:Si = 5 ($x = 8.33$), P:Si = 4 ($x = 10$), separately.

In order to validate the proposed design approach, atomistic simulations using the AIMD technique were initially conducted. Given that our AIMD simulation methodology relies on the melt-quenching of crystalline materials, the $\text{Li}_4\text{PS}_4\text{I}$ mentioned above which possesses the highest LiI content in reported crystalline sulfides, was selected for our simulations. Even though the LiI content (33.3 mol %) in the $\text{Li}_4\text{PS}_4\text{I}$ is lower than our experimental samples (40 mol%), the changes in the glassy framework structure reflected during the simulation process can still provide valuable insights. As illustrated in Fig. 2a, the PS_4^{3-} polyhedral structure in $\text{Li}_4\text{PS}_4\text{I}$ glass undergoes significant disruption upon equilibration at 600 K, which is evident from the sharp peaks located at $\sim 2.30 \text{ \AA}$ and 3.60 \AA of the P-P pair and the

broad peak observed in the range of 3.70-4.50 Å of the P-S pair in the corresponding $g(r)$ profile. The appearance of $P_4S_7^{4-}$ and $P_4S_6^{4-}$ can be inferred from these observations. Then, the impact of Si doping on I-rich Li_4PS_4I glass was investigated by using $Li_{4.125}Si_{0.125}P_{0.785}S_4I$ as a representative precursor, as shown in Fig. S7. As revealed in Fig. S8, the introduction of SiS_4^{4-} successfully stabilizes the initial PS_4^{3-} structure even at a temperature as high as 2000 K. Consequently, $Li_{4.125}Si_{0.125}P_{0.785}S_4I$ glass exhibits a predominantly monomeric structure upon equilibration at 600 K, as evidenced by the absence of observable dimers in the $g(r)$ profile (Fig. 2b), thereby validating that Si doping can effectively suppress dimers formation. Moreover, to clarify that the pair distribution results do not depend on the input structures employed during our modeling, two additional validation structures for both Li_4PS_4I and $Li_{4.125}Si_{0.125}P_{0.785}S_4I$ were constructed and applied to the consistent molecular simulations, respectively (Fig. S9). As a result, comparable outcomes across varying structures of the same material were observed (Fig S10). Furthermore, while the $g(r)$ profile of Li-I pair indicates a complete dissociation of LiI in both Li_4PS_4I and $Li_{4.125}Si_{0.125}P_{0.785}S_4I$ glass (Fig. S11a), the snapshots in Fig. 2a-b after equilibrium reveal that, compared to $Li_{4.125}Si_{0.125}P_{0.785}S_4I$, Li_4PS_4I exhibits a significant occupancy of S_x^{2-} in the interstitial positions. Considering the limited interstitial space, the presence of these non-framework S atoms will compress I atoms and cause them to move toward the framework due to the coulombic repulsion. Subsequently, by extracting the radial distribution between S atoms from the framework and I atoms, a tendency for some I and framework S to be closer is observed in Li_4PS_4I glass (Fig. S11b), which may hinder further LiI incorporation.”

Reviewer #3:

This manuscript introduces the concept of 'interstitial volume' to propose a method for designing the composition of sulfide glassy electrolytes with Lil additive. Both experimental and computational calculations have been well-executed to support this research. The concept of interstitial volumes formed from PS_4^{3-} monomers is interesting and provides valuable insights into how to incorporate Lil into glassy electrolytes. Before the publication of this manuscript, it is necessary to consider that the "interstitial volume" concept explored in this study can be applied more broadly. I believe that this research, which involves synthesizing amorphous solid electrolytes with the addition of Lil and understanding their structure, can be highly valuable. I recommend the publication of this study if the following questions are addressed.

Response: Thanks for your positive comments! We have addressed each point in our responses below, and we believe that your kind suggestions significantly enhance the overall quality of this work.

Comment 1. *Sulfide solid electrolytes like $\text{Li}_3\text{PS}_4\text{-Li}_4\text{MS}_4$ ($M = \text{Sn}, \text{Ge}, \text{Si}$, etc) with reasonable Li-ion conductivities have already been reported. Why did the authors choose Si, and can this “interstitial volume” concept be applied to various M^{4+} (including Sn, Ge), not only Si? Additionally, the precursor SiS_2 is challenging to synthesize and expensive.*

Reply to comment 1:

Firstly, the reason we chose Si is primarily driven by our consideration for the stability of the electrolyte against lithium metal. We aim to enhance the solubility of LiI without compromising the stability of the electrolyte against lithium metal. As previously reported, proper doping amounts of Si into sulfide electrolytes will not produce mixed conducting interphase (MCI) with certain electronic conductivity to result in a deterioration of the Li/electrolyte interface.^{22, 23} However, the doping of metal cations Ge^{4+} and Sn^{4+} may produce significant MCI, which will deteriorate the battery performance.^{24, 25, 26} Secondly, we consider that whether the “interstitial volume” concept can be applied to Sn and Ge is worth investigating. It is true that Sn and Ge, like Si, will not undergo electron transfer reactions with S to generate reduced-state dimers as $\text{P}_2\text{S}_6^{4-}$. However, based on the hard and soft acid-base theory, Sn-S bonds and Ge-S bonds are stronger than Si-S and P-S bonds.²⁷ This implies that it is more difficult for the glass modifier Li_2S to react with the glass former SnS_2 or GeS_2 to form monomers. Finally, regarding the cost of SiS_2 , we did consider this aspect during our research. Taking Li_2S as an example, which has a similarly high cost and synthetic challenges, the community still views the Li-sulfide electrolyte as a potential solid electrolyte material because the elemental cost of Li_2S is not high. Once there is a significant demand in the industry, more facile synthesis routes and more affordable material supplies are likely to emerge. Therefore, though SiS_2 currently carries a high price due to the complexity of its production process, we believe that the cost of SiS_2 materials can decrease with increased demand from the industry.

Comment 2. *In the existing $\text{Li}_3\text{PS}_4\text{-LiI}$ system, sufficient ion conductivity and stability were already reported. Please explain how the addition of Li_4SiS_4 can offer advantages for battery performance improvement beyond inhibiting the formation of p-p dimers.*

Response to comment 2:

Thanks for your valuable comments! Given that the $\text{Li}_3\text{PS}_4\text{-LiI}$ glass electrolyte already possesses relatively high ionic conductivity and good stability towards lithium, one of the main

objectives of our study is to explore ways to further enhance the performance of glass electrolytes. As we mentioned in the **Introduction**, our goal is to “push the boundaries of ionic conductivity and Li/electrolyte interfacial stability in glassy sulfide electrolytes.” In other words, we hope to break the limit of ionic conductivity and Li/electrolyte interfacial stability in the existing $\text{Li}_3\text{PS}_4\text{-LiI}$ system.

Based on the discussion in our manuscript, the ionic conductivity, relative density, stability against lithium, and electronic conductivity of solid electrolytes will all influence their compatibility with Li metal anode. For glassy sulfides, which are inherently characterized by high relative density and low electronic conductivity, enhancing their ionic conductivity and stability against lithium is crucial for improving their compatibility with Li metal anode. As mentioned in the manuscript, the introduction of Li_4SiS_4 can suppress dimer formation, allowing more LiI to dissolve in the glass framework, thereby increasing the ionic conductivity of the system. Additionally, the glass electrolyte with higher LiI content due to the introduction of Li_4SiS_4 will help to form an interface containing more LiI after reacting with metallic lithium, which leads to the passivation of the Li/ Electrolyte interface and enhances the stability of the electrolyte against lithium. Apart from those, the introduction of Li_4SiS_4 into the system can further improve the ionic conductivity. As demonstrated in Fig. R23a (represented as Fig. 3d in the manuscript), the Li mean square displacement (MSD) value of $\text{Li}_{4.125}\text{P}_{0.785}\text{Si}_{0.125}\text{S}_4\text{I}$ glass is higher than that of $\text{Li}_4\text{PS}_4\text{I}$ glass, indicating a more unrestrained environment for Li^+ motion enabled by Li_4SiS_4 doping. Furthermore, compared with $\text{Li}_4\text{PS}_4\text{I}$ glass, Li^+ transport is promoted near the SiS_4^{4-} clusters in the $\text{Li}_{4.125}\text{P}_{0.785}\text{Si}_{0.125}\text{S}_4\text{I}$ glass, which is consistent with the previous report on the Si-substituted Li_3PS_4 system (Fig. R23b, represented as Fig. 3d in the manuscript).¹⁸ Therefore, we propose that Li_4SiS_4 doping will result in additional increases in ionic conductivity, thereby further enhancing the compatibility of the electrolyte with lithium metal and improving battery performance.

Fig. R23 a The Lithium MSD plot for $\text{Li}_4\text{PS}_4\text{I}$ (orange) and $\text{Li}_{4.125}\text{Si}_{0.125}\text{P}_{0.785}\text{S}_{4\text{I}}$ (blue) glass at 600 K from AIMD simulations. **b** The Li-ion probability densities of $\text{Li}_4\text{PS}_4\text{I}$ (top) and $\text{Li}_{4.125}\text{Si}_{0.125}\text{P}_{0.785}\text{S}_{4\text{I}}$ (bottom) glass from AIMD simulations.

Comment 3. On page 20, line 489, it was mentioned that when the C:G ratio is 7, the highest critical current density values are achieved with sufficient pellet density, high ion conductivity, and low electrical conductivity. Even if these conditions are met, a formation of continuous unstable interfaces with lithium metal could significantly degrade performance. Please describe the perspective of performance improvement related to the interface with lithium metal or interfacial products.

Response to comment 3:

Thanks for your valuable comments! It is true that the interfacial products play a significant role on the performance improvement.

Before addressing this issue, we would like to note that the description in Fig. 6a may not be appropriate when responding to the comment 1 of Review #1. When producing this diagram, our consideration of a stable/unstable interface refers to whether the formed interface can effectively regulate the deposition of metallic lithium to prevent lithium penetration. It does not imply the stability of the electrolyte against metallic lithium. Actually, based on the previous research, high-Cl content LPSC can exhibit good stability against Li metal without generating

continuous unstable interfaces.^{2, 3} Also, the stability of the filler P:Si = 6 glass against Li metal is proved to be superior in this study. For that reason, both high-Cl content LPSC and the composite electrolyte are stable against metallic lithium. Therefore, we have revised the description in Fig. 6a (Fig. R24) to avoid misleading the readers.

Fig. R24 The schematic diagram of the composite electrolyte design.

Due to the stability of high-Cl content LPSC and P:Si = 6 glass against Li metal, which does not result in the formation of a continuously deteriorating interface, the combination of high-Cl content LPSC and P:Si = 6 glass exhibits outstanding stability to metallic lithium, which effectively suppresses the continuous generation of interfacial by-products with low ionic conductivity, thereby enhancing the performance of the battery.

The corresponding perspective is described in the manuscript as follows:

-Manuscript, Results, Page 20.

“...Additionally, based on the previous research, high-Cl content argyrodite can exhibit good stability against Li metal without generating continuous unstable interfaces. Also, the stability of the filler P:Si = 6 glass against Li metal is proved to be superior in this study. For that reason, the composite electrolyte can maintain the outstanding stability against Li metal to suppress the side reactions.” Finally, the incorporation of P:Si = 6 glass with low electronic conductivity may contribute to a decrease in overall electronic conductivity. (Fig. 6a)”

-Manuscript, Results, Page 22.

“...This superior performance is attributed to the sufficiently-high relative density, ultra-high ionic conductivity, high stability against Li metal and low electronic conductivity of C:G = 7 electrolytes”

Comment 4. When using this glassy sulfide as a solid electrolyte (SE) layer, Li/Li batteries achieve the highest CCD values of 4.0 mA/cm² at a capacity of 0.1 mAh/cm². However, accurately determining the CCD value when evaluating batteries with a capacity of 0.1 mAh/cm² is challenging. (see Joule 6, 1770, 2022). This is because lithium can be formed and disappear

within the porous regions of the electrolyte layer. To demonstrate a CCD of 4 mA/cm^2 , a minimum capacity of 1 mAh/cm^2 should be evaluated.

Response to comment 4:

Thanks for your valuable advice! We agree with your suggestion that demonstrating a CCD of 4 mA cm^{-2} requires a deposited capacity surpassing 1 mAh cm^{-2} . Building on this understanding, we realized that claiming the CCD of P:Si=6 glass as 4.0 mA cm^{-2} in the **Abstract** and **Results** is unsuitable and might lead the readers astray.

In our experiments, we conducted two distinct CCD tests on the electrolyte. The first involved a fixed deposition/stripping time of 1 hour (Fig. R25a, represented as Fig. 4a in the manuscript), while the other employed a fixed capacity of 0.1 mAh cm^{-2} (Fig. R25b, represented as Fig. 4b in the manuscript). Our choice to perform the CCD testing with a fixed capacity of 0.1 mAh cm^{-2} was influenced by a recent work on modifying Li/SSE interface, which suggests that the polarization increase could stem from void accumulations near the interfaces.²⁸ To counteract the impact of voids, a charging/discharging capacity of 0.1 mAh cm^{-2} was set in this recent work. However, it is essential to recognize that the CCD value obtained from this kind of CCD test may not fully represent the practical scenario, where a larger capacity ($> 1.0 \text{ mAh cm}^{-2}$) is consistently required. Therefore, we believe that the CCD value of 1.2 mA cm^{-2} , with a capacity of 1.2 mAh cm^{-2} derived from the fixed-time CCD test, better reflects the performance of our optimal glass electrolyte. Additionally, it is important to note that the CCD value of 1.2 mA cm^{-2} remains one of the highest reported for Li sulfide glass electrolyte (Table S4 in the SI), which still demonstrates the superiority of our optimal glass among other glass sulfides.

Fig. R25 **a** The summary of CCD of synthesized series electrolytes. **b** Galvanostatic Li plating/stripping profiles in the Li|| P:Si = 6 ||Li symmetric cell at step-increased current densities. The capacity is fixed to be 0.1 mAh cm^{-2} .

Acknowledging this, we have made the revision to our manuscript as follows:

-Manuscript, Abstract, Page 2.

The misleading description of "CCD= 4.0 mA cm⁻²" has been deleted.

"The resulting glass electrolyte exhibits a record-high ionic conductivity ($2.21 \times 10^{-3} \text{ S cm}^{-1}$) among glassy SSEs. In addition, a novel glassy/crystalline composite electrolyte was designed to address the shortcomings of high-Cl content argyrodite-type sulfides by using our synthesized glass as the filler. The composite electrolyte exhibits a record-high CCD value of 2.9 mA cm⁻² among sulfide electrolytes, and demonstrates ultra-stable cycling at a current density of 1 mA cm⁻² and a practical capacity of 3 mAh cm⁻²."

-Manuscript, Results, Page 15.

"P:Si = 6 glassy electrolytes can stably function until the current density is raised to 4.0 mA cm⁻² when the discharging/charging capacity is fixed to 0.1 mAh cm⁻², indicating that P:Si = 6 electrolytes can withstand the high current density work condition as long as the severe contact loss does not occur (Fig. 4b). However, it should be noted that the CCD value derived from this fixed-capacity test cannot fully represent the practical scenario, where a larger capacity (> 1.0 mAh cm⁻²) is consistently required."

Comment 5. *The copper foil in Figure 5a is unsuitable as a current collector due to the inevitable spontaneous corrosion reaction between copper and Li₆PS₅Cl. Stainless steel (SUS) or nickel (Ni) foil is considered a more appropriate substrate.*

Response to comment 5:

Thanks for your professional advice! We are grateful to learn that copper is an inappropriate material for the test of Li stripping/plating from this comment. Considering the spontaneous corrosion reaction between copper and sulfide electrolytes, the conclusion we derived from the CE test of using Li/Cu half-cell may be misleading. Therefore, in our revision, we selected the stainless steel (SUS) as the substrate to test the CE. In detail, to ensure effective contact between the SUS and electrolyte, we employed a modest current density of 0.1 mA cm⁻² and a cut-off capacity of 0.1 mAh cm⁻² for Li stripping/plating on the SUS. Also, all half-cells will undergo 10 cycles of Li stripping/plating to determine the average CE. As a result, the highest initial CE value of 86.08 % is achieved in the P:Si = 6 glass while Li₆PS₅Cl and 75Li₂S- 25P₂S₅ glass demonstrates lower initial CE values of 76.50 % and 72.62 %, respectively (Fig. R26 and Fig. R27, denoted as Fig.5a in the manuscript and Fig. S24 in the SI respectively). Furthermore, as the cycle progresses, P:Si = 6 glass demonstrates the highest average CE value of 86.68%,

surpassing that of $\text{Li}_6\text{PS}_5\text{Cl}$ (75.75%) and $75\text{Li}_2\text{S}-25\text{P}_2\text{S}_5$ (76.28%). The corresponding voltage-capacity profiles of $\text{Li}|\text{SUS}$ half cells are provided in Fig.R. The obtained results exhibit a certain similarity to those obtained from testing with Li/Cu half-cells, where $\text{P}:\text{Si} = 6$ glass demonstrates the best stability towards lithium. This implies that the interface formed by the reaction of $\text{P}:\text{Si} = 6$ glass with metallic lithium, generating a substantial amount of LiI , exerts a strong passivation effect, effectively impeding further progress of side reactions. Simultaneously, we observed a notable difference in the initial CE of $\text{P}:\text{Si} = 6$ glass when using Li/sus half-cells, where it surpasses that of $\text{Li}_6\text{PS}_5\text{Cl}$ and $75\text{Li}_2\text{S}-25\text{P}_2\text{S}_5$. This deviation from the scenario observed in Li/Cu half-cells may stem from variations in the reactivity of Cu with different sulfide solid electrolytes, potentially leading to somewhat misleading initial CE results.

Fig. R26 The CE profiles of three electrolytes.

Fig. R27 The voltage profiles of Li||SS half cells using different electrolytes as the interlayer in the 1st, 5th and 10th cycles.

Based on these findings, **The corresponding revision in our paper is listed as follows:**

-Manuscript, Results, Page 16, Fig. 5a.

Fig. 5 The comparison of Li compatibility between glassy P:Si =6 and state-of-art sulfides. **a** The CE profiles of three electrolytes. **b** Long time galvanostatic cycling profiles of three electrolytes. **c** The profiles of the CCD test on $\text{Li}_6\text{PS}_5\text{Cl}$ -based Li symmetric cell and the in-situ X-ray CT morphology of the $\text{Li}_6\text{PS}_5\text{Cl}$ /Li interphase before and after the CCD test. The scale bar is 100 μm . **d** The profiles of the CCD test on $75\text{Li}_2\text{S}-25\text{P}_2\text{S}_5$ glass-based Li symmetric cell and the in-situ X-ray CT morphology of the $75\text{Li}_2\text{S}-25\text{P}_2\text{S}_5$ glass /Li interphase before and after the CCD test. The scale bar is 100 μm . **e** The profiles of the CCD test on P:Si = 6 glass-based Li symmetric cell and the in-situ X-ray CT morphology of the P:Si = 6 glass /Li interphase before and after the CCD test. The scale bar is 100 μm .

-Manuscript, Results, Page 17-18.

Due to the reactivity of Cu with sulfides, the results it reflects may be somewhat misleading. Therefore, we opted to replace the test data from Li/Cu half-cells with data from Li/SUS half-cells, **making the following modifications to the manuscript:**

“Then, Li stripping/plating Coulombic efficiency (CE) values in three solid electrolytes were evaluated by using the Li||Current Collector half-cell. Considering that the copper may react with sulfide electrolytes, the stainless steel (SS) rather than the copper is applied as the current collector material in the test. To ensure adequate contact between the SUS and electrolyte, a modest current density of 0.1 mA cm⁻² and a cut-off capacity of 0.1 mAh cm⁻² were applied for Li stripping/plating on the SUS. In the first cycle, the highest initial CE value of 86.08 % is achieved in the P:Si = 6 glass while Li₆PS₅Cl and 75Li₂S- 25P₂S₅ glass demonstrates lower initial CE values of 76.50 % and 72.62 %, respectively (Fig. 5a). Furthermore, as the cycle progresses, P: Si=6 glass demonstrates the highest average CE value of 86.68%, surpassing that of Li₆PS₅Cl (75.75%) and 75Li₂S- 25P₂S₅ (76.28%). The corresponding voltage-capacity profiles of Li||SUS half cells are provided in Fig.S24.”

Based on the test results from Li/SUS half-cells, we reconducted an analysis of the reaction mechanisms of the three electrolytes. **The details are as follows:**

“The reaction between sulfide electrolytes and Li is triggered by the reduction of high-valence cation. In the context of Li₆PS₅Cl and 75Li₂S- 25P₂S₅ electrolytes, their reactions with lithium metal stem from the reduction of P-cation. Owing to the presence of P₂S₆⁴⁻ anion clusters with a lower LUMO value in 75Li₂S- 25P₂S₅ glass, it is more prone to reduction by lithium metal, thereby exhibiting a lower initial CE compared to Li₆PS₅Cl (Fig. S25). However, the average CE value of Li₆PS₅Cl is lower than that of 75Li₂S- 25P₂S₅ glass, since Li would intrude into the non-dense Li₆PS₅Cl pellet with cracks and result in more electrolytes to be reduced. Through XPS analysis, it is observed that despite the P:Si = 6 glass containing Si⁴⁺, only P-cation in P:Si = 6 electrolytes would be reduced into Li-P species accompanied by the generation of Li₂S, LiI and Li-Si-P species (Fig. S26). This corresponds well with the previous AIMD simulations and spectroscopic characterizations of Li₇SiPS₈ systems,⁷ as SiS₄⁴⁻ would retain its stability when contacting with Li and no reduced Si can be found in the SEI while the PS₄³⁻ will be reduced into Li-P species. Against this backdrop, we contemplate the key distinction between the P:Si = 6 glass and other two electrolytes lies primarily in its markedly elevated halide (LiI) content. When the P:Si = 6 glass reacts with metallic lithium, a substantial amount of LiI is anticipated to form at the interface. This dissimilarity may provide an explanation for its exceptional stability. Based on prior research findings, Li-P species generated from the reaction between sulfides and metallic lithium exhibit a certain level of electronic conductivity, leading to a challenge in effectively suppressing side reactions. In comparison to Li-P species with a bandgap of 0.70-1.76 eV, LiI possesses a broader band gap (4.37 eV) (Fig.S27), which may help to diminish the overall electronic conductivity of interphase and thereby

suppressing the further side reaction.

References

1. Wan H, *et al.* Critical interphase overpotential as a lithium dendrite-suppression criterion for all-solid-state lithium battery design. *Nat Energy* **8**, 473-481 (2023).
2. Liu Y, *et al.* Revealing the Impact of Cl substitution on the crystallization behavior and interfacial stability of superionic lithium argyrodites. *Adv Func Mater* **32**, 2207978 (2022).
3. Zeng DW, *et al.* Promoting favorable interfacial properties in lithium-based batteries using chlorine-rich sulfide inorganic solid-state electrolytes. *Nat Commun* **13**, 1909 (2022).
4. Takahashi M, *et al.* Investigation of the suppression of dendritic lithium growth with a lithium-iodide-containing solid electrolyte. *Chem Mater* **33**, 4907-4914 (2021).
5. Sedlmaier SJ, *et al.* Li₄PS₄I: A Li⁺superionic conductor synthesized by a solvent-based soft chemistry approach. *Chem Mater* **29**, 1830-1835 (2017).
6. Han FD, Yue J, Zhu XY, Wang CS. Suppressing Li dendrite formation in Li₂S-P₂S₅ solid electrolyte by LiI incorporation. *Adv Energy Mater* **8**, 1703644 (2018).
7. Riegger LM, *et al.* Instability of the Li₇SiPS₈ solid electrolyte at the lithium metal anode and interphase formation. *Chem Mater* **34**, 3659-3669 (2022).
8. Hung PK, Vinh LT, Kien PH. The 'native vacancy' and interstitial site for gas solubility in amorphous solid. *Journal of Non-Crystalline Solids* **356**, 1000-1005 (2010).
9. Sakuda A, Hayashi A, Tatsumisago M. Sulfide solid electrolyte with favorable mechanical property for all-solid-state lithium battery. *Sci Rep* **3**, 2261 (2013).
10. Liu S, *et al.* An Inorganic-Rich Solid Electrolyte Interphase for Advanced Lithium-Metal Batteries in Carbonate Electrolytes. *Angew Chem Int Ed Engl*, (2020).
11. Xing LD, *et al.* Deciphering the Ethylene Carbonate-Propylene Carbonate Mystery in Li-Ion Batteries. *Acc Chem Res* **51**, 282-289 (2018).
12. Ong SP, Mo Y, Richards WD, Miara L, Lee HS, Ceder G. Phase stability, electrochemical stability and ionic conductivity of the Li_{10±1}MP₂X₁₂ (M = Ge, Si, Sn, Al or P, and X = O, S or Se) family of superionic conductors. *Energy Environ Sci* **6**, 148-156 (2013).
13. Zhu YZ, He XF, Mo YF. First principles study on electrochemical and chemical stability of solid electrolyte-electrode interfaces in all-solid-state Li-ion batteries. *J Mater Chem A* **4**, 3253-3266 (2016).
14. Mo Y, Ong SP, Ceder G. First Principles Study of the Li₁₀GeP₂S₁₂ Lithium Super Ionic Conductor Material. *Chem Mater* **24**, 15-17 (2011).
15. Kraft MA, *et al.* Influence of Lattice Polarizability on the Ionic Conductivity in the Lithium Superionic Argyrodites Li₆PS₅X (X = Cl, Br, I). *J Am Chem Soc* **139**, 10909-10918 (2017).

16. Fuhrman Javitt L, *et al.* Electro-Freezing of Supercooled Water Is Induced by Hydrated Al³⁺ and Mg²⁺ Ions: Experimental and Theoretical Studies. *J Am Chem Soc* **145**, 18904-18911 (2023).
17. Li JR, *et al.* Fluorinated Interface Layer with Embedded Zinc Nanoparticles for Stable Lithium-Metal Anodes. *ACS Appl Mater Interfaces* **13**, 17690-17698 (2021).
18. Zhou LD, *et al.* An entropically stabilized fast-ion conductor: Li_{3.25}[Si_{0.25}P_{0.75}]S₄. *Chem Mater* **31**, 7801-7811 (2019).
19. Krzeszewski M, *et al.* Dipole Effects on Electron Transfer are Enormous. *Angew Chem Int Ed Engl* **57**, 12365-12369 (2018).
20. Wang S, *et al.* Lithium Chlorides and Bromides as Promising Solid-State Chemistries for Fast Ion Conductors with Good Electrochemical Stability. *Angew Chem Int Ed Engl* **58**, 8039-8043 (2019).
21. Zhang Z, *et al.* Na₁₁Sn₂PS₁₂: a new solid state sodium superionic conductor. *Energy Environ Sci* **11**, 87-93 (2018).
22. Wang ZX, *et al.* Doping effects of metal cation on sulfide solid electrolyte/lithium metal interface. *Nano Energy* **84**, 105906 (2021).
23. Bai Y, Zhao YB, Li WD, Meng LH, Bai YP, Chen GR. New insight for solid sulfide electrolytes LSiPSI by using Si/P/S as the raw materials and I doping. *ACS Sustain Chem Eng* **7**, 12930-12937 (2019).
24. Krauskopf T, Richter FH, Zeier WG, Janek J. Physicochemical concepts of the lithium metal anode in solid-state batteries. *Chem Rev* **120**, 7745-7794 (2020).
25. Li JR, *et al.* A deformable dual-layer interphase for high-performance Li₁₀GeP₂S₁₂-based solid-state Li metal batteries. *Chem Eng J* **431**, 134019 (2022).
26. Wang CH, *et al.* Stabilizing interface between Li₁₀SnP₂S₁₂ and Li metal by molecular layer deposition. *Nano Energy* **53**, 168-174 (2018).
27. Zhao FP, *et al.* A versatile Sn - substituted argyrodite sulfide electrolyte for all - solid - state Li metal batteries. *Adv Energy Mater* **10**, 1903422 (2020).
28. Wang T, *et al.* A self-regulated gradient interphase for dendrite-free solid-state Li batteries. *Energy Environ Sci* **15**, 1325-1333 (2022).

REVIEWER COMMENTS

Reviewer #1 (Remarks to the Author):

The manuscript was well-revised, but still need to improve.

The authors claimed the pellet was no crack in Fig. S13a, however, the crack was observed in the SEM image. Please add more SEM images, not only one image, to clarify the absent of cracks.

Second point is that the authors claimed electric modulus, but there is no data about the elastic modulus. The data of elastic modulus should be added.

Here are other comments.

1. In the introduction, "formation of S-S dimers and dimers", the expression leads to misunderstanding. Please correct it.
2. In Fig. S15, it is better to show EDS of pellets.

Reviewer #2 (Remarks to the Author):

The authors have carefully and thoroughly responded to my concerns and those of the other referees. The erroneous conclusions and analyses have been corrected, and I am generally satisfied with the authors' revisions.

One [minor] comment: I realize Fig. 2 is already quite crowded, but the authors may want to consider moving Fig. S11b there as well. I think the discussion of "crowded" sites is important, and the new discussion on p. 9-10 would benefit from having it in the main text. Fig. S11b is to me the most obvious visual cue.

Reviewer #3 (Remarks to the Author):

The authors have addressed my concerns regarding battery performances, such as CCD measurement and the suitable choice of current collector. However, there are still unclear points in your revised manuscript that need to be addressed before publication.

- On page 50 in the response letter, the authors explained that proper Si doping into the sulfide electrolyte will not form the mixed conducting interface. However, I cannot agree with this statement until you provide accurate evidence. It is ambiguous.
- I recommend deleting $\text{CCD} = 4.0 \text{ mA cm}^{-2}$ in Figure 4b.
- You proposed a combination design of P:Si = 6 and LPSCI. Why? What are the limitations of using only glassy electrolyte?
- Please demonstrate the chemical information of the formed interface of P:Si = 6 glassy electrolyte after contacting with Li metal by using XPS.
- I recommend that the authors carefully revise the explanation or expression of your electrochemical performances overall to avoid misleading the readers.

Manuscript Title (ID: NCOMMS-23-41433): Deciphering the Critical Role of Interstitial Volume in Glassy Sulfide Superionic Conductors

Reviewer 1#

The manuscript was well-revised, but still need to improve.

Response: Thank you for your recognition. Your professional feedback has significantly contributed to the improvement of the quality of our article. In particular, your suggestions regarding the characterization of electrolytes have been invaluable to us. We hope our responses can meet your expectations.

Comment 1. The authors claimed the pellet was no crack in Fig. S13a, however, the crack was observed in the SEM image. Please add more SEM images not only one image, to clarify the absent of crackes.

Response: Thank you for your valuable comments! Indeed, after you pointed out the issue of "no crack," we realized the inappropriate use of language in our writing. In practice, for cold-pressed pellets, the density is approximately 93.6%, indicating that there may still be some cracks and defects. Therefore, we have modified the improper term "crack-free" in the manuscript to "flat and dense" to avoid misleading readers. Once again, we appreciate your thorough review.

Below is the corresponding revision in our manuscript:

"Additionally, the pellet of the P:Si = 6 electrolyte exhibited a flat and dense surface upon cold-pressing (Fig. S13a). By hot-pressing the pellet at around the T_g overnight, a highly densified pellet with a relative density of ~100% was obtained (Fig. S13b)."

Comment 2. Second point is that the authors claimed electic modulus, but there is no data about the elastic modulus. The data of elastic modulus should be added.

Response: Thank you for your valuable comments! Currently, the most reliable approach of measuring bulk Young's modulus of sulfide solid electrolyte is the ultrasonic pulse echo method proposed by Prof. Tatsumisago.¹ However, the testing platforms available in our laboratory and university cannot realize the measuring of air-sensitive samples by using this method. Therefore, we regretfully cannot provide the relevant data. Despite this, through a thorough literature review, we discovered that Prof. Tatsumisago's research group has systematically measured

the Young's modulus of the Lil-doped $75\text{Li}_2\text{S}-25\text{P}_2\text{S}_5$ glassy electrolyte systems.² As shown in Table S1, the Young's modulus of the glassy electrolyte in the $75\text{Li}_2\text{S}-25\text{P}_2\text{S}_5$ electrolytes decreases with increasing Lil doping. Consequently, as depicted in Fig. R1, the relative density of the $75\text{Li}_2\text{S}-25\text{P}_2\text{S}_5$ electrolyte with Lil doping is higher compared to that without Lil.

In our manuscript, we mentioned, "*P:Si = 6 demonstrates a higher relative density than $75\text{Li}_2\text{S}-25\text{P}_2\text{S}_5$. This is primarily attributed to the halogen dopant Lil, which can effectively reduce the elastic modulus of sulfide electrolytes, thereby enhancing the relative density of cold-pressed electrolyte pellets.*" We believe that such an inference can be supported by Prof. Tatsumisago's research results.

Table S1. The Young's modulus of $(100-y)(0.75\text{Li}_2\text{S}-0.25\text{P}_2\text{S}_5)\cdot y\text{LiI}$ glass pellets.²

Sample	Young's Modulus
$75\text{Li}_2\text{S}-25\text{P}_2\text{S}_5$	21
$95(0.75\text{Li}_2\text{S}-0.25\text{P}_2\text{S}_5)-0.05\text{LiI}$	19
$90(0.75\text{Li}_2\text{S}-0.25\text{P}_2\text{S}_5)-0.10\text{LiI}$	19
$80(0.75\text{Li}_2\text{S}-0.25\text{P}_2\text{S}_5)-0.20\text{LiI}$	18
$70(0.75\text{Li}_2\text{S}-0.25\text{P}_2\text{S}_5)-0.30\text{LiI}$	17

Fig. R1 SEM images of the cross sections of $(100-y)(0.75\text{Li}_2\text{S}-0.25\text{P}_2\text{S}_5)\cdot y\text{LiI}$ glass pellets pressed at 360 MPa: (a) $y = 0$ mol %; (b) $y = 30$ mol %.²

Comment 3. In the introduction, "formation of S-S dimers and dimers" the expression leads to misunderstanding. Please correct it.

Response: Thank you for your valuable comments! We have changed all the P-P dimers and dimers in the manuscript to $\text{P}_2\text{S}_6^{4-}$ and $\text{P}_2\text{S}_7^{4-}$, respectively.

Comment 4. In Fig.S15, it is better to show EDS of pellets.

Response: Thank you for your valuable comments! We have added the EDS of the pellet in the SI (Fig. R2, denoted as Fig. S16 in the SI).

Fig. R2 The surface morphology of the P:Si =6 pellet and the corresponding EDS mapping results.

Below is the corresponding revision in our manuscript:

“Furthermore, energy dispersive spectroscopy (EDS) mapping results demonstrate a homogenous distribution of elements in the micro-particle and the pellet of the P:Si = 6 electrolyte (Fig. S15-S16).”

Reviewer 2#

The authors have carefully and thoroughly responded to my concerns and those of the other referees. The erroneous conclusions and analyses have been corrected, and I am generally satisfied with the authors' revisions.

Response: Thank you for your recognition. Your professional feedback has significantly contributed to the improvement of the quality of our article, particularly in the realm of theoretical studies. We have learned a lot during the interactions with you.

One [minor] comment: I realize Fig. 2 is already quite crowded, but the authors may want to consider moving Fig. S11b there as well. I think the discussion of "crowded" sites is important, and the new discussion on p. 9-10 would benefit from having it in the main text. Fig. S11b is to me the most obvious visual cue.

Response: Thank you for your valuable comments! We have moved the Fig. S11b to the Fig. 2. Below is the revised Fig. 2.

Fig. 2 Investigations of anion clusters in the sulfide glass after the Si-incorporation. **a** The $g(r)$ profiles of P-P and P-S pair in the equilibrium Li_4PS_4 glass after AIMD simulations. **b** The $g(r)$ profiles of P-P, P-S and Si-S pair in the equilibrium Si-doped $Li_{4.125}Si_{0.125}P_{0.785}S_4$ glass after AIMD simulations. **c** The $g(r)$ profile of S-I pair in Li_4PS_4 glass (without Si) and Si-doped $Li_{4.125}Si_{0.125}P_{0.785}S_4$ glass (with Si). **d,e** The snapshot of the equilibrium Li_4PS_4 glass after AIMD simulations (**d**) and the snapshot of the equilibrium Si-doped $Li_{4.125}Si_{0.125}P_{0.785}S_4$ glass after AIMD simulations (**e**). The glass framework is represented as the solid gray lines. The light purple balls represent P atoms, the blue balls represent Si atoms, the orange balls represent

S atoms dissolved in the interstitial sites, the yellow balls represent S atoms in the framework and the deep brown ball represent I atoms. **f** Raman, ^{31}P and ^{29}Si ss-NMR spectra of blank, P:Si =6 and P:Si =4 electrolytes.

g Ratios of anion clusters and minimum $V_{\text{Li}_2\text{S}_x}$ derived from Raman, ^{31}P and ^{29}Si ss-NMR spectra.

Reviewer 3#

The authors have addressed my concerns regarding battery performances, such as CCD measurement and the suitable choice of current collector. However, there are still unclear points in your revised manuscript that need to be addressed before publication.

Response: Thanks for your valuable suggestions! Your suggestions have contributed a lot to making our article more rigorous. We have addressed each point in our responses below, and we hope that you will be satisfied with our responses.

Comment 1. *On page 50 in the response letter, the authors explained that proper Si doping into the sulfide electrolyte will not form the mixed conducting interface. However, I cannot agree with this statement until you provide accurate evidence. It is ambiguous.*

Reply to comment 1: Thanks for your advice! Following your correction, we have acknowledged that the statement "Proper Si doping will not form MCI" may be inaccurately expressed. In fact, our decision to incorporate Si doping is based on a previous study examining the impact of different metal or metalloid cations doping on the anode stability of the $\text{Li}_7\text{P}_3\text{S}_{11}$ system. As point out by this study, the proper doping of Si did not significantly deteriorate the Li/Solid electrolyte interface and improved the CCD value of the $\text{Li}_7\text{P}_3\text{S}_{11}$ system.³ A similar phenomenon can be found in another previous work on LiSiPSI electrolytes, where the symmetric cells using LiSiPSI electrolytes did not experience a notable increment of voltage polarization.⁴ However, it is not appropriate to conclude that those electrolytes with Si doping will not form MCI when in contact with Li. Therefore, we have changed the expression to "Proper doping amounts of SiS_2 in the $\text{Li}_2\text{S-P}_2\text{S}_5$ system will not result in a severe deterioration of the Li/electrolyte interface, as previously reported".

Comment 2. *I recommend deleting $\text{CCD} = 4.0 \text{ mA cm}^{-2}$ in Figure 4b.*

Reply to comment 2: Thank you for your advice! The " $\text{CCD} = 4.0 \text{ mA cm}^{-2}$ " has been deleted in Fig. 4b. Below is the revised Fig. 4b.

Fig. 4 Li compatibility evaluation for P:Si = 6 electrolytes. **a** The summary of CCD of synthesized series electrolytes. **b** Galvanostatic Li plating/stripping profiles in the Li|| P:Si = 6 ||Li symmetric cell at step-increased current densities. The capacity is fixed to be 0.1 mAh cm⁻². **c** Galvanostatic Li plating/stripping profiles in the Li|| P:Si = 6 ||Li symmetric cell at a current density of 0.1 mA cm⁻² and a cut-off capacity of 0.5 mAh cm⁻². **d** Galvanostatic Li plating/stripping profiles in the Li|| P:Si = 6 ||Li symmetric cell at a current density of 1.0 mA cm⁻² and a cut-off capacity of 0.1 mAh cm⁻².

Comment 3. You proposed a combination design of P:Si = 6 and LPSCI. Why? What are the limitations of using only glassy electrolyte?

Reply to comment 3: Thanks for your question! Firstly, we would like to address the limitation of using only glassy electrolyte. Despite the notable benefits of our optimal glass in relative density and electronic conductivity over high-Cl content LPSC, the CCD level (1.2 mA cm⁻² with a capacity of 1.2 mAh cm⁻²) of our optimal glass is only marginally higher than that of high-Cl content LPSC (1.1 mA cm⁻² with a capacity of 1.1 mAh cm⁻²). This is primarily due to the lower ionic conductivity of our glass compared to high-Cl content LPSC, resulting in a higher Li nucleation overpotential at the interface. The LPSC/glassy composite structure integrates the high relative density and low electronic conductivity attributes of our optimal glass with the high ionic conductivity attributes of high-Cl content LPSC electrolyte, which not only mitigates mechanical failures of the electrolyte but also ensures a reduction in the overpotential for Li nucleation at the interface. As a result, the composite electrolyte exhibits a CCD value of 2.9 mA cm⁻², which is far higher than that of the initial glass electrolyte and high-Cl content LPSC.

Comment 4. Please demonstrate the chemical information of the formed interface of P:Si = 6 glassy electrolyte after contacting with Li metal by using XPS.

Reply to comment 4: Thanks for your valuable advice! We have performed the XPS characterization on the Li / P:Si =6 interface. Through XPS analysis, it is observed that despite the P:Si = 6 glass containing Si^{4+} , only PS_4^{3-} in P:Si = 6 electrolytes would be reduced into Li-P species accompanied by the generation of **Li_2S , LiI and Li-P species** (Fig. R3, denoted as Fig. S27 in the SI). This corresponds well with the previous AIMD simulations and spectroscopic characterizations of the $\text{Li}_7\text{SiPS}_8/\text{Li}$ interface in Prof. Janek's work,⁵ that no evident elemental Si was detected in the interphase, and the interfacial products of Li and Li_7SiPS_8 mainly consisted of Li-P species.

Fig. R3. The I 3d, S 2p, Si 2P, P 2P spectra of Li/P:Si =6 interface.

Comment 5. I recommend that the authors carefully revise the explanation or expression of your electrochemical performances overall to avoid misleading the readers.

Reply to comment 5: Thank you for your advice! We have thoroughly reviewed all expressions related to electrochemical performance, including the introduction, abstract, discussion, and conclusion, ensuring that the existing expressions do not mislead the readers. All modified statements are highlighted in yellow throughout the manuscript.

References

1. Sakuda A, Hayashi A, Tatsumisago M. Sulfide solid electrolyte with favorable mechanical property for all-solid-state lithium battery. *Sci Rep* **3**, 2261 (2013).
2. Kato A, Yamamoto M, Sakuda A, Hayashi A, Tatsumisago M. Mechanical properties of $\text{Li}_2\text{S}-\text{P}_2\text{S}_5$ glasses with lithium halides and application in all-solid-state batteries. *ACS Appl Energy Mater* **1**, 1002-1007 (2018).
3. Wang ZX, *et al.* Doping effects of metal cation on sulfide solid electrolyte/lithium metal interface. *Nano Energy* **84**, 105906 (2021).
4. Bai Y, Zhao YB, Li WD, Meng LH, Bai YP, Chen GR. New insight for solid sulfide electrolytes LSiPSI by using Si/P/S as the raw materials and I doping. *ACS Sustain Chem Eng* **7**, 12930-12937 (2019).
5. Riegger LM, *et al.* Instability of the Li_7SiPS_8 solid electrolyte at the lithium metal anode and interphase formation. *Chem Mater* **34**, 3659-3669 (2022).

REVIEWERS' COMMENTS

Reviewer #1 (Remarks to the Author):

The manuscript is well revised.

Reviewer #3 (Remarks to the Author):

The authors have incorporated the reviewer's suggestions into the revised manuscript. Therefore, I recommend that the work be published.